# Multilayer regulation underlies the functional precision and evolutionary potential of the olfactory system

Jérôme Mermet [1], Steeve Cruchet[1], Asfa Sabrin Borbora[1], Daehan Lee [1,2], Phing Chian Chai[1], Andre Jang [3], Karen Menuz [3,4] & Richard Benton [1] ✉

Sensory neurons must be reproducibly specified to permit accurate neural representation of external signals but also able to change during evolution. We studied this paradox in the *Drosophila* olfactory system by establishing a single-cell transcriptomic atlas of all developing antennal sensory lineages, including latent neural populations that normally undergo programmed cell death (PCD). This atlas reveals that transcriptional control is robust, but imperfect, in defining selective sensory receptor expression. A second layer of precision is afforded by the intersection of expression of functionally-interacting receptor subunits. A third layer is defined by stereotyped PCD patterning, which masks promiscuous receptor expression in neurons fated to die and removes "empty" neurons lacking receptors. Like receptor choice, PCD is under lineage-specific transcriptional control; promiscuity in this regulation leads to previously-unappreciated heterogeneity in neuronal numbers. Thus, functional precision in the mature olfactory system belies developmental imprecision that might facilitate the evolution of sensory pathways.

Sensory systems mediate detection of the environment and provide the brain with a spatio-temporal code of neuronal activity that enables recognition, interpretation, and appropriate behavioral responses to a stimulus. However, the external world of stimuli shifts as species colonize new ecological niches. Thus, sensory systems must also be capable of change over evolutionary timescales.

This paradox of functional precision but evolutionary flexibility is particularly notable in the olfactory system of *Drosophila melanogaster*. Intensive anatomical, molecular and functional analyses of the major olfactory organ, the third antennal segment (hereafter, antenna), have defined a highly stereotyped organization in which ~1200 neurons are categorized into nearly 50 distinct classes of olfactory sensory neurons (OSNs), as well as several types of hygrosensory and thermosensory neurons[1–8]. Each class of sensory neuron is characterized by the expression of a specific "tuning" receptor (belonging to the Odorant receptor (Or), Ionotropic receptor (Ir) or so-called "Gustatory" receptor (Gr) families), which defines sensory specificity, together with one or more broadly-expressed "co-receptors" (Orco for Ors; Ir8a, Ir25a, Ir76b and Ir93a for Irs). A few classes of neurons express more than one tuning receptor, typically encoded by tandemly-arranged gene duplicates. OSNs are grouped in stereotyped combinations of 1-4 neurons underlying sensory hairs (sensilla) of several distinct morphological classes: antennal basiconic (ab, large and small), trichoid (at), intermediate (ai) and coeloconic (ac) (Fig. 1a). Hygrosensory neurons (and some OSNs) are housed in sensilla within the sacculus, an internal multi-chambered structure, while thermosensory neurons are housed in the arista and sacculus (Fig. 1a). The ciliated dendrites of sensory neurons are housed in the sensillar hair, while the axons project to the antennal lobe in the brain, where they innervate a glomerulus unique to each type of neuron.

The organization of the mature olfactory system is thought to derive from hard-wired developmental mechanisms. Each sensillum

[1]Center for Integrative Genomics, Faculty of Biology and Medicine, University of Lausanne, Lausanne, Switzerland. [2]Department of Biological Sciences, Sungkyunkwan University, Suwon, Republic of Korea. [3]Department of Physiology and Neurobiology, University of Connecticut, Storrs, CT, USA. [4]Connecticut Institute for Brain and Cognitive Sciences, University of Connecticut, Storrs, CT, USA. ✉e-mail: Richard.Benton@unil.ch

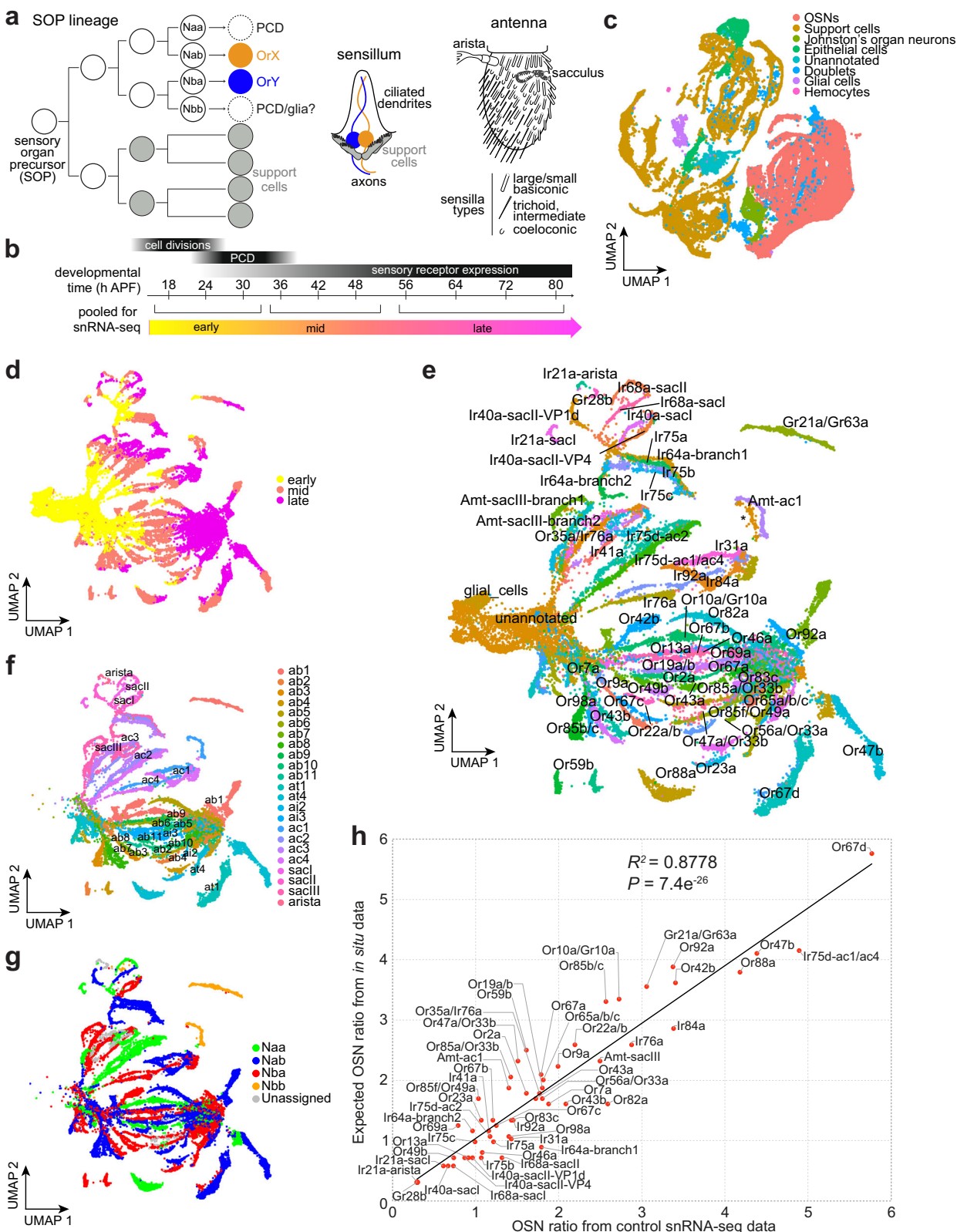

develops from a single sensory organ precursor (SOP) cell that is specified in the antennal imaginal disk in the larva (Fig. 1a). The canonical view is that an SOP gives rise to a short, fixed lineage of asymmetric cell divisions that produces eight terminal cells with distinct molecular identities[9–11]. Four of these become support cells (which have functions in sensillum construction and secretion of perireceptor proteins[12]), while the other four – termed Naa, Nab, Nba,

or Nbb – can potentially give rise to OSNs[9,11]. Although two sensillum types contain 4 OSNs, all other sensilla house fewer neurons. This is thought to be due to programmed cell death (PCD) of precursor cells during the pupal stage[9–11]; in many ac lineages one cell, likely the Nbb precursor, has been proposed to differentiate as a glial cell[10,13–15]. Abundant evidence supports the contribution of OSN-specific gene regulatory networks in defining the fate of surviving neurons, notably

**Fig. 1 | A developmental atlas of antennal sensory neurons. a** Development and anatomy of the *D. melanogaster* peripheral olfactory system. **b** Experimental design: antennal imaginal discs/antennae were dissected at time points every 6-8 h from 18-80 h after puparium formation (APF) in control (*peb-Gal4/+;;UAS-unc84:GFP/+*) and PCD-blocked (*peb-Gal4/+;UAS-p35/+;UAS-unc84:GFP/+*) conditions. Samples were pooled into three temporal phases (early, mid, late) prior to FACS and 10x sequencing. Tapered black bars indicate the timing of the main developmental processes. **c** UMAP of all cell types in the developmental atlas, integrating control and PCD-blocked datasets. **d** UMAP of all olfactory sensory neurons in (**c**) in the atlas – integrating control and PCD-blocked datasets – colored by developmental phases. **e** UMAP of annotated neuronal lineages, integrating control and PCD-blocked datasets. Unannotated neurons could not be assigned to

any lineage; almost all of these are from early developmental phases, but one cluster (specific to the PCD-blocked dataset) was detected in late developmental phases (asterisk, close to the Amt-ac1 lineage). Some *repo* expressing glial cells were also detected (see Supplementary Fig. 3a, b). **f** UMAP of the cells in **e** – masking unannotated neurons and glial cells – colored by sensillar class (ab, antennal basiconic; at, antennal trichoid; ai, antennal intermediate; ac, antennal coeloconic; sac, sacculus). **g** UMAP of the same cells as in **f** colored by neuron precursor type. **h** Scatter plot of the relative abundance of each neuronal population in the developmental atlas (control dataset only) with their relative abundance as quantified in situ[8]. See Source Data. Black line = linear regression fit. The coefficient of determination ($R^2$) and *P* value (F-test) are indicated.

in the precise transcriptional activation (or inhibition) of receptor genes[16–19]. Such deterministic transcriptional codes are thought to be central to the functional stereotypy of the olfactory system.

Comparison of the *D. melanogaster* olfactory system with other insects, however, reveals remarkable evolvability, with changes in receptor function, receptor expression and OSN number, often linked to adaptation of species to new ecological niches[20–22]. For example, in *Drosophila sechellia*, an extreme specialist on noni fruit, olfactory channels detecting the host fruit exhibit altered receptor tuning and expanded OSN populations[23–25].

The generation of new receptors through tandem gene duplication and functional diversification through sequence changes are conceptually straightforward processes that are well-documented[18,26,27]. By contrast, how novel cell types within a sensillum might emerge is much less well-understood. One clue came from the demonstration that blocking PCD is sufficient to result in the formation of functional OSNs[28], implying a latent potential of OSNs fated to die to evolve as new cell types. A deeper understanding of this potential is precluded by our almost complete lack of knowledge of how PCD is patterned in the developing OSN lineages and the molecular properties of individual neurons fated to die.

In this work, we generate a high-resolution, developmental atlas of the antennal neuronal lineages, encompassing those that become functional neurons as well as those that undergo PCD. We use this atlas to define the first molecular determinants specifying PCD of OSNs. Notably, we also discovered previously overlooked promiscuity in the patterning of receptor expression and PCD in the olfactory system, providing clues as to how it might adapt during evolution.

## Results

### An atlas of developing and dying antennal sensory neurons

To generate a high-resolution, comprehensive, spatio-temporal atlas of the antennal OSN lineages, we labeled these cells by using *pebbled-Gal4* (*peb-Gal4*), which is expressed in all neural lineages (as well as many non-neuronal cells) from before 18 h after puparium formation (APF)[29], to drive a nuclear GFP reporter (*UAS-unc84:GFP*)[30]. In parallel, to characterize the developmental potential of neurons that are ultimately lost due to PCD during ~22–32 APF[9,10,15,28], we blocked OSN death by using *peb-Gal4* to also drive *UAS-p35*, encoding the baculoviral P35 caspase inhibitor[28]. For both control and PCD-blocked genotypes, we dissected antennal tissues from animals sampled every 6-8 h from 18-80 h APF, spanning the vast majority of their development (Fig. 1b). Antennae were pooled into "early" (18-30 h APF), "mid" (36-48 h APF) and "late" (56-80 h APF, with one exception, see Methods) developmental phases prior to FACS-based isolation of GFP-positive nuclei (Fig. 1b). Using the 10x Genomics Chromium platform, we sequenced the transcriptomes of ~54k and ~32k nuclei from control and PCD-blocked antennae, respectively, detecting on average ~1000 genes/nucleus (Supplementary Fig. 1a).

Unless mentioned otherwise (see Methods and legends), for all downstream analyses we integrated control and PCD-blocked datasets, assuming that the vast majority of cells would form equivalent

clusters in these datasets, and that a smaller number of "undead cells" would potentially form clusters unique to the PCD-blocked dataset. To broadly catalog antennal cell types, we used marker genes extracted from the Fly Cell Atlas[7] (see Methods) (Supplementary Fig. 1b–d). After excluding Johnston's organ mechanosensory neurons, we found that sensory neurons represent ~39% of cells (~21,000) in the control dataset and ~43% (~14,000 cells) in the PCD-blocked dataset (Supplementary Fig. 1e), consistent with the latter containing many undead neurons (considered in more detail below). The remaining cells in our datasets mostly represent sensillar support cells (Fig. 1c and Supplementary Fig. 1e), suggesting that *peb-Gal4* labels both neuronal and non-neuronal branches of the SOP lineages; the latter cell types can be investigated in future studies.

We first annotated sensory neurons within these datasets by developmental phase (Fig. 1d and Supplementary Fig. 2). All "branches" of cell clusters comprised a continuum through early-, mid- and late-pupal stages (Fig. 1d), each presumably reflecting the development of different neuronal lineages. Each phase has distinct transcriptional profiles (Supplementary Fig. 2): early-pupal stage neuronal markers were enriched for genes involved in translation (likely reflecting enhanced protein synthesis capacity); mid-pupal stage neuronal markers were enriched in genes involved in signaling, cell-adhesion, axonogenesis and ion transport (concordant with the wiring of antennal neurons in the brain[31]); late-pupal stage neurons expressed higher levels of genes involved in ion transport and synaptic transmission (consistent with mature cell functions in neuronal signaling). Cells from the mid-developmental phase appeared to be the most transcriptionally divergent between lineages, with late-stage neurons converging to a more similar gene expression profile (as noted previously[5]) (Fig. 1d).

### High-resolution annotation of developing sensory neurons

We subclustered these lineages at high resolution, annotating many clusters based on the (chemo)sensory receptor and co-receptor gene(s) expressed in cells of the control dataset (Supplementary Fig. 3). To annotate cells from earlier developmental stages – which mostly lack expression of a diagnostic receptor gene – we used an iterative, retrograde annotation method (see Methods). In brief, marker genes were extracted for each individual neuron cluster – and/or groups of neurons housed in the same sensillum ("sensillar markers") – and used to identify and annotate additional clusters. These clusters were used as sources of additional earlier marker genes (see example iterations in Supplementary Fig. 4). Ultimately, we could annotate ~90% of neurons (Fig. 1e); most of the remaining cells correspond to the earliest time points, which were difficult to distinguish transcriptionally. A subset of these early cells expresses glial markers (Fig. 1e and Supplementary Fig. 3b); these might correspond to the Nbb-derived glia within the neuronal lineages[10,13,14]. The ~19,000 annotated neurons in our control atlas represent more than 15-fold coverage of this sensory organ (~1200 neurons/antenna)[2,32].

Our annotation of sensory neurons allowed us to document the complements of cell adhesion molecules, neurotransmitter receptors

and ion channels for individual neuron types (Supplementary Fig. 5), substantially extending previous analyses[5]. Such information might point to additional molecules defining the specific anatomical and functional properties of different sensory channels. In the context of understanding the development of the olfactory system, we were also able to extract markers for all sensillar classes (Fig. 1f and Supplementary Fig. 6), and those distinguishing the four neuron precursor types across essentially all lineages (Fig. 1g and Supplementary Fig. 7). The latter were enriched for genes encoding neural guidance molecules (Supplementary Fig. 7c and Supplementary Data 1), concordant with the segregation of OSNs from different precursor types in the antennal lobe[10].

Our atlas encompasses all documented antennal sensory channels[8]. This contrasts with previous single cell/nuclear RNA-sequencing (sc/snRNA-seq) analyses of the *D. melanogaster* developing antenna, which were able to match only about one-third of known OSN types across three time points (24 h APF, 48 h APF, and adult)[5,6] due to limited cell numbers. To further assess the completeness of our dataset, we compared the relative abundance of each sensory neuron class in the control dataset with that expected by in situ analysis (typically using RNA FISH or sensory receptor promoter reporter lines)[8]. These values exhibited a remarkably strong linear relationship, indicating the highly quantitative nature of cell representation in our datasets (Fig. 1h).

### Essential lineage-specific transcription factors

As a first assessment of the predictive ability of our developmental atlas and to understand how precision in the olfactory map arises, we examined transcription factors (TFs). Across pupal stages each neuronal population expresses a unique, though often highly overlapping, combination of 138 TF genes (Supplementary Fig. 8). We tested the functional relevance of TF expression through study of three very narrowly-expressed TF genes, which we reasoned might have selective and non-redundant roles: *lozenge* (*lz*), encoding an AML/Runt family TF, which is expressed exclusively in ab10 Or67a and Or85f neurons[3,8], and the tandemly-organized paralogs *ladybird early* (*lbe*) and *ladybird late* (*lbl*), encoding NK-like homeobox TFs, which are co-expressed in Or23a and Or83c neurons in ai2 (Fig. 2a and Supplementary Fig. 8).

In all cases TF expression precedes that of the *Or* genes (Fig. 2b–e). We first validated these predicted expression patterns in situ, revealing selective expression of the TFs in the expected neuron populations (Fig. 2f, i). We knocked down TF expression by using *peb-Gal4* to drive TF RNAi transgenes (Fig. 2g, h and 2j, k). *lz*[RNAi] led to severe or complete loss of expression of *Or67a* and *Or85f*, respectively, while not affecting a control neuron population expressing *Or67b* housed in another small basiconic sensillum. *lbe*[RNAi] had, respectively, a mild or very strong effect on expression of *Or23a* and *Or83c*. While *lbl*[RNAi] alone barely affected the expression of either receptor, it strongly enhanced the *lbe*[RNAi] phenotype, indicating partial redundancy of these TFs in ai2 neurons. This function appears to be specific, as a control receptor Or43a (from ai3) was unaffected (Fig. 2k). Together, these data demonstrate an essential and selective role of *lz* and *lbe/lbl* in controlling ab10 and ai2 OSN development, respectively.

As *lz* and *lbe/lbl* are common to both OSNs within their respective sensilla, we hypothesized that these TFs act prior to the last lineage division producing these distinct OSN types, rather than directly in receptor expression. To test this hypothesis, we repeated the TF RNAi experiment in combination with a thermosensitive *tub-Gal80*[ts] transgene, which blocks Gal4-induced RNAi at 18 °C but not at 29 °C. In ab10 sensilla of flies grown from egg-laying to adults at 18 °C or 29 °C, the numbers of Or67a and Or85f neurons matched those of control or *lz*[RNAi] flies, respectively, observed in the previous experiment (Supplementary Fig. 9a, b). When *lz*[RNAi] was induced only in adults or only from egg-laying until 48 h APF, we observed an identical phenotype: a partial decrease in Or67a neuron numbers and a complete loss of

Or85f neuron numbers (Supplementary Fig. 9a, b). These results suggest that Lz functions in both neuronal specification and maintenance of receptor expression in Or85f neurons and similarly (albeit partially) in Or67a neurons. The incompletely penetrant phenotype in the latter neurons might reflect a partly redundant function of Lz with other TFs and/or lower efficiency of *lz* RNAi in these cells. Interestingly, Lz was one of the first TFs implicated in antennal development, with a widespread role in antennal sensillar patterning that presumably occurs at an earlier stage within the antennal imaginal disc[33]. Our findings reveal later, lineage-specific functions for Lz, illustrating how TFs can have multiple contributions within the development of this sensory system.

In ai2 sensilla of flies grown from egg-laying to adults at 18 °C, the numbers of Or23a and Or83c neurons were similar to those of control flies observed previously (Fig. 2j, k). When *lbe*[RNAi]/*lbl*[RNAi] was induced only in adults, no changes in ai2 neuron numbers were observed (Supplementary Fig. 9c, d), suggesting that these TFs are required for ai2 OSN specification but not to maintain receptor expression. However, this conclusion is tempered by our failure to recover *lbe*[RNAi]/*lbl*[RNAi] flies grown at 29 °C to examine a developmental function of Lbe/Lbl. This hypothesis remains plausible, however, given that these TFs have previously been associated with cell fate specification in muscle and heart cells[34].

### Precision and promiscuity of sensory receptor transcription

The analyses above indicate the comprehensive and functionally predictive nature of our atlas, setting the stage for exploring how precision in the olfactory system is established. We first examined the spatio-temporal expression patterns of receptor genes (Supplementary Fig. 10). In early developmental stages, only one gene was robustly detected, the *Ir25a* co-receptor (discussed further below) (Supplementary Fig. 10). By mid-stages, transcripts for over a dozen tuning Irs or Ors in different populations of neurons, as well as *Ir8a* and *Ir93a* co-receptors, were detectable (Supplementary Fig. 10). By late stages, essentially all neuronal populations were reliably expressing a specific tuning receptor gene(s) (Supplementary Fig. 10). As expected, the majority of these expressed a single tuning receptor, but our data confirmed the known cases of tuning receptor co-expression. The latter include the two subunits of the $CO_2$ receptor (*Gr21a/Gr63a*)[35,36], cases of co-expression of genomically- and phylogenetically-distant receptor genes (e.g., *Or56a/Or33a*), and several examples of co-expression of closely-related, tandemly-arranged receptor paralogs (e.g., *Or19a/Or19b*, *Or22a/Or22b*), where genes in these clusters likely retain conserved *cis*-regulatory sequences after gene duplication. We note that co-occurrence of *Ir75a*, *Ir75b* and *Ir75c* transcripts does not reflect co-expression of these genes, but rather readthrough transcription in this cluster; protein-coding transcripts are expressed in distinct populations of OSNs[18,24]. While both Gr21a and Gr63a are essential for $CO_2$ responses[35–37], it is unclear whether the other cases of co-expression are functionally significant. Co-expressed paralogs might be functionally redundant or represent a transient evolutionary state as one paralog undergoes pseudogenization or neofunctionalization[22,38].

Our analyses revealed several cases of unexpected receptor expression. For example, *Or35a* is weakly expressed in ac4 Ir84a OSNs (Fig. 3a), in addition to its well-described expression in Or35a/Ir76a ac3 neurons (Supplementary Fig. 10; *Or35a* expression in Ir84a neurons is only just visible in this globally-scaled representation). We confirmed these transcriptomic data in situ, detecting *Or35a* transcripts in Ir84a neurons in ac4 sensilla (Fig. 3b). This in situ analysis also revealed expression of *Or35a* in some Ir76a neurons (Fig. 3b), even though it was not originally observed by snRNA-seq (Supplementary Fig. 10), suggesting it was turned on in these cells at a later time point than we profiled transcriptionally. It is unlikely that Or35a is functional in ac4 neurons, as *Orco* is not expressed in these cells (Supplementary Fig. 10 and[39]). Indeed, these neurons do not respond to ligands that activate

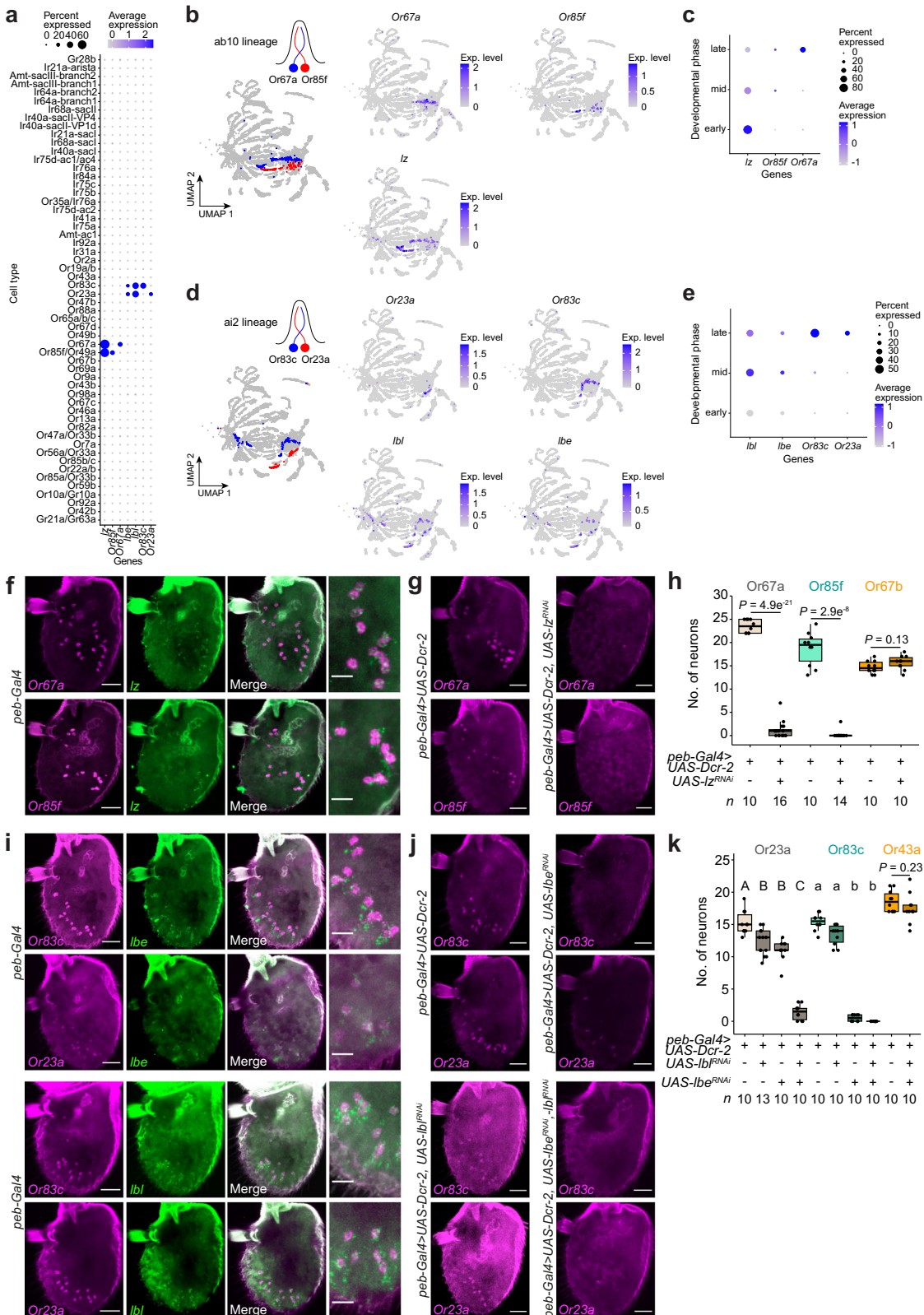

Or35a in ac3[8,40,41]. We tested whether Or35a has the potential to be functional in Ir84a OSNs through transgenic expression of Orco in these cells, but this did not produce responses to Or35a-dependent ligands (Fig. 3c), suggesting that *Or35a* expression is too low, that the transcript does not encode a functional protein (for example if aberrantly spliced[42]) and/or that other factors are required for Or35a function (Fig. 3d). *Or35a* transcripts were also detected in neurons of

sacculus chamber III (sacIII) (Supplementary Figs. 10 and 11a, b), which correspond to the Amt-expressing ammonia-sensing neurons[43]; here it is also unlikely to be functional as these cells do not express Orco (Supplementary Fig. 10).

Another example of more promiscuous expression was observed for *Ir31a*, which was detected in both Or47a and Or67d neurons, in addition to its own population (Fig. 3e, f and Supplementary Fig. 10).

**Fig. 2 | Novel lineage-specific transcription factors. a** Expression of the TFs *lozenge* (*lz*), *ladybird early* (*lbe*), *ladybird late* (*lbl*) and the indicated *Or*s across cell types (control dataset, except dying lineages, here and in other panels). In this and all other dot plots, expression levels have arbitrary units (see Methods) and are scaled independently for each gene. Receptor gene transcripts are only expressed in a fraction of the corresponding annotated cell types, which likely reflects a combination of low expression levels of these genes (particular prior to the final maturation of OSNs), as well as "dropout" expression[77]. **b** UMAPs highlighting the neurons in the ab10 lineage (left) and the expression of the indicated genes (right). **c** Expression of the indicated genes in ab10 neurons grouped by developmental phase. **d, e** As in **b, c** for the ai2 lineage. **f** RNA FISH on whole-mount antennae of control (*peb-Gal4*) animals with the indicated probes (*n* = 10-16 antennae). Scale bars, 25 μm (or 10 μm for single confocal Z-slice, high-magnification images on the right, here and in other panels). **g** RNA FISH on whole-mount antennae of control (*peb-Gal4,UAS-Dcr2*) and *lz^RNAi^* (*peb-Gal4,UAS-Dcr-2/+ ;UAS-lz^RNAi^/+;*) animals.

**h** Quantification of experiments in **g**, together with a control *Or67b* neuron population. Here and elsewhere, box plots illustrate individual data points overlaid on boxes showing the median (thick line), first and third quartiles, while whiskers indicate data distribution limits. *n* (number of antennae, here and elsewhere) is indicated underneath. *P* values are shown (two-sided *t* test). See Source Data for these and all other quantifications of histological data in this work. **i** RNA FISH on whole-mount antennae of control (*peb-Gal4*) animals (*n* = 10-16 antennae). **j** RNA FISH experiment on whole-mount antennae of control (*peb-Gal4,UAS-Dcr-2*), *lbl^RNAi^* (*peb-Gal4,UAS-Dcr-2/+ ;;UAS-lbl^RNAi^/+*), *lbe^RNAi^* (*peb-Gal4,UAS-Dcr-2/+ ;UAS-lbe^RNAi^/+*), and *lbl^RNAi^,lbe^RNAi^* (*peb-Gal4,UAS-Dcr-2/+ ;UAS-lbe^RNAi^/+;UAS-lbl^RNAi^/+*) animals. **k** Quantification of experiments in (**j**), together with a control *Or43a* neuron population. Letters indicate significant differences; Wilcoxon rank sum test followed by Bonferroni correction for multiple comparisons. *P* value is shown for Or43a neurons (two-sided *t* test).

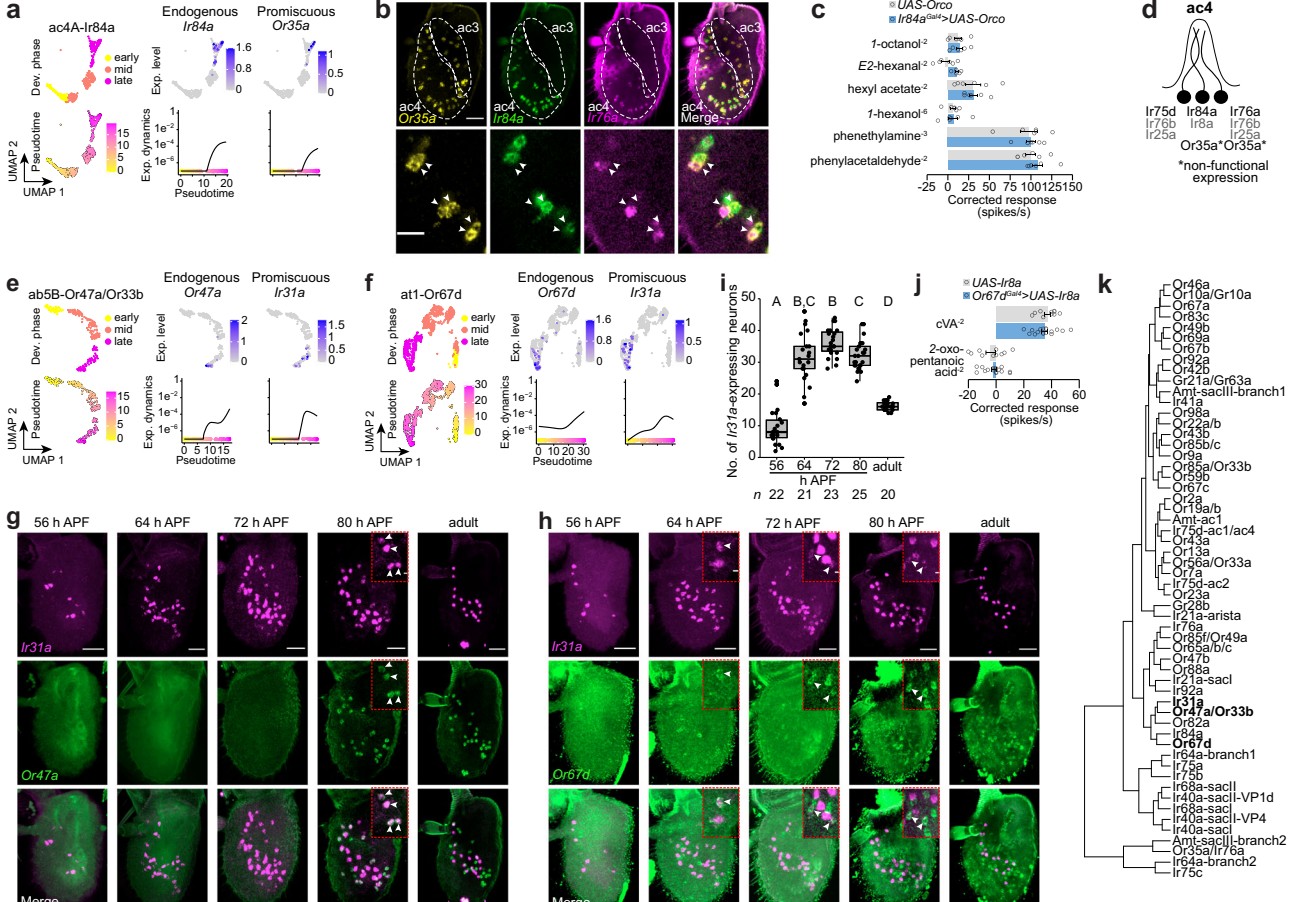

**Fig. 3 | Precision and promiscuity of sensory receptor transcription. a** Top: UMAPs of the ac4A lineage illustrating the developmental phases and receptor expression patterns (control dataset). Cells were extracted from the dataset using sensillar markers (Fig. 1f and Supplementary Fig. 6), and reclustered (see Methods). Bottom: a pseudotime UMAP and receptor expression dynamics. **b** Top: RNA FISH on whole-mount antennae of control (*peb-Gal4*) (*n* = 10–12). ac3 and ac4 sensilla zones were defined manually based upon non-pairing/pairing of *Or35a*-expressing and *Ir84a*-expressing neurons. Bottom: single confocal Z-slice, in which pairs of arrowheads highlight ac4 sensilla with two neurons expressing *Or35a*: one co-expresses *Ir84a* and one *Ir76a*. Scale bars, 25 μm (top) or 10 μm (bottom). **c** Electrophysiological responses to the indicated ligands in ac4 sensilla from antennae of control animals (*UAS-Orco, n* = 6 sensilla, here and elsewhere) and animals overexpressing *Orco* in ac4A neurons (*Ir84a^Gal4^/UAS-Orco, n* = 6). Solvent-corrected responses (mean ± SEM) are shown; see Source Data for statistical analyses. **d** Summary of tuning receptors (black) and co-receptors (gray) expressed in

ac4 neurons. **e** As in **a** for the ab5B lineage. **f** As in **a** for the at1 lineage. **g** RNA FISH on whole-mount antennae of control (*peb-Gal4*) animals at different developmental timepoints. Inset images represent 3 adjacent confocal Z-slices, and arrowheads highlight Or47a neurons co-expressing *Or47a* and *Ir31a*. Scale bars, 25 μm or 3 μm (insets). **h** RNA FISH on whole-mount antennae of control (*peb-Gal4*) animals at different developmental phases. Inset images represent 3 adjacent confocal Z-slices, and arrowheads highlight Or67d neurons co-expressing *Or67d* and *Ir31a*. Scale bars, 25 μm or 3 μm (insets). **i** Quantification of *Ir31a*-expressing neuron numbers from the experiments in **g** and **h**. Letters indicate significant differences: *P* < 0.05 in pairwise comparisons (Wilcoxon rank sum test followed by Bonferroni correction for multiple comparisons. **j** Electrophysiological responses to the indicated ligands in at1 sensilla from antennae of control animals (*UAS-Ir8a, n* = 10) and animals overexpressing *Ir8a* in at1 neurons (*Or67d-Gal4/UAS-Ir8a, n* = 17). Solvent-corrected responses (mean ± SEM) are shown. **k** Hierarchical clustering of antennal neuron types based upon differentially expressed TFs (from Supplementary Fig. 8).

Closer examination of the transcriptomic data revealed that *Ir31a* expression reaches highest levels prior to the branch termini in these lineages; this pattern contrasts with *Or47a* and *Or67d* expression, which peak at the end of the branches (Fig. 3e, f). We validated these transcriptomic data by performing RNA FISH for *Ir31a* on antennae at multiple developmental timepoints (Fig. 3g, h). Remarkably, we found that the number of *Ir31a*-expressing cells increased during pupal stages (up to ~35 cells at 72 h APF) before decreasing to the final number in adult antennae (~15 cells) (Fig. 3g–i). Concordant with the snRNA-seq data (Fig. 3e, f), several of these cells co-express *Or47a* or *Or67d* (Fig. 3g, h), indicating that the additional *Ir31a*-positive cells observed during pupal stages likely reflect ectopic expression of this *Ir* beyond its own neuron population. However, we did not robustly detect *Ir31a* in Or47a or Or67d neurons in adult antennae (Fig. 3g, h and[7]). Consistently, neither of these neuron populations responds to Ir31a-dependent ligands[44–46]. This is not due to the absence of the co-receptor Ir8a (which is never detected in these neuron populations (Supplementary Fig. 10 and[7]), as ectopic expression of the Ir8a co-receptor in Or67d neurons failed to confirm sensitivity to the best known Ir31a agonist, 2-oxopentanoic acid (Fig. 3j). We conclude that *Ir31a* is only transiently expressed in Or47a and Or67d neuron populations.

We wondered whether this transient "ectopic" expression of *Ir31a* in these Or neurons is due to similarity in the gene regulatory networks of these distinct cell types. Using information of differentially-expressed TFs (Supplementary Fig. 8) as a proxy for neurons' gene regulatory networks, we performed hierarchical clustering analysis of all OSN classes (Fig. 3k). Notably, Ir31a, Or47a, and Or67d neurons cluster closely within this tree. This observation raises the possibility that *Ir31a*'s transient expression in Or47a and Or67d neurons is due to the action of TFs that are commonly expressed in these three populations. The clustering of OSN populations based upon TF profiles might also help to explain previous observations of ectopic expression patterns of transgenic reporters for receptors. For example, the original *Or67d* reporters were expressed in both Or67d and Or82a neurons[3,47], and these neuron types are closely clustered in the tree (Fig. 3k).

We extended our survey to receptors expressed in other chemosensory organs (Supplementary Fig. 12). Amongst Ors, we detected transcripts of the maxillary palp receptor *Or42a* in Ir31a neurons (Supplementary Figs. 11c, 12). Again, this is unlikely to be functional as *Orco* is not expressed in these cells (Supplementary Fig. 10)[39], and these neurons do not respond to Or42a ligands[41]. Several *Gr*s were detected in a number of antennal cell types, consistent with observations from previous transcriptomic and transgenic studies (Supplementary Figs. 11d, e and 12)[5,48,49]. The functional significance, if any, of *Gr*s in the antenna is unclear[50]. We favor a hypothesis that such *Gr* expression reflects promiscuous transcription of these genes, possibly due to the overlap of the set of TFs present in these antennal neurons and the gustatory neurons in which these receptors are normally expressed.

Finally, our datasets confirm the previous observations of broad and partially overlapping expression of various co-receptor genes (Supplementary Fig. 10)[5,51,52]. Whether co-receptors function in every neuron in which they are expressed is unclear. Genetic and electro-physiological analyses indicate an olfactory requirement for Ir8a only with selectively-expressed acid-sensing tuning Irs (e.g., Ir31a, Ir64a), and Ir25a/Ir76b together only with amine-sensing Irs (e.g., Ir41a, Ir76a)[51,53], despite broader expression all three of these co-receptor genes (Supplementary Fig. 10)[5,51,52]. Loss of Ir25a has been described to affect Or neuron sensitivity to stimuli, with very mild increase or decrease of responses depending upon the neuron and the odor[52], but the mechanistic basis for such phenotypes is unclear. To our knowledge, there is only one case of a neuron (in ac3) that contains a functional complement of tuning and co-receptor Ors and Irs[8]. We suggest

that contributions of broadly-expressed co-receptors are constrained by the selective expression of partner tuning subunits: just as tuning receptors without co-receptors are likely to be non-functional, co-receptors without partners might have no or minimal sensory contributions.

## Heterogeneous life and death fates of sensory neuron types

We next turned our attention to the neurons that are normally removed by PCD during antennal development. While such cells can develop into functional sensory neurons when PCD is blocked[28], we know little about their molecular and developmental properties. To identify cells in the atlas corresponding to those that undergo PCD, we examined the expression of the pro-apoptotic genes *reaper* (*rpr*), *grim*, *sickle* (*skl*) and *head involution defective* (*hid*) as one or more of these genes are transcriptionally upregulated prior to PCD in many developmental contexts[54] (Fig. 4a). We detected higher expression of *rpr*, *grim* and *skl* in the PCD-blocked dataset than the control dataset, with largely overlapping expression patterns of these genes in a subset of cell clusters representing putative undead neurons (Supplementary Fig. 13a, b). By contrast, *hid* was expressed at similar levels in both PCD-blocked and control datasets (Supplementary Fig. 13a, b), indicating that this gene is not a selective marker of cells that are fated to die (as noted in other tissues[54]). We therefore combined expression enrichment of *rpr*, *grim*, and *skl* (but not *hid*) into a single "RGS score", as a quantitative measure of the likelihood that a cell type was fated to die (Fig. 4b, c).

We next decomposed the atlas of integrated control and PCD-blocked datasets (Fig. 1e) into individual sensilla, using sensillar markers (Fig. 1f and Supplementary Fig. 6) and compared cells from each dataset. We reasoned that some undead neurons might form cell clusters unique to the PCD-blocked dataset, as investigated in the next section. However, as many undead neurons express receptors characteristic of normal populations of OSNs[28], we first asked whether any such cells are embedded in clusters of these normal cells, thereby simply creating a larger cluster in the PCD-blocked dataset. Given the highly quantitative representation of cell populations in our antennal atlas (Fig. 1h), to identify such undead neurons, we first compared the relative proportion of each neuronal type in control and PCD-blocked datasets (Supplementary Fig. 13c and Fig. 4d). While neurons housed in the same sensillum generally displayed similar representations within and across datasets (e.g., Or56a/Or33a and Or7a neurons in ab4, or Or67c and Or98a neurons in ab7 (Fig. 4d)), in several cases in the control dataset we observed different proportions of co-housed neurons. For example, in ab10, Or85f neurons are underrepresented compared to Or67a neurons, and at4 Or65a/b/c neurons are less abundant than Or47b and Or88a neurons (Fig. 4d). Importantly, these mis-matched neuronal representations were at least partially re-equilibrated in the PCD-blocked atlas (Fig. 4d and Supplementary Fig. 13c), suggesting that individual cell types within a subset of sensilla are developmentally lost due to death.

Focusing first on ab10, we investigated this possibility initially by comparing the pro-apoptotic gene RGS score within individual neuron lineages (Fig. 4e). We observed that the Or85f branch has a higher score, consistent with the occurrence of PCD in a subset of these cells. In agreement with these transcriptomic data, RNA FISH for these receptor transcripts revealed a lower number of Or85f neurons compared to Or67a neurons in the antenna; while all of the former are paired with the latter, we detected several cases of isolated Or67a neurons in control animals, but not when PCD was blocked (Fig. 4f, g).

Next, we examined at4 (Fig. 4h). Here, the underrepresented Or65a/b/c neurons display a higher RGS score compared to Or47b and Or88a lineages. In situ, we detected many fewer Or65a neurons than Or47b and Or88a neurons; while the latter two cell types were always paired, only a subset formed a triplet with Or65a neurons in control animals (Fig. 4i, j). These observations match those from electron

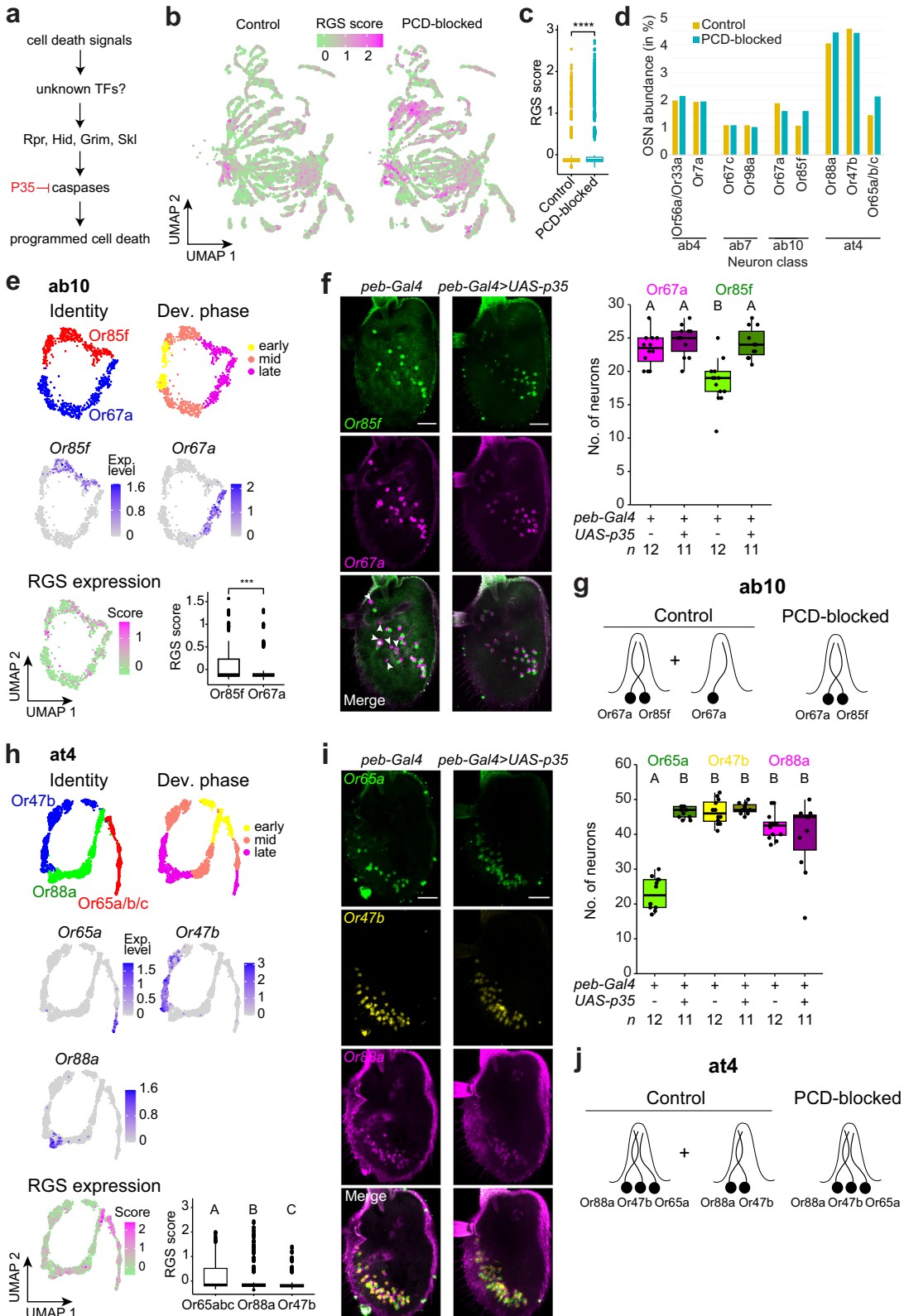

microscopic studies describing the existence of at4 sensilla housing only two neurons[55]. In PCD-blocked antennae, only the number of Or65a neurons was increased – and all were closely associated with Or47b and Or88a neurons – indicating that a subset of these neurons are also naturally removed by PCD (Fig. 4i, j). While these in situ data revealed a 1:1:1 correspondence of these neuron types, we note that within the PCD-blocked transcriptomic dataset, Or65a/b/c neurons

were still underrepresented (Fig. 4d), suggesting that this dataset lacks some undead neurons (discussed further below).

Together, these observations indicate that PCD acts within several sensilla lineages to selectively remove a subset of OSNs normally considered to represent a surviving lineage (as opposed to the lineages that are fated to die, which we consider in the next section). It is surprising that such heterogeneity has not been reported previously.

**Fig. 4 | Heterogeneous life and death fates of specific OSN populations. a** PCD pathway in *D. melanogaster*. **b** UMAPs illustrating the combined expression levels of the pro-apoptotic genes *reaper* (*rpr*), *grim* and *sickle* (*skl*) – quantified as a "RGS" score – in control (*peb-Gal4/+;;UAS-unc84:GFP/+*) and PCD-blocked (*peb-Gal4/+;UAS-p35/+;UAS-unc84:GFP/+*) datasets. **c** RGS score in control and PCD-blocked datasets. **** = *P* < 0.0001, Wilcoxon rank sum test. **d** Abundance of ab4, ab7, ab10 and at4 neuronal classes in control (yellow) and PCD-blocked (blue) datasets, calculated as the percentage of nuclei for each class relative to the total number of nuclei (from 36 h APF, excluding those forming new clusters in the PCD-blocked dataset). Data reproduced from Supplementary Fig. 13c. **e** Reconstruction of ab10 sensillum development in the integrated control and PCD-blocked datasets. We masked a cell cluster that is unique to the PCD-blocked dataset; this is considered further in Supplementary Fig. 15a–e. UMAPs indicate the identity and the developmental phases (top), expression of olfactory receptors (middle) and RGS score within the lineage (bottom). Box plot indicates the ranked (left > right) RGS score in

ab10 neurons (**** = *P* < 0.0001, Wilcoxon rank sum test). **f** RNA FISH on whole-mount antennae of control (*peb-Gal4*) and PCD-blocked (*peb-Gal4/+;UAS-p35/+*) animals. Arrowheads indicate Or67a neurons unpaired with Or85f neurons. Scale bars, 25 μm. Quantifications are shown on the right. **g** Inferred subtypes of ab10 sensilla in control and PCD-blocked antennae. **h** Reconstruction of at4 sensillum development in the integrated control and PCD-blocked datasets. UMAPs indicate the identity and the developmental phases (top), the expression of olfactory receptors (middle), and the RGS score within the lineage (bottom). The box plot indicates the ranked (left > right) RGS score in at4 neurons. **i** RNA FISH on whole-mount antennae of control (*peb-Gal4*) and PCD-blocked (*peb-Gal4/+;UAS-p35/+*) animals. Scale bars, 25 μm. Quantifications are shown on the right. **j** Inferred subtypes of at4 sensilla in control and PCD-blocked antennae. **f, h, i** Letters indicate significant differences: *P* < 0.05 in pairwise comparisons (Wilcoxon rank sum test followed by Bonferroni correction for multiple comparisons).

In part this might reflect the relatively limited data co-visualizing neurons within the same sensilla in whole-mount antennae – as opposed to cryosections (e.g. ref. 3) – which is essential to view the pairing patterns of the entire cell population. Additionally, it is possible that during electrophysiological recordings, sensilla that do not have the "expected" numbers of neurons are disregarded from detailed study. The functional significance of sensillum heterogeneity is unclear. While the lack of specific neurons in sensilla would eliminate the ephaptic inhibition that can occur between co-housed neurons[56,57], the loss of these cells might not necessarily be adaptive, as discussed below.

## Diverse states of neurons fated to die during development

We also discovered distinct types of undead neurons represented by cell branches in the developmental trajectory of several sensilla that were present only in the PCD-blocked dataset. For example, in ac3I/II sensilla – housing Or35a/Ir76a and either Ir75b (ac3I) or Ir75c (ac3II) neurons (the lineages of these two subtypes could not be fully distinguished and were considered together) – we observed two extra branches that were extinguished around 30-36 h APF in control antennae but maintained in late pupae in PCD-blocked antennae (Fig. 5a). These clusters have a high RGS score, supporting their classification as undead neurons (Fig. 5a, b). One cluster displays a signature of Naa precursor type and the other tentatively Nbb, complementing the assigned Nab and Nba identities of Ir75b/Ir75c and Or35a/Ir76a neurons, respectively[9–11] (Fig. 5c). We surveyed chemosensory receptors in these clusters, finding that the undead Naa neuron expresses *Ir75d* (Fig. 5d) – as well as *Ir* co-receptors (Supplementary Fig. 14a) – similar to the expression of *Ir75d* in Naa neurons in ac1, ac2 and ac4 sensilla in wild-type antennae[8,11]. We validated these transcriptomic data in situ by demonstrating the pairing of an *Ir75d*-expressing neuron with Ir75b/Ir75c neurons when PCD was blocked but not in control antennae (Fig. 5e–g). By contrast, the undead, putative Nbb neuron did not detectably express any tuning receptors, although we did observe the expression of multiple *Ir* co-receptors in these cells (Supplementary Fig. 14a).

Another case of undead neuron clusters was found in sacIII sensilla, which normally house Ir64a and Amt neurons. We again observed two additional cell clusters in the PCD-blocked dataset with elevated RGS scores (Fig. 5h, i). Here, one Naa cluster expressed *Ir75d*, while in the other (of unclear precursor type) we detected *Ir41a* (Fig. 5j, k). In both cases, we also detected the corresponding *Ir* co-receptors (Supplementary Fig. 14b). We confirmed in situ the presence of Ir75d and Ir41a neurons neighboring Ir64a neurons in sacIII in PCD-blocked antennae, but not in wild-type antennae (Fig. 5l). The transcriptomic data suggested that a subset of the undead Ir75d neurons might also express *Ir64a* (Fig. 5k), but we did not observe obvious co-expression in situ (Fig. 5l).

In several additional sensillar classes, we observed undead neuron populations that did not detectably express any receptors (at least up to 80 h APF). In at1, beyond the *Or67d*-expressing neuron, we identified a second, Nba-derived neuron in the PCD-blocked dataset (Fig. 5o–s), consistent with the previous electrophysiological detection of a second neuron in these sensilla[28]. In ab10 of PCD-blocked antennae – beyond the increase in Or85f neuron numbers described above – we detected an extra neuron of unclear precursor type (Supplementary Fig. 15a–e). Finally, in ab5 and sacI, we detected very small populations of likely Naa- and Nbb-derived undead neurons, respectively (Supplementary Fig. 15f–o). Tuning receptor transcripts were not detected in any of these, and only undead neurons from sacI expressed co-receptors.

Beyond all the cases where we could identify undead neurons (Supplementary Fig. 13d), for the majority of sensillar classes, we did not detect clear evidence for undead neurons (e.g., ai3 and ab4) (Supplementary Fig. 15p–q). This was surprising because, under the canonical model of the antennal SOP lineage[9–11], we expected all sensilla to have the potential to produce four terminal cells. This might reflect a technical artefact, for example, a failure to efficiently block PCD in all lineages or that nuclei of undead neurons are more fragile and lost during isolation. It is also possible that such undead cells were simply not recognized as belonging to specific sensilla. This would be the case for any Nbb-derived glia, which undergo PCD within the ab/at/ai lineages[10,13,14]. Indeed, analysis of glial cells identified those fated to survive and those that undergo PCD, although P35 did not appear to block PCD in these cells (Supplementary Fig. 13e–f). We did not identify additional populations of more mature undead neurons that are not associated with a particular sensillum or that have robust ectopic receptor expression, with one exception (asterisk in Fig. 1e). It is conceivable that the canonical OSN lineage is not universal and that some sensilla housing two neurons result from lack of a final cell division in the lineage, rather than PCD of two of the daughters of such a division.

## Mamo is required to promote PCD of Ir75d neurons

Although PCD is the most common fate of sensory neuron precursors – if we consider death as a single fate across all sensillum classes – we know essentially nothing about how it is stereotypically specified. Our data indicate that transcriptional activation of *rpr*, *grim* and/or *sickle* is likely to be the key inductive step, similar to other tissues[54]. However, very little is known about the gene regulatory network upstream of these pro-apoptotic genes in any lineage, and whether this is the same or different between distinct sensilla. As a first step, we sought TFs required to promote PCD in neurons in specific sensillar types, focusing first on the dying lineages in ac3I/II and sacIII that express *Ir75d*.

Comparison of the transcriptomes of undead Ir75d neurons in the PCD-blocked dataset with the normal Ir75d neurons housed in ac1, ac2 and ac4 revealed, as expected, that *rpr*, *grim* and *skl* are more highly

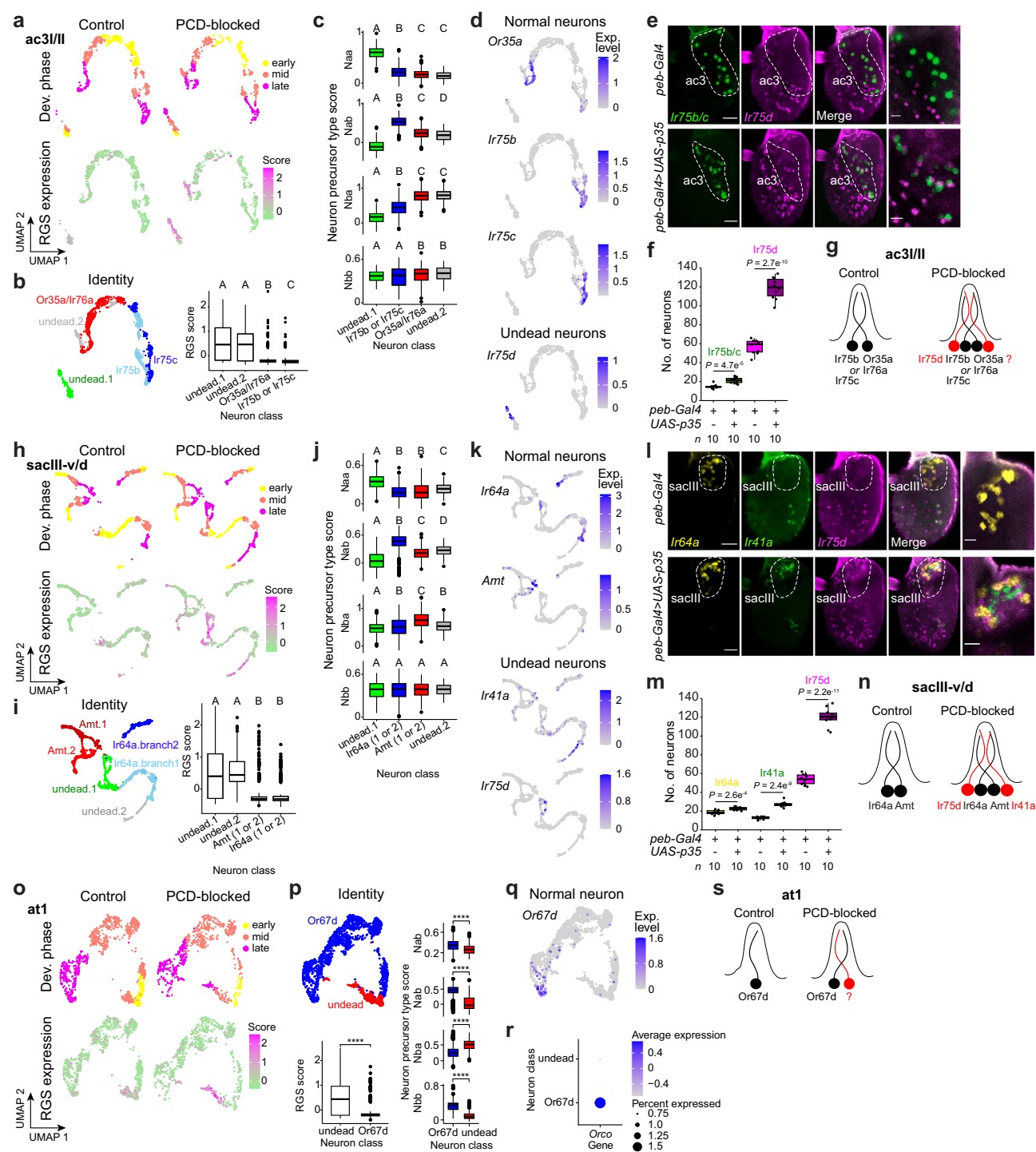

expressed in the former (Fig. 6a and Supplementary Data 2). The gene displaying the greatest enrichment in the undead Ir75d neurons was *mamo* (*maternal gene required for meiosis*) (Fig. 6a). *mamo* encodes a zinc finger C2H2 protein, which we hypothesized was a PCD-promoting TF. Consistently, in *mamo*[RNAi] antennae we observed an increase in the number of *Ir75d*-expressing neurons (Fig. 6b). These were located both in ac3I/II (paired with *Ir75b/Ir75c* expressing neurons), as well as in sacIII (paired with Ir64a neurons) (Fig. 6c–f). These phenotypes were not seen upon RNAi of several other TF genes that have enriched expression in undead Ir75d neurons (Fig. 6a and Supplementary Data 3). We note, however, that Mamo is expressed across a large number of OSN populations (including normal Ir75d neurons in ac2) (Fig. 6g), indicating that this TF is unlikely to be instructive alone

for PCD fate but rather functions in a context-dependent manner to promote death.

The *mamo*[RNAi] phenotype is similar to that of PCD-blocked antennae (Fig. 5a–n) but, in principle, loss of this TF could simply lead to ectopic *Ir75d* expression in another neuron type within ac3 and sacIII sensilla. Indeed, Mamo was previously characterized for its role in defining the fate of Kenyon cells in the mushroom body[58]. However, several observations argue against this possibility in OSNs. First, in *mamo*[RNAi] antennae the Ir75d neuron in ac3 is also paired with the Or35a/Ir76a neuron (Fig. 6h), verifying that ac3 sensilla contain distinct neurons expressing Ir75b (or Ir75c), Or35a/Ir76a and Ir75d. Second, loss of Mamo does not appear to affect fate specification of several living OSN populations in which it is expressed (Fig. 6g), including

**Fig. 5 | Identification of novel populations of undead neurons. a** UMAPs of the ac3I/II neuronal lineages in control and PCD-blocked datasets illustrating the developmental phases (top) and RGS expression (bottom), revealing cells with high RGS score that are present exclusively in PCD-blocked animals. **b** Left: lineage annotation of ac3I/II sensilla in integrated control and PCD-blocked data. Right: ranked RGS scores. **c** Neuron precursor type score for each neuron class in the ac3I/II integrated dataset. **d** UMAPs of the integrated ac3I/II dataset illustrating the expression of sensory receptors in normal and undead neurons. **e** RNA FISH on whole-mount antennae of control (*peb-Gal4*) and PCD-blocked (*peb-Gal4/+;UAS-p35/+*) animals. Right column images show a higher magnification of single confocal Z-slices in the ac3 zone. Scale bars, 25 μm (left images) or 10 μm (right images). **f** Quantifications of the experiments in **e**. *P* values are shown (two-sided *t* test). **g** Inferred types of ac3I/II sensilla in control and PCD-blocked antennae. **h** As in **a** for sacIII-v/d lineages. **i** As in **b** for sacIII-v/d lineages. **j** As in **c** for the sacIII-v/d integrated dataset. **k** As in **d** for the integrated sacIII-v/d dataset. **l** RNA FISH on whole-

mount antennae of control (*peb-Gal4*) and PCD-blocked (*peb-Gal4/+;UAS-p35/+*) animals. The sacIII zone is indicated. Images on the far right show a higher magnification of single confocal Z-slices in the sacIII zone. Scale bars, 25 μm (left images) or 10 μm (right images). **m** Quantification of OSN numbers from the experiments in **l**. *P* values are shown (two-sided *t* test). **n** Inferred states of sacIII-v/d sensilla in control and PCD-blocked antennae. **o** As in **a** for at1 lineages. **p** Top: lineage annotation of the at1 sensillum in control and PCD-blocked integrated data. Bottom: ranked RGS scores (left) and precursor type scores (right). **** indicates *P* < 0.0001, Wilcoxon rank sum test. **q** Endogenous expression of *Or67d* in the control and PCD-blocked integrated data. No receptor was robustly detected in the undead neuron population. **r** Expression of *Orco* in at1 cells in the control and PCD-blocked integrated data. **s** Inferred states of the at1 sensillum in control and PCD-blocked antennae. **b, c, i, j** Letters indicate significant differences: *P* < 0.05 in pairwise comparisons (Wilcoxon rank sum test followed by Bonferroni correction for multiple comparisons).

Or35a/Ir76a, Ir75b, Ir75c and Ir64a neurons, although we noticed modest changes in Ir75b/Ir75c and Ir64a neuron numbers upon *mamo*[RNAi] (Fig. 6b). Third, we traced the projections of *Ir75d*-expressing neurons to the antennal lobe using an *Ir75d promoter*-CD4:GFP reporter. In control animals, these neurons converge on the VL1 glomerulus (Supplementary Fig. 16). A similar convergence was observed in both PCD-blocked and *mamo*[RNAi] genotypes (Supplementary Fig. 16), consistent with both types of genetic manipulations of PCD producing equivalent undead Ir75d neurons with the same projection properties as normal Ir75d neurons.

Together, these data implicate Mamo as part of the gene-regulatory network inducing cell death of Ir75d neurons in both ac3I/II and sacIII lineages, revealing a novel function of this TF. However, there also appear to be Ir75d neurons that die in a *mamo*-independent manner, as inhibition of PCD with P35 led to more Ir75d neurons than in *mamo*[RNAi] antennae (Fig. 6i). Reviewing *Ir75d* expression patterns in PCD-blocked antennae (Fig. 5e and l), we noticed pairs of Ir75d neurons in the ac4 region. Visualizing markers for ac4 (*Ir84a* and *Ir76a*), we confirmed the existence of two Ir75d neurons in these sensilla when PCD is blocked but not in *mamo*[RNAi] antennae (Fig. 6j, k). Thus, the ac4 lineage has the potential to form a second Ir75d neuron, which is normally fated to die through other, unknown, TFs.

### Slp2 is required to promote PCD of at1 neurons

We next investigated the undead neuron population in at1. As we did not identify a sensory receptor in this neuron to permit comparison with a normal cell population (as we did for undead Ir75d neurons), we sought TFs enriched in the undead cell cluster compared with the co-housed Or67d neuron (Fig. 7a and Supplementary Data 2). The at1 undead neuron exhibited higher expression of the forkhead TF gene *sloppy paired 2* (*slp2*) (Fig. 7a, b). To examine the requirement for *slp2*, we performed electrophysiological recordings in at1 sensilla in control and *slp2*[RNAi] animals, as we could not visualize the undead neuron through RNA FISH for a receptor transcript. Control at1 sensilla house a single neuron, detected as spikes of a uniform amplitude, corresponding to the cVA-responsive Or67d neuron (Fig. 7c–e). By contrast, in a large fraction of *slp2*[RNAi] at1 sensilla, we detected an additional, smaller spike amplitude, indicative of a second, undead neuron, phenocopying the consequences of blocking PCD with P35 (Fig. 7c–e and [28]).

To further test if Slp2 was sufficient to promote PCD, we misexpressed *slp2* in developing Or67d neurons. We used the *at1-Gal4* driver, which is selectively expressed in the at1 lineage from the SOP stage until around 30 h APF, although it only covers about half of the at1 SOPs[9]. Strikingly, this manipulation led to a reduction in Or67d neuron number by approximately 50% (Fig. 7f, g). Together, these data demonstrate that Slp2 activity is necessary and sufficient to promote PCD within the at1 lineage. Similar to Mamo, Slp2 was previously characterized for its role in cell fate diversification within neuroblast

divisions, notably as part of the temporal series of TFs controlling optic lobe neuron generation[59,60]. It thus appears that the antennal SOP lineages co-opt pleiotropic neuronal TFs as part of the gene regulatory networks that promote PCD.

### Context-dependent requirement for Slp2 in promoting PCD

Beyond the undead at1 OSNs, we noticed that *slp2* is expressed in several other populations of OSNs. These include those fated to die (in ab10 and ab5 sensilla), lineages partially eliminated by PCD during antennal development (Or65a/b/c neurons in at4 and Or85f neurons in ab10) as well as in several classes of normal surviving neurons (Or19a/b, Or43a, Or69a, Or88a, Or10a/Gr10a) (Fig. 8a). The expression pattern of *slp2* therefore suggested a broader role for this TF in fate specification in the antenna. To test this hypothesis, we surveyed the consequences of *slp2*[RNAi] on other populations of OSNs. Loss of *slp2* had no effect on Or43a and Or69a expression, and led to a small decrease in Or19a neuron numbers (Fig. 8b, c). *slp2*[RNAi] did however result in increases in ab10 Or85f neurons (Fig. 8d–f) and at4 Or65a/b/c neurons (Fig. 8g–i), restoring the 1:1 relationship with other neurons in their respective sensilla. This genetic manipulation phenocopies the effect of PCD inhibition with P35 (Fig. 4f, i). At at4, we note that Or88a neurons express appreciable levels of *slp2* (Fig. 8a), but this TF did not appear to have a major role in their specification (Fig. 8h).

These observations argue that the contribution of Slp2 to PCD (or other developmental processes) is context-specific: in some cells (e.g., the dying Nba precursor in at1) it has an essential function, while in others (e.g., Or88a neurons), it has little or no role. at4 Or65a/b/c neurons and ab10 Or85f neurons represent intriguing intermediate cases, as they appear to undergo PCD heterogeneously in a *slp2*-dependent manner. We suggest that such cases of PCD represent "collateral damage" resulting from the expression of Slp2 in these cells that, perhaps due to developmental noise, reaches a minimal threshold of expression to promote PCD in some but not all neurons. In line with this notion, we noticed that different wild-type strains exhibited varying degrees of Or65a/b/c neuron loss: *w*[1118] flies have (like our *peb-Gal4* strain) low numbers of Or65a/b/c neurons compared to Or47b neurons, *Canton-S* has an equal number of these two populations, while *Oregon-R* has an intermediate number of Or65a/b/c neurons (Fig. 8j–l). Similarly, in the ab10 sensillum, Or67a neurons appear supernumerary over Or85f neurons only in *w*[1118], while the numbers of co-housed neurons in control sensilla (ab4 and ab6) are fairly balanced, despite both exhibiting inter-strain variability (Supplementary Fig. 17). These observations are consistent with the possibility that this neuron loss trait is not a fixed, adaptive phenotype of *D. melanogaster*.

### Discussion

The structural and functional properties of neural circuits are often considered as optimized to fulfill their role in controlling animal

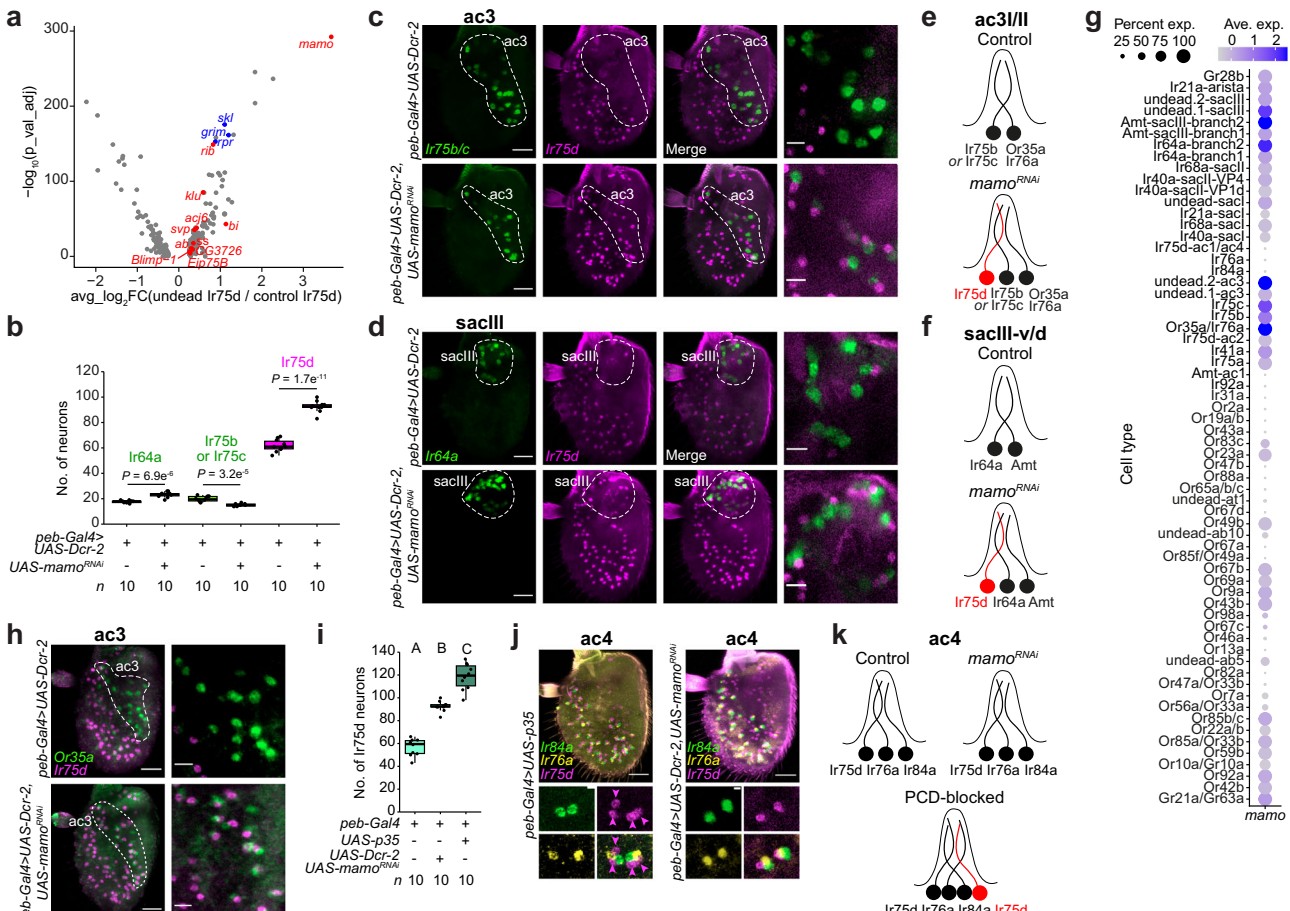

**Fig. 6 | Mamo specifies PCD in Ir75d neuron lineages. a** Volcano plot illustrating differentially-expressed genes between undead Ir75d OSNs (in ac3 and sacIII) and normal Ir75d OSNs (in ac1, ac2 and ac4) (pseudobulk analysis, x axis = log₂FC("undead"/control), y axis = -log₁₀(adjusted P) (two-sided Wilcoxon rank sum test followed by Bonferroni correction using all detected genes), log₂FC > (or <) 0.25, % of positive nuclei >0.25). Pro-apoptotic genes are highlighted in blue, and TFs enriched in undead Ir75d neurons are highlighted in red. See Supplementary Data 2. **b** Quantification of OSN numbers in the indicated genotypes. P values are shown (two-sided t test). **c, d** RNA FISH on whole-mount antennae of control (peb-Gal4,UAS-Dcr-2) and mamo^RNAi (peb-Gal4,UAS-Dcr-2/+;UAS-mamo^RNAi/+) animals. ac3 (**c**) and sacIII (**d**) zones are shown. Right column images show a higher magnification within the ac3 and sacIII zones of a single confocal Z-slice, showing undead Ir75d neurons in mamo^RNAi ac3 and sacIII sensilla. Scale bars, 25 μm (left images) or 10 μm (right images). **e, f** Inferred states of ac3I/II (**e**) and sacIII-v/d (**f**) sensilla in control and mamo^RNAi antennae. **g** Dot plot showing the expression of mamo in all

annotated neuron lineages (control and PCD-blocked integrated datasets). **h** RNA FISH on whole-mount antennae of control (peb-Gal4,UAS-Dcr-2) and mamo^RNAi (peb-Gal4,UAS-Dcr-2/+;UAS-mamo^RNAi/+) animals. The ac3 zone is indicated. Right column images show a higher magnification in the ac3 zone in a single confocal Z-slice. Scale bars, 25 μm, or 10 μm for high-magnification images. **i** Quantification of Ir75d neurons in the indicated genotypes; data is replotted from Figs. 5f, 6b, to highlight the mismatch of Ir75d neuron numbers in mamo^RNAi and PCD-blocked animals. Letters indicate significant differences: P < 0.05 in pairwise comparisons (Wilcoxon rank sum test followed by Bonferroni correction for multiple comparisons). **j** RNA FISH on whole-mount antennae of PCD-blocked (peb-Gal4/+;UAS-p35/+) and mamo^RNAi (peb-Gal4,UAS-Dcr-2/+;UAS-mamo^RNAi/+) animals, showing an additional Ir75d expressing neuron in ac4 sensilla of PCD-blocked animals. Bottom images show higher magnification of a single Z-slice. Scale bars, 25 μm (top images), or 3 μm (bottom images). **k** Inferred states of ac4 sensilla in control, mamo^RNAi and PCD-blocked antennae.

behavior. In reality, however, these properties represent just a snapshot in evolutionary time, neither precisely the same as in the past, nor necessarily maintained in the future. Understanding the nature of this snapshot in the context of a continuous process of change can offer insights into how nervous systems evolve. The insect olfactory system is a particularly attractive model for studying this phenomenon: this sensory system can be subject to strong environmental selection pressures as the bouquet of external volatile cues changes, and the typically large, rapidly reproducing populations of insect species provide the necessary genetic substrate for evolutionary modifications. Using *D. melanogaster* as a model, we have characterized developmental properties of OSN lineages to reveal features of this species' olfactory system underlying its functional stereotypy, as well as how these offer the potential for evolution.

Through high-resolution spatio-temporal transcriptomic profiling of developing neurons, we first confirmed the expected global

precision of receptor transcription in the olfactory system, typically a single tuning receptor per neuron type[3–6,8]. However, we also reveal that this control is imperfect. We describe several examples of co-expression of tuning receptors of the same or different families, both known[4,35,47] and new. Such co-expression – sometimes transient – presumably reflects similarities in the gene regulatory networks controlling receptor expression in distinct neuronal classes, as suggested by the high degree of overlap in the set of TFs we found in different cell types. The promiscuity in receptor expression is constrained, nevertheless, by the necessity to have the correct complement of tuning receptor and co-receptor subunits to form a functional complex[51,53,61]. This seemingly ectopic expression of receptor genes might permit a degree of evolvability as it is possible that only transcriptional activation of a complementary (co-)receptor subunit is necessary to reconstitute a functional sensory receptor. However, as we have shown in two cases that artificial co-receptor expression is insufficient to

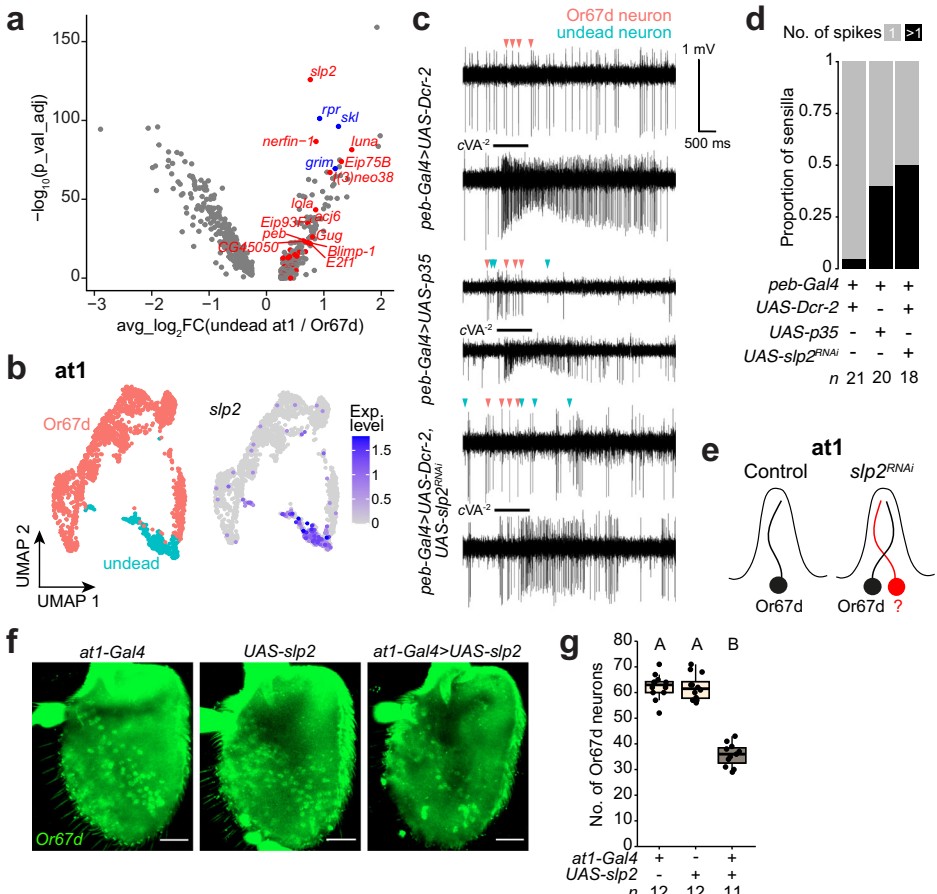

**Fig. 7 | Slp2 activity is necessary and sufficient to specify PCD in the at1 lineage.**
**a** Volcano plot illustrating differentially-expressed genes between at1 undead and Or67d OSN populations (pseudobulk analysis, x axis = log₂FC("undead"/control), y axis = -log₁₀(adjusted P) (two-sided Wilcoxon rank sum test followed by Bonferroni correction using all detected genes), log₂FC > (or <) 0.25, % of positive nuclei > 0.25). Pro-apoptotic genes are highlighted in blue, and TFs whose expression is enriched in the undead at1 neurons are highlighted in red. See Supplementary Data 2 for the top 20 genes enriched in undead and Or67d at1 sensilla neurons.
**b** UMAPs of the at1 lineage (integrated control and PCD-blocked datasets), illustrating neuron identity and *slp2* expression. **c** Traces of spontaneous (top) and cVA-evoked (bottom) electrophysiological activity from at1 sensilla in the antennae of control (*peb-Gal4,UAS-Dcr-2*), PCD-blocked (*peb-Gal4/+;UAS-p35/+*) and *slp2^RNAi^*

(*peb-Gal4,UAS-Dcr-2/+;UAS-slp2^RNAi^/+*) animals. Red and blue arrows indicate spikes of the Or67d and undead neurons, respectively. **d** Quantification of the proportion of at1 sensilla housing 1 or >1 spike amplitude (as assessed from traces of spontaneous activity) in the indicated genotypes. **e** Inferred states of the at1 sensillum in control and *slp2^RNAi^* antennae. **f** RNA FISH experiment on whole-mount antennae of control animals (*UAS-slp2-ORF-3HA/+* and *at1-Gal4/+*) and in animals over-expressing *slp2* specifically and transiently in the at1 lineage during development[9] (*UAS-slp2-ORF-3HA/at1-Gal4*) with probes targeting *Or67d*. Scale bar, 25 μm.
**g** Quantification of Or67d neurons from **f**. Letters indicate significant differences: *P* < 0.05 in pairwise comparisons (Wilcoxon rank sum test followed by Bonferroni correction for multiple comparisons).

reconstitute the function of "ectopic" tuning receptor activity, we suspect that levels of ectopic tuning receptor might also need to be enhanced. There are also other possible requirements for functionality, such as perireceptor proteins or morphological specializations of the neuron/sensillum.

Our atlas of PCD-blocked antennae allowed us to characterize the development of many lineages fated to die. These lineages exhibit diverse properties: the undead neurons we identified were derived from different precursor types within diverse sensillar classes. Some of these robustly express a tuning receptor gene; here PCD can counteract promiscuous receptor expression, serving as a further regulatory layer to ensure precision in receptor patterning in the mature sensory system. Other undead neurons lack a detectable tuning receptor; however, our detection of co-receptor transcripts even in these "empty" neurons (e.g., in ac3 and sacI sensilla) supports the idea that broad co-receptor expression extends to cells destined to die, thereby reflecting a more amenable evolutionary substrate for subsequent re-emergence of new neuron types from such dying lineages. Why some neurons express a receptor gene and others do not is an interesting question. One possibility is that this property reflects their

evolutionary age: lineages that more recently evolved a PCD fate might retain the gene regulatory network to permit receptor expression, while evolutionarily older dying lineages might have drifted in fate thus losing the capacity to express specific tuning receptors. Whatever the reason, because OSN survival does not depend upon functional receptors[61,62], the absence of "empty" neurons in the extant olfactory system[61,62] implies that during the emergence of a new sensory pathway from a dying lineage, changes in the gene regulatory network to turn off PCD and turn on a tuning receptor must be closely coordinated.

In this context, our identification of the first TFs (Mamo and Slp2) required for specification of PCD in OSN lineages provides an important entry-point into understanding how life/death fate decisions occur and evolve. The broader expression of both of these TFs beyond lineages fated to die emphasizes their context-dependent function, presumably because they are embedded within gene regulatory networks that influence survival, death, and/or differentiation of sensory lineages. Further characterization of these, and other, TFs in dying lineages will help reveal whether and how they directly control pro-apoptotic gene expression, and why they promote PCD in some

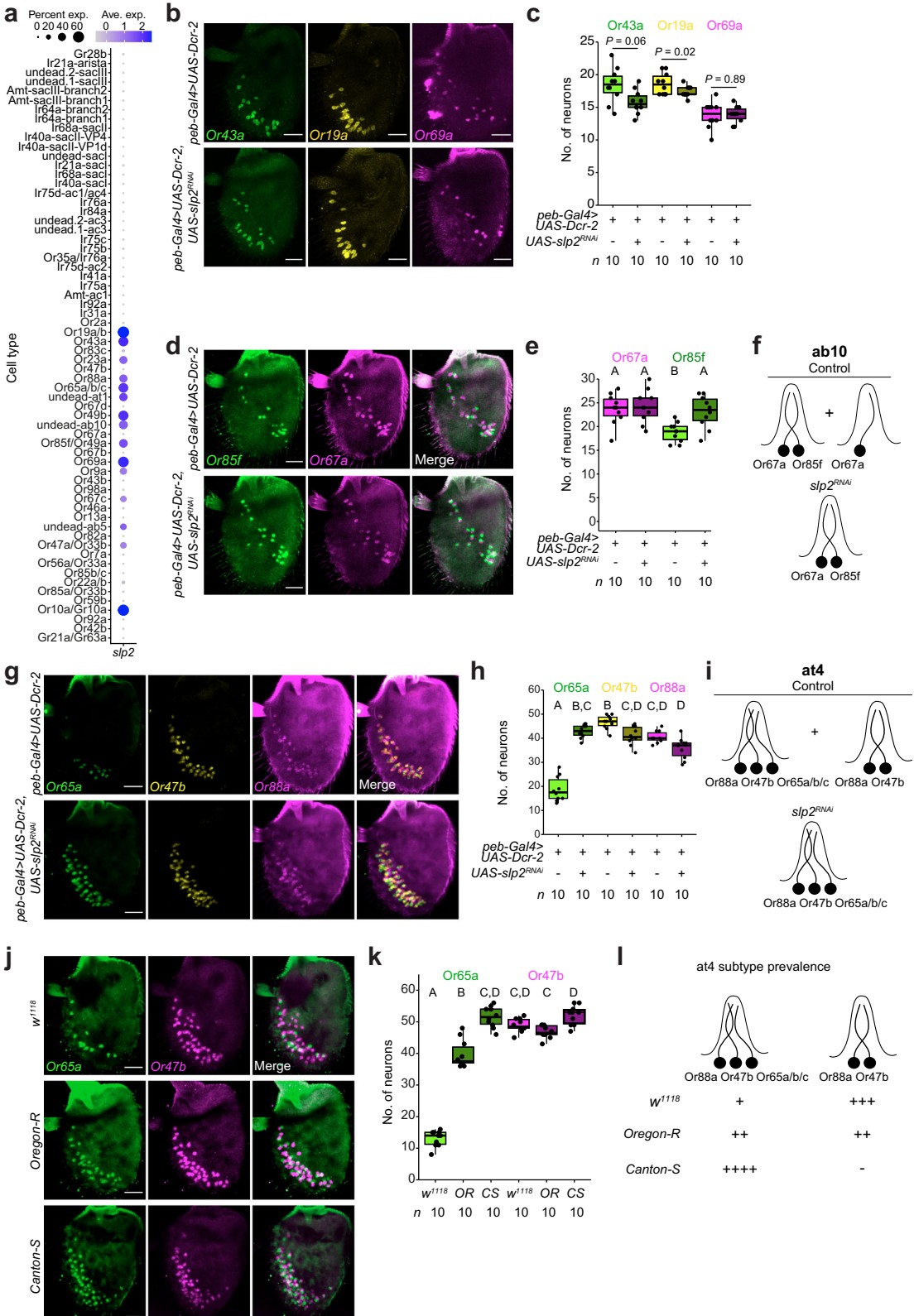

lineages but not others. Such knowledge will be key to understanding how patterning of PCD can change during evolution to generate or remove individual sensory neuronal populations, and how this is coordinated with the selective expression of receptors.

While several lineages are entirely condemned to death (e.g., ac3 Ir75d-expressing neurons), we found, unexpectedly, that some dying neurons represent subsets of normal surviving lineages (e.g., Or65a/

b/c neurons in at4). Such heterogeneity can be interpreted in different ways. The phenomenon might be an adaptive trait, for example, to limit the number of just one class of neurons within a specific sensillum type. However, the variation in Or65a/b/c population size across different *D. melanogaster* genotypes argues against this possibility, although we cannot exclude that such intraspecific phenotypic diversity arises from local/lab adaptation of specific

**Fig. 8 | Context-specific requirement for Slp2 in PCD. a** Dot plot showing the expression of *slp2* in all annotated neuron lineages (control and PCD-blocked integrated datasets). **b** RNA FISH on whole-mount antennae of control (*peb-Gal4,UAS-Dcr-2*) and *slp2^RNAi^* (*peb-Gal4,UAS-Dcr-2/ + ;;UAS-slp2^RNAi^/+*) animals. Scale bar, 25 µm. **c** Quantification of neurons from **b**. *P* values are shown (two-sided *t* test). **d** RNA FISH on whole-mount antennae of control (*peb-Gal4,UAS-Dcr-2*) and *slp2^RNAi^* (*peb-Gal4,UAS-Dcr-2/ + ;;UAS-slp2^RNAi^/+*) animals. Scale bar, 25 µm. **e** Quantification of neurons from **d**. **f** Inferred states of the ab10 sensillum in control and *slp2^RNAi^* antennae. **g** RNA FISH on whole-mount antennae of control (*peb-Gal4,UAS-Dcr-2*)

and *slp2^RNAi^* animals (*peb-Gal4,UAS-Dcr-2/ + ;;UAS-slp2^RNAi^/+*). Scale bar, 25 µm. **h** Quantification of neurons from **g**. **i** Inferred states of the at4 sensillum in control and *slp2^RNAi^* antennae. **j** RNA FISH on whole-mount antennae of *w^1118^*, *Oregon-R* and *Canton S* animals. Scale bar, 25 µm. **k** Quantification of neurons from **j**. **l** Inferred states of the at4 sensillum in the antennae of widely-used *Drosophila melanogaster* strains. The number of "+" signs reflects the relative prevalence of at4 subtypes (- = absent). **e, h, k** Letters indicate significant differences: *P* < 0.05 in pairwise comparisons (Wilcoxon rank sum test followed by Bonferroni correction for multiple comparisons).

strains. Alternatively, heterogeneous PCD might reflect promiscuity in transcriptional specification of PCD resulting from overlap in gene regulatory networks of surviving and dying lineages, akin to the promiscuity observed in receptor expression. In this context, the observed heterogeneity in PCD specification might reflect a transitory evolutionary state of these pathways, such as the initial stages of loss of a sensory population. In extreme cases, promiscuous PCD might result in sensilla devoid of OSNs, as observed in rare instances[55].

Taken together, our work both provides a new understanding of the multilevel mechanisms that define the functional precision of the *Drosophila* olfactory system and highlights previously overlooked variability at each of these levels that might provide a substrate for the molecular and cellular changes in these sensory pathways over evolutionary timescales. This study sets the stage for comparison of the antennal neuronal populations of phylogenetically diverse Drosophilias and other insects to trace and understand the diversification in their olfactory pathways.

## Methods

### *Drosophila* culture and transgenic line generation
Flies were reared in vials containing standard wheat flour/yeast/fruit juice medium maintained in incubators with 12 h light:12 h dark cycles at 25 °C. Published strains are listed in Supplementary Table 1.

The *Ir75d promoter-CD4:tdGFP* construct was generated by amplifying a 1994 bp DNA fragment from genomic DNA of the reference *D. melanogaster* strain (RRID: BDSC_2057) using the following forward and reverse PCR-primers GGGGACAAGTTTGTACAAAAA AGCAGGCTTCAgcaatggtaatattaaacta and GGGGACCACTTTGTACAA GAAAGCTGGGTCatccggcaactgattgcccca; this region encompasses 1850 bp 5′ regulatory sequence and 144 bp of exon 1, as in a previous promoter construct[41]. The amplified sequence was inserted into *pDESTHemmarG* (Addgene #31221) via Gateway recombination and confirmed by sequencing. The construct was integrated into attP40 (chromosome II) using phiC31-mediated transgenesis by BestGene Inc.

### Antennal dissection and nuclear isolation
10-15 virgin *peb-Gal4* females were placed in vials with 5-10 *UAS-unc84:GFP* or *UAS-p35;UAS-unc84:GFP* males for 5 days, after which adults were removed. White pupae (corresponding to 0 h after puparium formation (APF)) were carefully transferred to fresh vials and aged for 18, 24, 30, 36, 42, 48, 56, 64, 72, or 80 additional hours. Developing antennae from aged pupae were dissected in ice-cold Schneider's medium (Gibco, 21720024) and immediately transferred to 1.8 ml Eppendorf tubes containing 100 µl Schneider's medium, flash-frozen in liquid nitrogen, and stored at −70 °C. The numbers of antennae dissected in control (*peb-Gal4/+;UAS-unc84:GFP/+*) and PCD-blocked (*peb-Gal4/+;UAS-p35/+;UAS-unc84:GFP/+*) genotypes are as follows (time-point in h APF (*n* antennae control / *n* antennae PCD-blocked): 18 (49 / 57), 24 (72 / 52), 30 (67 / 56), 36 (52 / 51), 42 (45 / 48), 48 (45 / 55), 56 (59 / 49), 64 (48 / 60), 72 (54 / 62) and 80 (47 / 53).

Samples were thawed on dry ice and nuclear suspensions prepared as described[5,7]. Suspensions from "early" (18/24/30 h APF), "mid" (36/42/48 h APF), and "late" (56/64/72/80 h APF) developmental time points were pooled together, with the exception of time point 80 h

APF in the control genotype, which was pooled with mid time points. This latter pooling reflected the initial experimental design, but 80 h APF control cells could be effectively re-classified in the late time point for all subsequent analyses (see below the "Lineage developmental and pseudotime reconstruction" section). After the addition of Hoechst 33342 (Thermo Fisher Scientific, 62249), samples were loaded into a FACS Aria flow cytometer.

### Single-nuclear RNA-sequencing
For each pooled sample, 2 × 20,000 GFP-positive nuclei (or 1 × 20,000 for early nuclei in the PCD-blocked condition) were sorted and immediately loaded onto the Chromium Next GEM Chip (10x Genomics). Sequencing libraries were prepared with the Chromium Single Cell 3′ reagent kit v3.1 dual index, following the manufacturer's recommendations. Libraries were quantified by a fluorometric method, and quality was assessed on a Fragment Analyzer (Agilent Technologies). Sequencing was performed on an Illumina NovaSeq 6000 v1.5 flow cell for 100 cycles according to 10x Genomics' recommendations (28 cycles read1, 10 cycles i7 index read, 10 cycles i5 index, and 90 cycles read2). Demultiplexing was performed with bcl2fastq2 Conversion Software (v2.20, Illumina). Raw snRNA-seq data was first processed through Cell Ranger (v6.1.2, 10x Genomics) with default parameters except -include introns- that was set to TRUE. A custom *D. melanogaster* reference genome and transcriptome from FlyBase (Drosophila_melanogaster.BDGP6.28.101) were used for mapping. Two marker genes, *GFP* and *p35*, were added to the GTF and FASTA files prior to building the custom genome reference with cellranger mkref (v6.1.2) function, following the 10x Genomics protocol.

### Integration of developing antennal snRNA-seq datasets
Ambient RNA contamination removal was applied on each of the cellranger output matrices from early, mid, and late pools in control and PCD-blocked genotypes using SoupX[63] (v1.6.1, default parameters) and then subsequently integrated and analyzed using Seurat (v4.3.0.1) in RStudio. Matrices were normalized using SCTransform normalization[64] (default parameters) and integrated using reciprocal PCA workflow (SelectIntegrationFeatures (nfeatures = 3000), PrepSCTIntegration, FindIntegrationAnchors (reference=control), and IntegrateData) described in (https://satijalab.org/seurat/articles/integration_rpca.html). PCA was used for clustering of the integrated datasets as follows: RunPCA (npcs=50), RunUMAP (reduction = "pca", dims:1:50), FindNeighbors(reduction = "pca", dims:1:50), FindClusters (resolution=0.5), resulting in 49 clusters. Gene expression levels shown in the dot plots, box plots, violin plots, and UMAPs are residuals from a regularized negative binomial regression, and have arbitrary units.

### Cell type annotation
Marker genes (cutoff used: log₂FC > 3) of various cell types composing the adult antenna (sensory neurons, epithelial cells, hemocytes, muscle cells, glial cells, and Johnston's organ cells) were extracted from the Fly Cell Atlas dataset[7] via the SCope interface[65]. For support cells, *cut* and *shaven* were used as marker genes. For each cluster, the expression level (score) of cell type marker genes was computed using the AddModuleScore function implemented in Seurat. This function

calculates the average expression levels of each program (gene sets) on a single cell level, subtracted by the aggregated expression of control gene sets. All analyzed genes are binned based on averaged expression, and the control genes are randomly selected from each bin. Cells were annotated through manual inspection of cell type scores in each cluster.

## Neuron class annotation

Third antennal segment sensory neurons from the control and PCD-blocked integrated datasets were subclustered at high resolution as follows: RunPCA (npcs=45), RunUMAP (reduction = "pca", dims=1:45), FindNeighbors(reduction = "pca", dims=1:45, k.param=10), FindClusters (resolution=6), resulting in 162 clusters. The expression of diagnostic tuning receptor and co-receptor genes in the control dataset was used for initial cluster annotation. Clusters expressing more than one diagnostic receptor were further subclustered and annotated following the same pipeline. Because chemosensory receptor gene expression occurs relatively late during antennal development, these genes could not be used alone to discriminate neuron classes at earlier developmental stages. We therefore iteratively extracted marker genes of each neuron class as follows: FindAllMarkers (assay = "SCT", logfc.threshold = 0.25, min.pct = 0.25, only.pos = T, test.use = "wilcox", p_val_adj <0.05) and evaluated the expression level of these markers in the unannotated clusters using the AddModuleScore function. Unannotated clusters showing the highest scores for a given lineage were assigned to that lineage, which were then incorporated into the next iteration of marker extraction and scoring. In cases of conflicting scoring of various marker lists, we inspected the expression of a few top marker genes and privileged the shortest and continuous differentiation trajectories of lineages. After three iterations (0,1,2) of cluster annotation based on individual neuron classes, further annotation of the remaining unannotated clusters was based on sensillum type by grouping co-housed OSN lineages to extract sensillar markers as follows: FindAllMarkers (assay = "SCT", logfc.threshold = 0.25, min.pct = 0.25, only.pos = T, test.use = "wilcox", p_val_adj <0.05) followed by AddModuleScore of the various marker genes. Overall, this iterative process allowed us to annotate 90% of the neurons; the remaining unannotated fraction mostly correspond to early phase cells. Marker genes distinguishing coeloconic and sacculus neuron populations were extracted from[18,66].

## Precursor type annotation

We used published data[9,11] to assign most normal (surviving) OSNs from the control dataset to a precursor type (Naa, Nab, Nba, or Nbb). Marker genes for each precursor type were extracted as follows: FindAllMarkers (assay = "SCT", logfc.threshold = 0.25, min.pct = 0.4, only.pos = T, test.use = "wilcox", p_val_adj <0.05). In a strategy similar to the OSN type annotation, the expression scores (usingthe AddModule score function implemented in Seurat) of precursor type marker genes were iteratively used to assign unmapped OSN classes to a precursor type (with the exception of aristal neurons that could not be unambiguously assigned to any precursor type). Precursor type marker genes are listed in Supplementary Data 1.

## Lineage developmental and pseudotime reconstruction

Individual sensillar lineages were extracted from the integrated datasets and subclustered using a similar workflow as described above, but with lineage-specific parameters: RunPCA, RunUMAP, FindNeighbors, FindClusters. Each cluster was assigned to early, mid, and late developmental phases based on PCD-blocked data and then extrapolated to the control data (this was necessary due to the pooling of the 80h APF time point with the mid stages in the latter, as mentioned above). Individual lineages were imported into monocle3 (v1.2.9)[67–70] and pseudotime inferred as follows: cluster_cells, learn_graph (use_partition = F), order_cells (start_end was chosen at the tip of "early" stage

cells), and gene expression dynamics (as shown in $\log_{10}$(residual fit)) plotted using plot_genes_in_pseudotime (color_cells_by = "pseudotime") function.

## Identification of undead neurons

During reconstruction of individual lineages, we systematically checked for the presence of clusters that were exclusive to the PCD-blocked dataset using the DimPlot (split.by = "condition") function in Seurat. We confirmed that any such PCD-blocked specific clusters corresponded to undead neuron lineages by quantifying the expression of the pro-apoptotic genes *rpr*, *grim* and *skl* (RGS score) using the AddModuleScore function. If there was no difference in clusters between control and PCD-blocked animals, we quantified the number of cells (from 36 h APF) of each lineage in both conditions. In a few cases (e.g., at4 Or65a/b/c neurons), there was a clear increase in the number in the PCD-blocked dataset. We further validated such additional cells as being undead neurons by quantifying and ranking RGS score at the sensillar level using the AddModuleScore, ViolinPlot (features = "RGS", sort = T) function. Typically, lineages with "embedded" undead neurons had the highest RGS score.

## Differential gene expression analysis

Differentially expressed genes (DEGs) were extracted from various Seurat objects using the FindAllMarkers function with the following parameters applied for various comparisons: for developmental time marker genes (Supplementary Fig. 2): (object = "All OSNs", assay = "SCT", logfc.threshold = 0.25, min.pct = 0.25, only.pos = T, test.use = "wilcox", p_val_adj <0.05), for OSN marker genes (Supplementary Fig. 5): (object = "All annotated OSNs in control condition", assay = "SCT", logfc.threshold = 0.25, min.pct = 0.18, only.pos = T, test.use = "wilcox", p_val_adj <0.05), for sensillar marker genes (Supplementary Fig. 6): (object = "All annotated OSNs in control condition", assay = "SCT", logfc.threshold = 0.25, min.pct = 0.25, only.pos = T, test.use = "wilcox", p_val_adj <0.05), for OSN precursor type marker genes (Supplementary Fig. 7): (object = "All annotated OSNs in control condition except arista", assay = "SCT", logfc.threshold = 0.25, min.pct = 0.4, only.pos = T, test.use = "wilcox", p_val_adj <0.05). For each comparison, an analysis of Gene Ontology terms over-represented in marker genes was performed using clusterProfiler (v4.4.4)[71].

## Hierarchical clustering of sensory neurons based on differentially expressed transcription factors

A phylogenetic tree relating sensory neuron classes (Fig. 3k) was built based on a distance matrix constructed in the differentially expressed transcription factors (DE_TFs) space (Supplementary Fig. 8) using the BuildClusterTree function implemented in Seurat: BuildClusterTree (assay = "SCT", features = DE_TFs).

## Hybridization chain reaction RNA fluorescence in situ hybridization

RNA probes (Supplementary Table 2) were synthesized by Molecular Instruments, and we followed a published HCR RNA FISH protocol[72] with minor modifications. Female flies (2-7 days old) of interest were flash-frozen in liquid nitrogen, and antennae were passed through a mini-sieve (mesh-width = 80 μm) to remove other body parts and collected in Petri dishes containing fixation solution (1×PBS, 3% Triton X-100, 4% paraformaldehyde). For each staining, 30-40 antennae were immediately transferred into 1.5 ml Eppendorf tubes filled with 1 ml of fixation solution and placed on a platform for 1 h at room temperature (RT) with gentle shaking (35 rpm). Fixative solution was removed and samples washed 2× 5 min with 1 ml of PBS 1×, 3% Triton X-100 at RT, followed by 3× 5 min with 1 ml of 1×PBS, 0.1% Triton X-100 at RT. Wash solution was removed and samples pre-hybridized with 300 μl of probe hybridization buffer (Molecular Instruments) for 30 min at 37 °C. Pre-hybridization buffer was removed and replaced by probe

solution (5 µl of probe in hybridization solution, 500 µl final volume, except for *Ir75d*, *Ir41a*, *Or85f*, *Or88a*, *Or67d*, and *Ir76a* probes for which 10 µl were used) for 16 h at 37 °C. Probe solution was removed and samples washed 4× 15 min with 500 µl of pre-heated (37 °C) probe wash buffer (Molecular Instruments) at 37 °C, followed by 3× 5 min washes at RT with 500 µl of 5×SSCT solution (5×SSC, 0.1% Triton X-100). Samples were pre-amplified with 300 µl of amplification buffer (Molecular Instruments) for 30 min at RT. During this time, for each probe in a sample, 10 µl of hairpin amplifiers h1 and 10 µl of hairpin amplifiers h2 (or 15 µl each when probe volume was doubled) were heated at 95 °C for 90 s and snap-cooled to RT, with protection from light. Pre-amplification buffer was removed from the samples and replaced with the snap-cooled amplifiers in amplification buffer (Molecular Instruments), 500 µl final volume, and placed at RT (protected from light) for 16 h. Amplification solution was removed and samples washed 5× in 500 µl of 5×SSCT at RT (2× 5 min, 2× 30 min, and 1× 5 min). 5×SSCT was removed and 50 µl of Vectashield mounting medium (Vector Laboratories, H-1200) was added before mounting.

### Standard RNA fluorescence in situ hybridization and immunohistochemistry

For the experiments in Fig. 6h, we used a standard RNA FISH protocol[73] on female flies (2-7 day old), using RNA FISH probes generated with primers indicated in Supplementary Table 2. When combined with immunohistochemistry (IHC) (Supplementary Fig. 16a), we performed standard RNA FISH until TSA-Cy5 colorimetric detection, when samples were washed 5× 20 min at RT with 500 µl of TNT buffer and incubated for 1 h in 300 µl of blocking solution (1×PBS, 0.2% Triton X-100, 5% heat-inactivated goat serum). Supernatant was replaced by 500 µl of a chicken anti-GFP antibody containing blocking solution, and samples were placed at 4°C on a rotating wheel for 40 h. Samples were then washed 5× 20 min at RT with 500 µl of 1×PBS, 0.2% Triton X-100, and incubated for 1 h in 300 µl of blocking solution. Supernatant was replaced by 500 µl of secondary antibodies in blocking solution, and samples were placed at 4 °C on a rotating wheel for 40 h. Samples were washed 5× 20 min at RT with 500 µl of 1×PBS, 0.2% Triton X-100 and 50 µl of Vectashield mounting medium was added to each sample prior to mounting. Primary and secondary antibodies are listed in Supplementary Table 3.

### Immunohistochemistry in the central brain

Female flies (2–7 days old) were fixed in 2 ml of 1×PB (32 mM $NaH_2PO_4$, 68 mM $Na_2HPO_4$), 3% Triton X-100, 4% paraformaldehyde for 2 h at RT. Brains were dissected into ice-cold 1×PB under a binocular microscope and immediately transferred into 1.5 ml Eppendorf tubes containing 1 ml of 1×PB, 0.3% Triton X-100. Brains were washed 5× 15 min in 1 ml of 1×PB, 0.3% Triton X-100 at RT, and then incubated in 1 ml of blocking solution (1×PB, 0.3% Triton X-100, 5% heat-inactivated goat serum) for 1 h at RT. The supernatant was replaced by 500 µl of primary antibodies in blocking solution and the samples placed on a rotating wheel for 40 h at 4 °C. Brains were washed 5× 15 min in 1 ml of 1×PB, 0.3% Triton X-100 at RT, and then incubated in 1 ml of blocking solution for 1 h at RT. The supernatant was replaced by 500 µl of secondary antibodies in blocking solution and the samples placed on a rotating wheel for 40 h at 4 °C. Brains were washed 5× 15 min in 1 ml of 1×PB, 0.3% Triton X-100 at RT, and 50 µl of Vectashield mounting medium was added prior to mounting.

### Image acquisition and processing

Images from antennae and antennal lobes were acquired with confocal microscopes (Zeiss LSM710 or Zeiss LSM880 systems) using a 40× (antennae) or a 63× (antennal lobe) oil immersion objective. Images were processed using Fiji software[74].

### Electrophysiology

Single sensillum recordings on ac4 coeloconic sensilla were performed on female flies (2-4 days old) using glass electrodes filled with sensillum recording solution, essentially as described[43]. Coeloconic sensilla were identified based on their stereotyped locations on the antenna and responses to diagnostic odorants. The odor response was calculated from the difference in summed spike frequencies of all OSNs in response to a 0.5 s odor puff compared to a 0.5 s solvent puff, as described[43].

Single sensillum recordings from at1 were performed on female flies (2-7 days old) using tungsten electrodes, essentially as described[75]; sensilla were identified by their morphology, characteristic location and responses of Or67 d neuron to cVA. Odor responses were calculated from the difference between the OSN spike frequency during and before odor stimulation, as described[76], from which the solvent response was subtracted. Spike amplitude numbers in at1 (Fig. 7d) were scored independently by two experimenters blind to the genotype. Information on odor stimuli and the paraffin oil solvent is provided in Supplementary Table 4.

### Statistics and reproducibility

RNA FISH and IHC experiments were repeated at least twice to confirm the observed phenotypes. *n* values indicated underneath RNA FISH and IHC experiments correspond to the number of antennae or brains quantified. Box plots illustrate individual data points overlaid on boxes showing the median (thick line), first and third quartiles, while whiskers indicate data distribution limits. For quantification of RNA FISH and IHC experiments, letters above boxes indicate significant differences ($P < 0.05$, two-sided Wilcoxon rank sum test followed by Bonferroni correction for multiple comparisons). In pairwise comparisons, a two-sided *t* test was used, and the exact *P* values are indicated on the figure panels. For quantification of gene expression levels, a two-sided Wilcoxon rank sum test (followed by Bonferroni correction for multiple comparisons, where applicable) was used for statistical tests indicated above the boxes.

### Reporting summary

Further information on research design is available in the Nature Portfolio Reporting Summary linked to this article.

## Data availability

Raw sequencing data are archived in the NCBI Gene Expression Omnibus (GEO) under accession code GSE300078. Processed single-cell transcriptomic atlases are available from SCope (https://scope.aertslab.org/#/Drosophila_Developing_Antenna/). All other data supporting the findings of this study (i.e., quantifications of electrophysiological and histological experiments) are included in the Source Data. Biological materials generated in this study are available from the corresponding author upon request. Source data are provided with this paper.

## Code availability

Code used for standard data processing will be provided upon request.

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

## Acknowledgements

We are very grateful for the invaluable assistance of the UNIL Flow Cytometry Facility and Lausanne Genomic Technologies Facility, and thank Gwénaëlle Bontonou and Roman Arguello for advice on HCR FISH, and Kristopher Davie for assistance with data provision to SCope. We acknowledge Darren Williams, Romain Levayer, the Vienna *Drosophila* Resource Center, the Bloomington *Drosophila* Stock Centre (NIH P40OD018537), and the Developmental Studies Hybridoma Bank (NICHD of the NIH, University of Iowa) for reagents. We thank Nikos Konstantinides and members of the Benton laboratory for discussions and comments on the manuscript. Research in K.M.'s laboratory was supported by a National Institutes of Health award (R35GM133209). Research in R.B.'s laboratory was supported by the University of Lausanne, an ERC Advanced Grant (833548), and the Swiss National Science Foundation (310030_219185).

## Author contributions

J.M. conceived the project and performed most experiments and analyses, as well as supervised S.C., who performed histological experiments. A.S.B. performed electrophysiological experiments in at1. D.L. contributed to initial snRNA-seq sample preparation and analyses. P.C.C. generated the *Ir75d promoter-CD4:tdGFP* transgenic line. A.J. performed electrophysiological experiments in ac4, with input from and supervision by K.M. R.B. conceived and supervised the project. J.M. and R.B. wrote the paper with input from all co-authors.

## Competing interests

The authors declare no competing interests.
