## [Transparent Peer Review file · Nature Communications]

Multilayer regulation underlies the functional precision and evolutionary potential of the olfactory system

Corresponding Author: Professor Richard Benton

Version 0:

Reviewer comments:

Reviewer #1

(Remarks to the Author)

In *Drosophila*, antennal sensory neurons are organized beneath sensilla. Although sensilla can house four olfactory sensory neurons (OSNs), fewer are typically observed because programmed cell death kills several of the OSNs during pupal development. To investigate whether programmed cell death influences the formation of new OSN types, Mermet and colleagues generated a single-nucleus atlas of developing and apoptotic antennal OSNs.

Using a nuclear-GFP reporter line and the baculovirus P35 caspase inhibitor, they successfully profiled approximately 54,000 and 32,000 nuclei from control and apoptosis-inhibited samples, respectively, across pupal developmental stages. This dataset is unprecedented in scale compared to previous work by McLaughlin et al., which employed Smart-seq2 and yielded only ~4,000 cells. The authors effectively annotated their dataset using olfactory receptor markers and sensillum types through iterative methods, providing valuable biological context.

Leveraging this dataset, the authors examined transcription factor expression patterns across OSN lineages and identified a set of transcription factors that specify distinct OSN and their Or expression (e.g. *Iz*, *Ibe*, and *Ibl*). Additionally, they investigated the promiscuous expression of certain olfactory receptors, showing that their restriction is correlated to the absence of co-receptors.

The next step was to assign undead neuron to a lineage: They know which Ors are expressed in OSNs from a given sensillum, and they identify genes that are specific to this sensillum. By comparing the control dataset with the apoptosis-inhibited dataset, this allows them to identify "undead" OSNs and to assign them to a specific sensillum. These undead neurons resemble transcriptionally and cluster with their sister/cousins from the same sensillum. Differential gene expression analysis in the *ac3/II*, *sacIII* and *ac4* shows that the OSNs that should have died have high *mamo* expression. This suggests that *mamo* encodes a transcription factor that promotes apoptosis in these lineages. They can show the same thing in *at1* in which case *slp2* is the gene responsible for the PCD of the OSN.

This study provides an invaluable resource for understanding OSN lineage specification and the molecular mechanisms underlying the precise regulation of olfactory receptor expression.

This manuscript is highly relevant to the field of olfactory system development and beyond and should be published in Nature Communications.

However, there are several points that the authors should address before publication, if not with experiments, at least through extensive discussion of the issues.

Major comments:

1. The major issue is the very poor recovery of undead neurons in their snRNAseq. The paper focuses on ~7 lineages but there are many more lineages where one or several OSNs die and they do not seem to be present in the dataset. As the authors acknowledge in the text, this is surprising given the widely accepted mechanism that almost all sensilla typically contain 1-3 OSNs programmed to die. The authors propose several possible explanations for this observation:

- A. Incomplete inhibition of programmed cell death,
- B. Failure to assign some undead cells to specific sensilla, and
- C. Progenitors failing to undergo a final cell division, thereby preventing the detection of undead neurons.

While these possibilities are plausible, the authors should provide experimental evidence to assess which of these

mechanisms might contribute to the observed phenomenon in their dataset.

Specifically, the following questions should be addressed:

A. The first point is: where are the very many cells with high PCD index shown in Fig4B? The authors should analyze whether OSNs do survive when p35 is overexpressed (i.e. use DCP1 to see whether cells are there). More importantly, they should block PCD upstream, e.g. RHG RNAi..... Or there might be non-apoptotic cell death (unlikely in this case)! The reviewer acknowledges this would require redoing the whole experiment, while that would not change the conclusions of the paper! But this should be discussed.

Another possibility is a technical issue during the preparation of nuclei as P35-apoptotic nuclei might be more fragile and not recovered.

B. Are there more undead cells detected within glial clusters?

C. One possibility is that the progenitor dies before the last NotchON/NotchOFF division, which would explain why no OSN is recovered. This could be tested in sensilla where two or more OSNs die. Also, are there any cells in the dataset that resemble undead progenitor cells? It is difficult to envision progenitor cells persisting without undergoing either terminal differentiation or apoptosis.

Providing evidence to support or refute these possibilities would help the readers better understand the potential limitations of the dataset and the biological implications of the findings.

2. To construct the single-nucleus atlas of developing and dying OSNs, the authors dissected fly antennae at different developmental stages and grouped them into three main time points: early, mid, and late. The experimental design appears well thought out. However, according to the Methods section (lines 733–739), the sample from the 80 h APF time point in the control genotype was pooled with the mid-stage samples. Additionally, the developmental phases in Figure 1D were inferred using both library information and pseudotime analysis. This important detail should be explicitly mentioned in the main text to ensure that readers fully understand the dataset composition and the basis of its analysis.

3. The authors observed promiscuous expression of olfactory receptors. For instance, Or35a is weakly expressed in ac4 OSNs, which are expected to exclusively express Ir84a. The authors concluded that the ectopic expression of Or35a is unlikely to be functional due to the absence of Orco expression, as inferred from the dot plot in Figure 3A. However, I am uncertain how reliably the dot plot can determine Orco expression levels with high specificity. Many OSN types in the dot plot express odorant receptors (Ors), yet show very weak Orco expression (e.g., Or85a/Or33b). To strengthen their argument, the authors should validate whether Orco is indeed absent in these OSNs using antibody staining or in situ hybridization.

4. Additionally, the co-expression of Ir31a in Or47a and Or67d neurons appears unexpected. Do the authors believe this observation reflects a true biological signal? It is possible that these signals result from contamination by other cell types or from potential doublets that were not fully removed by the computational pipeline. The authors should comment on this possibility or conduct further experiments to confirm whether the observed expression patterns are biologically relevant. Please indicate whether these neurons express both Irco's and Orco?

Minor comments:

1.Methods section (lines 749–752): The sequencing configuration is set to 100 cycles while the v1.5 kit provides 38 additional cycles for dual indexing, this results in a total of 138 cycles. This could potentially impact sequencing quality, particularly in the final cycles, as it slightly exceeds the total number of bases sequenced ($28+10+10+91 = 139$). Did the authors encounter any issues related to sequencing quality, or is the kit robust enough to handle a slightly higher number of cycles?

2.Figure S3B: It would be more informative if the authors included the co-receptors Orco, Ir8a, and Ir25a in the dot plot to further confirm the identities of the clusters.

3.Fig.1G and Fig.S7A: What neuron precursor type markers did the authors use to calculate the score presented in these figures? The authors should provide a list of these markers.

4.Line 251-252: the citation of "Stocker et al., 1993" is missing from the references.

5.Fig.3H: The hierarchical clustering of cell types based on their transcription factors is intriguing. Would the authors consider modifying the row order in Figure S8 to align with the hierarchical clustering presented in Figure 3H? While this may not introduce additional biological insights, it would make it easier for readers to discern transcription factor patterns across different cell types.

6.Fig.S14: The authors detect undead neuron populations that do not express any receptors. Do these neurons express co-receptors? Additionally, in the case of ab5 and sacl sensilla, it is difficult to determine whether the observed undead neurons are genuine or simply low-quality nuclei from the PCD-blocked dataset. Are there any studies indicating that these two sensillum types undergo apoptosis of Naa or Nbb neurons?

7. In the abstract, the authors claim that program cell death act as a safeguard to remove "empty" neurons lacking receptors, based on the findings in Fig.S14. This statement is inaccurate as Orco mutant OSNs can survive despite lacking functional receptors. The authors should revise this claim for accuracy.

8. Line 499-501: The statement of mamo in mushroom body is incorrect. mamo is only detected in postmitotic neurons and specifies the α/β Kenyon cells, as reported by Liu et al. The authors should revise this statement accordingly.

Reviewer #2

(Remarks to the Author)

The manuscript titled “Multilayer regulation underlies the functional precision and evolvability of the olfactory system” by Mermet et al. examines the development and differentiation of olfactory sensory neurons in the *Drosophila* antennal lobe. The work has three major advances: (1) It establishes a comprehensive single cell transcriptomic atlas not only for the standard differentiation of sensory neurons but also for those destined for programmed cell death, providing a richer picture of antennal sensory lineage specification. (2) Although transcriptional programs reliably set up receptor expression, the data show that these programs are not as precise as canonically assumed. The inherent variability—or developmental noise—is masked mainly by subsequent cell death, yet it creates heterogeneity in the final neuronal populations. This noise is posited to provide the raw material for evolutionary change. (3) Key transcription factors that regulate programmed cell death (PCD) upstream of the reaper/hid pathway are identified, providing insights into how lineage-specific transcriptional control refines both receptor choice and cell fate such that only appropriately specified neurons contribute to the mature olfactory system.

Framing, context, and significance:

The manuscript’s title and framing emphasize evolutionary implications, yet aside from Figure 8L, there is no direct integration of evolutionary experiments. While the findings suggest a mechanism that could potentially facilitate evolutionary change, we recommend that you temper these claims in the title and abstract. Consider discussing these implications as possibilities rather than a central theme unless additional supporting evidence can be provided.

In line with the evolutionary framing, integrating the concepts of duplication and subfunctionalization mentioned on lines 115-117 back into the discussion would improve the coherence between these sections.

We recommend that you more thoroughly ground your discussion section in existing scholarship to provide context for your findings and clarify how your work builds upon or diverges from previous research.

On line 257, the authors indicate that the cell atlas is “functionally predictive”, yet the clustering has not been verified, and no functional studies of the cell types have been performed. Clarification on what “function” means in this context is needed. The description of “eavesdropping” (lines 312–315) is unclear. A more detailed explanation is warranted.

Methods, data, and analyses:

The pooling of cells from the 80 hr APF timepoint with the “mid” stage in the control dataset (lines 736–737) and the subsequent extrapolation of timepoints based on the PCD-blocked dataset (lines 825–827) is a significant methodological detail. We strongly recommend that this be clearly stated in the main text.

If programmed cell death (PCD) is posited to occur between 22–32 hrs APF (lines 141–142), the rationale for not collecting cells in a single time block that captures this window should be addressed. Additionally, Figure 1B depicts PCD as occurring from ~24–38 hrs APF, contrasting with the main text.

The integration of control and PCD datasets before clustering raises questions. Would it not be more robust to cluster cell types in the control dataset first and then map the PCD data onto these clusters? If the current approach is necessary due to the developmental time pooling, please elaborate on this decision.

The unexpectedly high OSN marker gene score in a portion of the cluster designated as Johnston’s organ neurons (Figure S1B) is confusing and may suggest that the initial clustering did not sufficiently differentiate cell types. Additional explanation is warranted for discerning readers.

In Figure S2A, the presence of long tails in many score distributions suggests that some neurons may have been misassigned to time clusters. A better explanation of the use and interpretation of the AddModuleScore function in the methods would be valuable.

Overall, the need for iterative clustering due to poor sequencing depth (a limitation of all single-cell sequencing and not particular to this study) could be explained more clearly. Currently, one has to dig into the iterative clustering in the methods to understand why, in specific figures (such as Figure 2B/D) the highlighted clusters in the UMAPs are not identical to the cells expressing the component genes.

In Figure S4A, the legend does not explain the discrepancy between the cluster representing Or47b (left) and the Or47b neurons on the right (which include some, but not all, of cluster 72). These inconsistencies indicate either unclear explanations of the clustering process or errors in data handling/processing.

It is unclear why, in some cases, neurons with the highest marker gene scores were not assigned to that cluster (e.g., clusters 43 and 0 in the middle panel of S4B and cluster 76 in the bottom panel of S4D). Particularly for cluster 76 in S4D, it is unclear how it was manually assigned to Or88a rather than Or65a/b/c when the score is similar for both. We suggest using a quantitative metric rather than manual assignment to improve cluster assignment fidelity. More worryingly, this indicates that marker gene scores do not unambiguously differentiate cluster assignment (vital to all further experiments).

Relatedly, how are sensillar boundaries identified (such as in Figure 3C)? Lines 959-960 in the methods indicate that sensilla can be identified by their morphology, location, and odor responses. So, how are they identified in cases where odor responses are not evaluated?

The exclusion of PCD and dying neurons in Figure 1G but not in other analyses requires clarification. In addition, please elaborate in the methods of the identification and assignment of lineage classes (Naa, Nab, Nba, Nbb).

In Figure 2A/C, the genes expressed in only ~20% of neurons classified as being of that type is not explained by the inclusion of early time points (since the % is low even in the late timepoint row of Figure 2C). If the explanation is that the iterative clustering process classifies neurons not expressing the marker gene as belonging to its type, then making this more straightforward for the reader in the text and figure legend would be helpful.

The paragraph beginning on line 281 is not supported by the shown data. Figure 3A is referenced to show Or35a expression in Ir84a neurons. But, 3A does not show such expression. The text says that Or35a expression in Ir76a neurons is not demonstrated in 3A but found with in situ stains. However, 3A does show Ir76a expression.

In Figure 4D, the absence of statistical analysis despite claims of significant differences (lines 378–380) should be addressed.

In Figure 5C, the data identifying undead cluster 2 as belonging to the Nbb lineage does not appear statistically significant when compared with Or35a/lr76a.

In Figure S12B, showing gene expression across time would be more informative—one would expect the cell death marker genes to be lowly expressed in the control dataset at late time points.

In Figure S12C, further explanation is needed regarding why certain neuron classes show reduced abundance in the PCD-blocked dataset. Since new clusters are excluded from these analyses, the relative reduction in neuron number is not intuitive.

For Figure S15, quantification of cell numbers would aid in visually conveying the differences observed.

For the electrophysiology experiments (Figure 7C/D), we recommend an automated sorting of spike amplitudes rather than manual scoring. The amplitude thresholds should be depicted in the figure, ideally with higher magnification.

What are the recordings of evoked activity adding versus just examining spontaneous activity?

It is also unclear whether the quantification in Figure 7D pertains to spontaneous activity, evoked activity, or both.

The finding that the proportion of neurons within the at4 sensilla differs between *Drosophila melanogaster* strains is interesting but lacks sufficient controls to verify the claimed significance. For example, differences in neuron numbers in other sensilla not thought to be affected by Slp2 are not evaluated. Additionally, since these inbred lab strains have been reproductively isolated for hundreds of generations, the difference in sensillar composition may have an undiscovered adaptive effect. This interpretation is supported by the relatively constrained proportions of these neurons within strains. It is not evaluated whether the same neurons die between individuals or if a random subset die.

Lines 564-565 indicate that Slp2 has “minimal or no effect” on Or43a. However, Figure 8C shows a significant effect.

Clarity:

The terminology “lineage” for cells in a sensilla (e.g., the text in Figure 2), implies that the cells are born from a single stem cell parent. However, this is not shown nor cited in the manuscript. In other cases, lineage refers to the Naa, Nab, Nba, and Nbb cells born from a single SOP. By convention, all cells from a single SOP would constitute a lineage, and cells born from different SOPs would be different lineages. The term lineage should be used carefully and consistently.

In Figure 1F (and others), please provide appropriate references if the membership of cells in each sensilla is based on prior knowledge (as indicated on lines 809–810).

In Figures 2H/K (and others), the methods do not indicate how the control populations (here, Or67b and Or43a) were selected.

Based on prior research, the authors acknowledge that observed co-expression of specific transcripts does not reflect true gene expression but is an error (lines 272-275). We appreciate the openness with which the authors share this limitation. However, the text could be edited for clarity as it was currently confusing.

Minor inconsistencies hamper readability for readers not previously knowledgeable about the fly antennal lobe. Some specific cases:

“Co-receptors” defined in line 81 do not include Ir93a, while this receptor is considered a co-receptor later on.

It is unclear whether the “antennal sensory neurons” in line 83 are a synonym for OSNs (introduced in lines 74-75) or refer to OSNs and other neurons in the antennal region.

The morphological classes of ASNs listed in line 85 omit the sacculus and arista. Additionally, the terms introduced do not match the sensilla types depicted in the right panel of Figure 1A.

While line 233 states that Figures 2B/D show temporal information, those figures do not convey this information.

In figure 3A, a cell type of interest is labeled as Or35a/lr76a, whereas it is instead referred to as Or35a in the main text (line 282). This confuses the reader as to whether the same cell type is being described.

In Figure 3C, please clarify what the arrows indicate and verify whether “lr75d” is a typographical error for “lr76a.”

On lines 286-288, Figure 3A is referenced to show that Orco protein is not expressed in the cells of interest. However, the figure shows transcript expression, not protein expression.

On line 927, please specify whether the “PB solution” used in the immunohistochemistry experiments is different from the PBS mentioned earlier.

In the paragraph beginning on line 421 (associated with Figure 5), the cell cluster Or35a/lr76a is considered part of ac3 sensilla. However, in a previous section (lines 284-285), this same cell cluster was said to be a part of the ac4 sensilla.

Additionally, several figures and captions would benefit from additional detail or minor modifications:

Consider including the main finding or takeaway for each set of panels in the captions so that readers need not reference the text to understand the significance.

The duplication of Figure S3A (right panel) and S4A (left panel) with a different coloring scheme was slightly confusing. We recommend using the same colors and mentioning when a figure is a duplicate of a previously shown figure.

In Figure 1H, consider revising the Y-axis label to “Expected OSN ratio based on in situ experiments.”

The scales for dot sizes in dot plots in Figure 2 and others should be made consistent across panels to allow for comparison.

For Figure 2A, clarity would be improved by reordering the rows such that the relevant cell types are on the bottom.

Alternatively, highlight the relevant cell types.

Only Figure 2H/K uses asterisks for significance rather than letters. Maintaining consistency in symbols between figures will improve readability.

In Figure 3A, please highlight the rows and columns mentioned in the text.

In Figure 3F (and others), please match labels of cell types between panels (i.e. always exclude or include sensillar class).

In Figure 4B, highlighting the unique “undead” populations would be helpful.

In Figure 5 C/J (and others), different Y-axes make comparisons difficult. Consider using the same range across panels.

In Figure 9, are the top two rows from the mid and late phases missing from the early phase by error or because they are not born yet?

We hope that these detailed comments help clarify the points of concern. We believe that addressing these issues will substantially improve the clarity, reproducibility, and overall impact of your manuscript.

Reviewer #3

(Remarks to the Author)

This study investigates the paradox of functional precision and evolutionary flexibility in the olfactory system of *Drosophila melanogaster*. By creating a single-cell transcriptomic atlas of developing antennal sensory lineages, including neurons that undergo programmed cell death (PCD), the authors reveal that transcriptional control over sensory receptor expression is robust but not entirely perfect. The research highlights three layers of precision: receptor subunit expression, intersection of functionally interacting receptors, and PCD patterning. Additionally, lineage-specific transcriptional control of PCD contributes to variability in neuronal numbers. The findings suggest that while developmental mechanisms drive the stereotypy of the olfactory system, variability in receptor expression and PCD may provide the flexibility necessary for evolutionary adaptations, such as the emergence of new sensory pathways.

Overall, this is an important study and this work offers novel insights into the molecular dynamics of the *Drosophila* olfactory system, particularly how latent, dying neurons may contribute to the evolution of new cell types. The use of high-resolution single-cell transcriptomics to generate an atlas of developing antennal sensory lineages, including latent populations destined for programmed cell death, is a valuable approach that opens up new avenues for investigating the evolution of sensory systems.

However, the paper could benefit from a more explicit discussion of the broader implications of these findings in other species and sensory systems, particularly how the observed developmental variability may influence the evolution of olfactory receptor diversity. Additionally, a deeper exploration of the molecular pathways involved in PCD and identified TFs (Mamo and Slp2) expression would strengthen the impact of the study.

Below are my specific comments:

1. Clustering and cell type annotation are not well described and presented:
 - a. Fig 1C: A better approach would be to first show the neuronal (OSN, JON) and support cell lineages with well-defined markers, and then specifically focus on the OSN lineage.
 - b. The Luo lab has previously profiled 24h and 42h sorted OSNs using Smart-seq2 for higher quality data. How do the current datasets align with these previous datasets?
 - c. The cell type-specific clusters are noisy (Fig S1D). For example, the OSN marker genes label many cells outside of the support/epithelial cells, and the Johnston's Organ Neuron (JON) marker genes show high expression in OSNs. These observations may indicate poor cell quality or data processing. This needs to be improved and clarified.
2. Fig 2K: The authors observed an interesting olfactory receptor loss phenotype. Is this phenotype due to just OR loss or OSN cell death (a developmental defect)? Are these genes only required for OSN development, or required to maintain OR expression in adult as well? It would be useful to perform an adult-specific RNAi experiment to solve this.
3. Fig 3A: The expression of the Or35 gene in Ir84a OSNs is not obvious or even visible (it can be seen in Amt neurons). It is unclear how this co-expression is identified.
4. Fig 3: The authors should more carefully check whether co-expression is observed at the cluster level or cell level. For example, in Fig 3F, Ir31a seems to be expressed in the same cluster as Or47a or Or67d, but not in the same cells. If this is the case, it will lead to a different interpretation of the data. This issue has been discussed in the Fly Cell Atlas (Fig S7B).
5. Fig 4F: With UAS-p35, it appears that there are a few additional Or85f green dots on the left edge of the antenna, away from the main sensillar area. What are these cells? Are they true Or85f+ cells? Could they contribute to the increase in Or85f+ cells, rather than blocking cell death in the ab10 Or67a/Or85f sensilla?
6. The authors show that Slp2 is specifically expressed in the "undead" cluster in Fig 7B. The expression pattern is specific and convincing. What about Mamo expression in Fig 6? Is its expression pattern also specific?
7. To further understand the mechanism, it will be necessary to investigate the epistatic regulation between the identified transcription factors (Mamo and Slp2) and apoptosis. For example, could UAS-MamoRNAi block UAS-Rpr (or UAS-hid) induced PCD in Fig 6? Could UAS-P35 rescue UAS-Slp2-induced PCD for Or67d in Fig 7F-G?
8. I do not entirely agree with the authors' interpretation of Fig 8J, specifically the comparison of three "wild-type" strains. If the variation in Or65a/b/c neurons among them is a result of different Slp2 levels (as the authors suggest), induced by different genetic backgrounds (W1118, OreR, and Canton-S), then the result doesn't necessarily support the idea of "promiscuous" receptor expression. More strictly, if the authors observe such variation within the same genetic background, it would better support the idea of "promiscuity."

Reviewer #4

(Remarks to the Author)

Reviewer #5

(Remarks to the Author)

Version 1:

Reviewer comments:

Reviewer #1

(Remarks to the Author)

This paper received three incredibly detailed and in depth sets of reviews, many of which converge on similar points. Considering the large number of points raised, the authors have done an admirable job at responding honestly and often firmly to the reviewers' many points and it would be difficult to add any more requests or to dispute the answers. I therefore strongly believe that this paper should be published with no further changes as it represents an already very high standards paper with high significance that has been made even more rigorous by the review process. It would be useless to further delay publication.

Reviewer #2

(Remarks to the Author)

Thank you all of my concerns were addressed in this revision.

Reviewer #3

(Remarks to the Author)

The Authors have carefully addressed all my concerns by performing additional experiments and analyses, and added clarification when needed. I support the publication of this work.

Reviewer #4

(Remarks to the Author)

Reviewer #5

(Remarks to the Author)

NCOMMS-25-10413: RESPONSE TO REVIEWERS

We thank the reviewers for their careful reading and constructive criticisms of our manuscript. Below, we provide responses to each of the raised issues. While we have added several new results to the manuscript, and substantially revised the text, there are several cases (for more minor issues raised by only a single reviewer) where we have restricted provision of new data only in Reviewer Figures. The decisions of whether to include new data in the manuscript or this document were made striving to balance responding to the comments of the reviewers but avoiding the manuscript becoming overwhelmingly long and complicated, We are aware that the manuscript is already extremely data-rich, with 8 large figures and 17 Supplementary Figures (and now 7 Reviewer Figures!).

Reviewer #1

In *Drosophila*, antennal sensory neurons are organized beneath sensilla. Although sensilla can house four olfactory sensory neurons (OSNs), fewer are typically observed because programmed cell death kills several of the OSNs during pupal development. To investigate whether programmed cell death influences the formation of new OSN types, Mermet and colleagues generated a single-nucleus atlas of developing and apoptotic antennal OSNs.

Using a nuclear-GFP reporter line and the baculovirus P35 caspase inhibitor, they successfully profiled approximately 54,000 and 32,000 nuclei from control and apoptosis-inhibited samples, respectively, across pupal developmental stages. This dataset is unprecedented in scale compared to previous work by McLaughlin et al., which employed Smart-seq2 and yielded only ~4,000 cells. The authors effectively annotated their dataset using olfactory receptor markers and sensillum types through iterative methods, providing valuable biological context.

Leveraging this dataset, the authors examined transcription factor expression patterns across OSN lineages and identified a set of transcription factors that specify distinct OSN and their Or expression (e.g., *Iz*, *Ibe*, and *Ibl*). Additionally, they investigated the promiscuous expression of certain olfactory receptors, showing that their restriction is correlated to the absence of co-receptors. The next step was to assign undead neuron to a lineage: They know which Ors are expressed in OSNs from a given sensillum, and they identify genes that are specific to this sensillum. By comparing the control dataset with the apoptosis-inhibited dataset, this allows them to identify "undead" OSNs and to assign them to a specific sensillum. These undead neurons resemble transcriptionally and cluster with their sister/cousins from the same sensillum. Differential gene expression analysis in the *ac3/II*, *sacIII* and *ac4* shows that the OSNs that should have died have high *mamo* expression. This suggests that *mamo* encodes a transcription factor that promotes apoptosis in these lineages. They can show the same thing in *at1* in which case *slp2* is the gene responsible for the PCD of the

OSN.

This study provides an invaluable resource for understanding OSN lineage specification and the molecular mechanisms underlying the precise regulation of olfactory receptor expression.

This manuscript is highly relevant to the field of olfactory system development and beyond and should be published in Nature Communications.

However, there are several points that the authors should address before publication, if not with experiments, at least through extensive discussion of the issues.

Major comments:

1. The major issue is the very poor recovery of undead neurons in their snRNAseq. The paper focuses on ~7 lineages but there are many more lineages where one or several OSNs die and they do not seem to be present in the dataset. As the authors acknowledge in the text, this is surprising given the widely accepted mechanism that almost all sensilla typically contain 1-3 OSNs programmed to die. The authors propose several possible explanations for this observation:

- A. Incomplete inhibition of programmed cell death,
- B. Failure to assign some undead cells to specific sensilla, and
- C. Progenitors failing to undergo a final cell division, thereby preventing the detection of undead neurons.

While these possibilities are plausible, the authors should provide experimental evidence to assess which of these mechanisms might contribute to the observed phenomenon in their dataset.

Specifically, the following questions should be addressed:

A. The first point is: where are the very many cells with high PCD index shown in Fig4B? The authors should analyze whether OSNs do survive when p35 is overexpressed (i.e., use DCP1 to see whether cells are there). More importantly, they should block PCD upstream, e.g., RHG RNAi..... Or there might be non-apoptotic cell death (unlikely in this case)! The reviewer acknowledges this would require redoing the whole experiment, while that would not change the conclusions of the paper! But this should be discussed.

RESPONSE: The reviewer makes several points in this comment, which we address in turn:

“where are the very many cells with high PCD index shown in Fig4B?”: In Figure 4b, we do not annotate the lineages where we see high PCD score, as these are

dissected in more detail at the level of individual sensilla in subsequent figures, revealing that undead cells can be embedded in known lineages (Figure 4) or form new cell types (Figure 5 and Supplementary Figure 15). We systematically analysed every lineage, so there are in fact no “unmapped” undead cells in Figure 4b. As a summary of these analyses, and also in response to a comment from Reviewer 2, we now provide in Supplementary Figure 13d an annotated version of the UMAP from Figure 4b in which we highlight the various undead neuron types we were able to annotate.

“The authors should analyze whether OSNs do survive when p35 is overexpressed (i.e., use DCP1 to see whether cells are there).”: we were not fully sure we understand this question, as we show in multiple experiments in this paper that we observe additional, surviving cells when P35 is expressed (e.g., Figure 4f, 4i, 5e, 5l). Moreover, in our previous work (Prieto-Godino et al., 2020) we provided extensive examples of additional neurons when PCD is blocked, both in individual lineages as well as through global quantification of neurons (through Elav staining) (Figure 1 and Figure 3 of that paper). We are therefore confident that OSN PCD is blocked by p35, which is the most routinely used tool for this purpose in *Drosophila* for several decades. As we commented in the text, it is possible that P35 does not block PCD efficiently in all lineages. In this context, we note a discordance between our previous observation of additional Or43a and Or19a neurons *in situ* in P35-expressing antennae (Figure 3B in (Prieto-Godino et al., 2020)) and our failure to detect such cells in our PCD-blocked atlas. One possibility is that our snRNA analysis fails to detect *Or43a* and *Or19a* transcripts in undead OSNs (due to dropout); another is that the presence of an additional UAS transgene (*UAS-unc84:GFP*) in the flies used to generate the snRNA-seq atlases results in a dilution of the Gal4 and so a reduction of efficiency in induction of *UAS-p35* expression in snRNA-seq experiment compared with the *in situ* analyses (where there is no *UAS-unc84:GFP* transgene). We feel such technical speculations are rather complex, hence our decision to simply make a concise statement that P35 might not block PCD efficiently in all lineages.

Following the reviewer’s comment, we examined the temporal expression of *p35* and *unc84:GFP* transgenes in our atlases (**Reviewer Figure 1**). Notably, while *unc84:GFP* transcripts (detecting the exogenous *GFP* reads) are expressed at fairly uniform levels throughout development, *p35* displays comparatively low expression in early stages (when pro-apoptotic gene expression is highest). While it is impossible to know how this might relate to P35 protein levels, this observation suggests that latency in *p35* transcript induction might be a reason why PCD is not efficiently inhibited in all lineages.

Reviewer Figure 1. UMAPs of the developmental phases, *RGS* score, *p35* and *unc84:GFP* expression in OSNs of the control and PCD-blocked datasets, and violin plots of *RGS*, *p35* and *unc84:GFP* expression in these datasets in each developmental phase.

“More importantly, they should block PCD upstream, e.g., *RHG RNAi*...”: We have performed such an experiment in our previous paper (Prieto-Godino et al., 2020), analysing global neuron numbers in the antenna when *rpr*, *grim* and *hid* are down-regulated by RNAi. As we mostly used P35 to block PCD in that study, we continued with this genetic tool in the present work. We note that based upon our analysis of the expression of pro-apoptotic genes in the atlas (Supplementary Figure 13), it appears that *rpr*, *grim* and *skl* (but not *hid*) would be the relevant pro-apoptotic genes; to our knowledge, there is unfortunately no single transgene that permits their simultaneous RNAi. More importantly, given the apparent latency of *peb-Gal4* induced *p35* transcription described above (**Reviewer Figure 1**), it is unclear whether pro-apoptotic gene RNAi would be globally effective. In short, it is difficult to predict which is the superior tool for universal blocking of PCD. However,

we do not think this changes the conclusions that can be made from the undead neuronal lineages that we have been able to identify in the current work.

While repeating the snRNA-seq of antennae in which PCD is blocked using RNAi of pro-apoptotic genes is certainly beyond the scope of this revision (as the reviewer noted above, it would entail repeating the entire experiment), we have examined several of the key undead neuron phenotypes arising from blocking PCD with the *rpr/grim/hid* RNAi transgenic fly (Lee et al., 2013) (**Reviewer Figure 2**). In brief, with this genetic manipulation, we detect (i) undead Or65a neurons in at4 sensilla, (ii) undead Ir75d neurons in ac3 sensilla, and (iii) an additional spike amplitude in at1 sensilla. These results confirm that the cells are normally fated to die through a canonical PCD pathway. (We add that the penetrance of the undead neuron phenotypes are comparable to those of the P35 experiment, arguing that P35 is as good a tool as *rpr/grim/hid* RNAi.)

Reviewer Figure 2. Histological and electrophysiological detection of undead neurons resulting from pro-apoptotic gene RNAi. **(a)** RNA FISH on whole-mount antennae of PCD-blocked animals using *rpr/grim/hid* triple RNAi line (*peb-Gal4/+;UAS-miRGH/+*) with probes targeting the indicated transcripts. Scale bars, 25 μ m. Quantifications are shown below (neuron numbers re-plotted from Figure 4i for *peb-Gal4/+* and *peb-Gal4/+;UAS-p35/+* genotypes). Blocking PCD with either genetic tool results in similar expansion of Or65a neurons compared to control. **(b)** RNA FISH on whole-mount antennae of PCD-blocked animals using the same tools as in (a), with probes targeting the indicated transcripts. Scale bars, top row 25 μ m, bottom row 10 μ m (projection and zoom of 3 adjacent Z-slices in the ac3 area). Quantifications are shown below (neuron numbers re-plotted from Figure 5f for *peb-Gal4/+* and *peb-Gal4/+;UAS-p35/+* genotypes). Blocking PCD with either genetic tool results in Ir75d undead neurons in ac3 sensilla – as with use of P35 (Figure 5e) – although we note that the total number of Ir75d undead neurons is higher when using UAS-p35 to block PCD (potentially hinting at a role of *skl* in promoting PCD of some Ir75d neurons). **(c)** Traces of spontaneous (top) and cVA-evoked (bottom) electrophysiological activity from an at1 sensillum

in the antenna of PCD-blocked (*peb-Gal4/+;UAS-miRGH/+*) animals. Red and blue arrows indicate spikes of the Or67d and undead neuron(s), respectively. Quantification of the proportion of at1 sensilla housing 1 or >1 spike amplitude (as assessed from traces of spontaneous activity) in the indicated genotypes. Data re-plotted from Figure 7d for *peb-Gal4,UAS-Dcr-2* and *peb-Gal4/+;UAS-p35/+* genotypes. Blocking PCD with either genetic tool results in the appearance of undead neurons in at1 sensilla.

(a,b) A-C letters indicate significant differences: $P < 0.05$ in pairwise comparisons (Wilcoxon rank sum test followed by Bonferroni correction for multiple comparisons).

“Or there might be non-apoptotic cell death (unlikely in this case)!”: formally, there might be a caspase-independent cell death of some OSNs, but this seems rather unlikely, and we felt that including this remote hypothesis only unnecessarily lengthens an already-long paragraph in the results.

Another possibility is a technical issue during the preparation of nuclei as P35-apoptotic nuclei might be more fragile and not recovered.

RESPONSE: This is a good idea, and we have added this possible explanation to the text. It is unclear, of course, why certain classes of P35-rescued undead neurons should be more susceptible than others (to account for why we see undead neurons in certain sensillar lineages but not others), but we feel it is not necessary to speculate further than we do already.

B. Are there more undead cells detected within glial clusters?

RESPONSE: The reviewer raises the good point that many antennal glia (which might be derived from Nbb precursors) undergo PCD, specifically those in the *ab/at/ai* lineages (Rodrigues and Hummel, 2008; Sen et al., 2004; Sen et al., 2005). We examined glial cells in our atlases and indeed find that many have a high RGS score (Supplementary Figure 13e-f). However, we did not detect clusters of glia cells specific to the PCD-blocked dataset (nor do we detect late-stage cells in the “dying” cluster), suggesting that P35 does not block PCD in these cells. This might be because of the temporal pattern of *peb-Gal4*-induced P35 expression (**Reviewer Figure 1**).

C. One possibility is that the progenitor dies before the last NotchON/NotchOFF division, which would explain why no OSN is recovered. This could be tested in sensilla where two or more OSNs die. Also, are there any cells in the dataset that resemble undead progenitor cells? It is difficult to envision progenitor cells persisting without undergoing either terminal differentiation or apoptosis.

RESPONSE: If the reviewer’s comment here is related to our statement in the text: *“It is conceivable that the canonical OSN lineage is not universal and that some sensilla housing two neurons result from lack of a final cell division in the lineage, rather than PCD of two of the daughters of such a division.”*, we wish to clarify that we did not propose that the progenitor dies before the last Notch-ON/Notch-OFF division, but rather that this division might *not* occur and the progenitor differentiates as a single mature neuron. In this scenario there is no neuron that

undergoes PCD that could be rescued by expression of P35. The alternative scenario suggested by the reviewer (a progenitor dies prior to the final division) is actually very unlikely because all two-OSN sensilla exhibit markers of Nab and Nba cells (Benton et al., 2025), which derive from different progenitors. In any case, we did not observe undead cells that display obvious characteristics of progenitor cells in our datasets.

Regarding how to test whether there are deviations from the canonical lineage model: this is far from trivial and goes beyond the scope of this work, simply because we do not have genetic tools that would allow us to label specific SOP lineages to examine the patterns of division and PCD giving rise to particular sensilla. The first lineage mapping of OSNs from (Endo et al., 2007) was a massive, painstaking effort using random (heat-shock-based) labelling; while this study established the canonical pattern of SOP lineages, it would not be possible to determine if there might be deviations in some lineages, in particular because they are not perfectly synchronized across all SOPs. There is one exception: an at1 SOP lineage driver we described in (Chai et al., 2019); here we could follow the patterns of cell divisions and eventual death of 3 of the OSN precursors producing the final 1-OSN sensilla, but such insights are limited to this lineage.

Providing evidence to support or refute these possibilities would help the readers better understand the potential limitations of the dataset and the biological implications of the findings.

RESPONSE: We hope to have addressed the reviewer's concerns in our comments, new experiments and additions to the text described above. As we state in the text, it was surprising to us not to have recovered more classes of undead neurons in our PCD-blocked atlas, but it is far from easy to distinguish between the several possible (and non-mutually exclusive) technical and/or biological reasons for this. Regardless, we do not think this limitation fundamentally changes the conclusions of the work, which is the first to characterise particular classes of undead neurons at the single-cell level, highlight heterogeneous types of PCD, and identify the first TFs required to promote PCD in specific lineages (as well as many other insights on the development of surviving lineages).

Future work, for example, generating a completely new atlas with a different tool to block PCD, or through identification of specific early OSN lineage markers might help resolve some of the open questions in completing our identification of all the (presumed) cells fated to die. However, we find it more interesting to delve into the mechanistic details of how PCD is induced in specific lineages, which is now possible with our identification of the roles of *mamo* and *slp2* in *Ir75d* neurons and *at1* sensilla neurons, respectively.

2. To construct the single-nucleus atlas of developing and dying OSNs, the authors dissected fly antennae at different developmental stages and grouped them into three main time points: early, mid, and late. The experimental design appears well thought out. However, according to the Methods section (lines 733–739), the sample from the 80 h APF time point in the control genotype was pooled with the

mid-stage samples. Additionally, the developmental phases in Figure 1D were inferred using both library information and pseudotime analysis. This important detail should be explicitly mentioned in the main text to ensure that readers fully understand the dataset composition and the basis of its analysis.

RESPONSE: We now flag this exception in the experimental design more explicitly in the main text, but prefer to refer the reader to the Methods to provide the full details, rather than break up the flow of the text with this technical issue that, ultimately, does not impact the downstream analyses.

3. The authors observed promiscuous expression of olfactory receptors. For instance, Or35a is weakly expressed in ac4 OSNs, which are expected to exclusively express Ir84a. The authors concluded that the ectopic expression of Or35a is unlikely to be functional due to the absence of Orco expression, as inferred from the dot plot in Figure 3A. However, I am uncertain how reliably the dot plot can determine Orco expression levels with high specificity. Many OSN types in the dot plot express odorant receptors (Ors), yet show very weak Orco expression (e.g., Or85a/Or33b). To strengthen their argument, the authors should validate whether Orco is indeed absent in these OSNs using antibody staining or in situ hybridization.

RESPONSE: We acknowledge the low detection of *Orco* in our snRNA-seq atlas, but the claim of absence of Orco in Ir84a neurons is entirely concordant with previous expression studies, which do not detect *Orco* RNA in coeloconic neurons *in situ* (e.g., (Benton et al., 2009)). To solidify this claim, we have now performed the recommended experiment (**Reviewer Figure 3**), which reveals no detectable Orco protein in Ir84a neurons.

Reviewer Figure 3. No expression of Orco in Ir84a neurons. Immunostaining for Orco and GFP in whole-mount antennae of *Ir84a^{Gal4}/UAS-CD8:GFP* animals (n = 10). Scale bar, 10 μ m.

4. Additionally, the co-expression of Ir31a in Or47a and Or67d neurons appears unexpected. Do the authors believe this observation reflects a true biological signal? It is possible that these signals result from contamination by other cell types or from potential doublets that were not fully removed by the computational pipeline. The authors should comment on this possibility or conduct further

experiments to confirm whether the observed expression patterns are biologically relevant. Please indicate whether these neurons express both Irco's and Orco?

RESPONSE: To further investigate the expression of *Ir31a* in Or47a and Or67d neurons, we have now performed *Ir31a* RNA FISH in antennae in developing and mature antennae (Figure 3g-i). Strikingly – and consistent with the snRNA-seq data – we detect *Ir31a* in many more neurons during pupal development than in adult antennae. Moreover, at least some of these neurons are co-labeled with probes for *Or47a* and *Or67d* in developing, but not adult, antennae, validating the transcriptomic observations as a true biological signal. This experiment is, to our knowledge, the most detailed *in situ* temporal analysis of receptor expression, which reveals a remarkable case of ectopic developmental receptor expression that becomes refined to the mature pattern in adult antennae.

Regarding co-receptor expression: in our snRNA-seq dataset, we reliably detect *Orco* in Or47a neurons, but only very weakly in Or67d neurons (Supplementary Figure 10, 15i and Figure 5r). The latter observation likely reflects the known weak expression of *Orco* in trichoid neurons (Benton et al., 2006); nevertheless, it is well-established that Or67d neurons require *Orco* for their physiological function (e.g., (Jin et al., 2008)). We do not detect expression of the *Ir31a* co-receptor *Ir8a* in either Or neuron population, which is why we consider *Ir31a* expression to be ectopic (and only during development, now shown explicitly in Figure 3g-i). As mentioned in the text: “*neither of these neuron populations respond to Ir31a-dependent ligands (de Bruyne et al., 2001; Munch and Galizia, 2016; van der Goes van Naters and Carlson, 2007). This is not due to the absence of the co-receptor Ir8a (which is never detected in these neuron populations (Supplementary Fig. 10 and (Li et al., 2022)), as ectopic expression of the Ir8a co-receptor in Or67d neurons failed to confirm sensitivity to the best known Ir31a agonist, 2-oxopentanoic acid (Fig. 3j).*”

Minor comments:

1.Methods section (lines 749–752): The sequencing configuration is set to 100 cycles while the v1.5 kit provides 38 additional cycles for dual indexing, this results in a total of 138 cycles. This could potentially impact sequencing quality, particularly in the final cycles, as it slightly exceeds the total number of bases sequenced ($28+10+10+91 = 139$). Did the authors encounter any issues related to sequencing quality, or is the kit robust enough to handle a slightly higher number of cycles?

RESPONSE: We thank the reviewer for the comment, which led us to discover a small typographical error in our Methods: there should be 90 not 91 cycles read2. We have now corrected this issue. This configuration is standard for Illumina NovaSeq and so hopefully assuages the reviewer's concern.

2. Figure S3B: It would be more informative if the authors included the co-receptors Orco, Ir8a, and Ir25a in the dot plot to further confirm the identities of the clusters.

RESPONSE: The expression of the co-receptors has been added to the figure to confirm cluster annotation.

3. Fig. 1G and Fig. S7A: What neuron precursor type markers did the authors use to calculate the score presented in these figures? The authors should provide a list of these markers.

RESPONSE: The list of marker genes defining neuron precursor types are provided in Supplementary Data 1.

4. Line 251-252: the citation of “Stocker et al., 1993” is missing from the references.

RESPONSE: We have added this citation.

5. Fig. 3H: The hierarchical clustering of cell types based on their transcription factors is intriguing. Would the authors consider modifying the row order in Figure S8 to align with the hierarchical clustering presented in Figure 3H? While this may not introduce additional biological insights, it would make it easier for readers to discern transcription factor patterns across different cell types.

RESPONSE: We appreciate the interest of the request but note that Supplementary Figure 8 and Figure 3k (formally Figure 3h) cannot be directly compared as they represent slightly different outputs. In Figure 3k, the hierarchical clustering is calculated based on a PCA reduction in the gene (i.e., differentially expressed TFs) expression distance matrix (as mentioned in the Methods), while the data shown in the dot plot of Supplementary Figure 8 are re-scaled independently for each gene, the default parameter in Seurat. This gene-specific scaling is essential for visualization purposes, otherwise highly-expressed genes would “bleach” out differential-expression of more lowly-expressed genes. (We now mention this standard plotting approach in the legend for the first dot plot of the manuscript (Figure 2a)). Because of the different nature of the analyses and representations shown in Supplementary Figure 8 and Figure 3k, it would not be trivial to extract information on differentially-expressed TFs from Supplementary Figure 8 that might drive phylogenetic similarity. We therefore prefer to keep the row order in Supplementary Figure 8 consistent with all other figures in the manuscript showing gene expression across all OSNs (e.g., Figure 2a, Figure 8a).

6. Fig. S14: The authors detect undead neuron populations that do not express any receptors. Do these neurons express co-receptors? Additionally, in the case of ab5 and sacl sensilla, it is difficult to determine whether the observed undead neurons are genuine or simply low-quality nuclei from the PCD-blocked dataset. Are there

any studies indicating that these two sensillum types undergo apoptosis of Naa or Nbb neurons?

RESPONSE: We have now included the expression of relevant co-receptors in neurons that do not detectably express tuning receptors, both in Supplementary Figure 15 and Figure 5. For Ir neurons, we do detect co-receptor transcripts, while for Or neurons, *Orco* is generally not detected (though we acknowledge this might be drop-out expression).

In ab5 and sacl, we do not think that the undead neuron populations, though small in size, are low-quality nuclei as these were not filtered out during our analyses. It is possible that only a small fraction of them could be rescued upon P35 induction, due to the limitations discussed above.

Finally, regarding the question about whether Naa and Nbb cells undergo PCD in ab5 and sacl lineages: there is no direct evidence, because there are no specific genetic drivers to permit visualization of these individual lineages (see more extended comment about this on pages 7-8). Their precursor identity can only be inferred from the genetic (Endo et al., 2011) and transcriptomic (this study) evidence that the surviving neurons derive from Nab and Nba precursors, thus the dying ones must be Naa and Nbb (which is true of every 2-OSN sensillum).

7. In the abstract, the authors claim that program cell death act as a safeguard to remove “empty” neurons lacking receptors, based on the findings in Fig.S14. This statement is inaccurate as *Orco* mutant OSNs can survive despite lacking functional receptors. The authors should revise this claim for accuracy.

RESPONSE: There seems to be a slight misunderstanding: we are aware that neurons that lack functional receptors due to experimental mutation of *Orco* (or individual tuning receptor genes) do survive. However, the “empty” neurons referred to in the abstract (and illustrated in Supplementary Figure 15, Supplementary 14 and Figure 5) are those that we detected by blocking PCD but which lack expression of a tuning receptor. To our knowledge, there are no examples of surviving “empty” neurons in the wild-type olfactory system. Thus, PCD appears to be critical to remove such cells, given that they would otherwise persist even if they lacked functional receptor expression. We have added a short phrase to the Discussion about this point.

8. Line 499-501: The statement of *mamo* in mushroom body is incorrect. *mamo* is only detected in postmitotic neurons and specifies the α'/β' Kenyon cells, as reported by Liu et al. The authors should revise this statement accordingly.

RESPONSE: We have corrected our inaccuracies, but kept the statement fairly general (“*Indeed, Mamo was previously characterized for its role in defining the fate of Kenyon cells in the mushroom body*”), as we felt that the additional details about post-mitotic expression of *Mamo* and the precise sub-type of Kenyon cells in which it acts were not crucial in this context.

Reviewer #2

The manuscript titled “Multilayer regulation underlies the functional precision and evolvability of the olfactory system” by Mermet et al. examines the development and differentiation of olfactory sensory neurons in the *Drosophila* antennal lobe. The work has three major advances: (1) It establishes a comprehensive single cell transcriptomic atlas not only for the standard differentiation of sensory neurons but also for those destined for programmed cell death, providing a richer picture of antennal sensory lineage specification. (2) Although transcriptional programs reliably set up receptor expression, the data show that these programs are not as precise as canonically assumed. The inherent variability—or developmental noise—is masked mainly by subsequent cell death, yet it creates heterogeneity in the final neuronal populations. This noise is posited to provide the raw material for evolutionary change. (3) Key transcription factors that regulate programmed cell death (PCD) upstream of the reaper/hid pathway are identified, providing insights into how lineage-specific transcriptional control refines both receptor choice and cell fate such that only appropriately specified neurons contribute to the mature olfactory system.

Framing, context, and significance:

The manuscript’s title and framing emphasize evolutionary implications, yet aside from Figure 8L, there is no direct integration of evolutionary experiments. While the findings suggest a mechanism that could potentially facilitate evolutionary change, we recommend that you temper these claims in the title and abstract. Consider discussing these implications as possibilities rather than a central theme unless additional supporting evidence can be provided.

RESPONSE: This comment pertains to how one might present a given set of data, which is inevitably a slightly subjective decision. We acknowledge our manuscript is not steeped in “evolutionary experiments” in the way that our (and many others’) previous work describes differences in the olfactory system across different species of drosophilids (e.g., (Auer et al., 2020; Prieto-Godino et al., 2017)). However, considering how different sensilla *within* a species have evolved distinct properties, for example, in terms of numbers of neurons and receptor expression, are still relevant evolutionary questions. We believe that our emphasis on how sensory systems can be both precisely defined during development but plastic over evolutionary timescales is a still-little addressed, but important, problem bridging neuroscience, evolutionary biology and genetics. In framing our manuscript in this way – as opposed to as purely a developmental study of the *D. melanogaster* olfactory system – our hope was to stimulate work by other researchers in the field.

Nevertheless, to temper the expectations of the reader, we have now changed the title to “Multilayer regulation underlies the functional precision and evolutionary potential of the olfactory system” (rather than “evolvability of the

olfactory system”). We note the abstract is fairly factual in terms of describing the results, and only closes with one, somewhat mild, statement regarding evolutionary implications: “... *might facilitate the evolution of sensory pathways.*”

In line with the evolutionary framing, integrating the concepts of duplication and subfunctionalization mentioned on lines 115-117 back into the discussion would improve the coherence between these sections.

RESPONSE: We discuss these ideas within the pertinent section of the Results describing cases of receptor co-expression. These are quite well-established concepts in the field (for example, we already discussed the phenomena at length in (Ramdya and Benton, 2010), now cited at this point in the Results). As the present work does not substantially add new data to our appreciation of the phenomenon, we felt it less crucial to reiterate these points again in the Discussion (see also comments below regarding this section).

We recommend that you more thoroughly ground your discussion section in existing scholarship to provide context for your findings and clarify how your work builds upon or diverges from previous research.

RESPONSE: We are not fully sure what aspects of the Discussion needed grounding; we have more citations (80) than the normal journal limit (70), and we hope to have referred to most or all of the previous relevant studies (directly, or via recent review articles). We note that much of the context for our work is outlined in the Introduction, and we also include punctual discussion points throughout the Results (for example, earlier work on *lozenge*, *mamo* and *slp2*, previously-described cases of receptor co-expression (and the potential evolutionary significance; see comment above), *Gr* expression in OSNs, our own first examination of the effect of blocking PCD in OSNs, cases of heterogeneous cell numbers in sensilla identified by EM etc.). We feel it is more useful to the reader not to repeat such individual points at the end but rather have a fairly streamlined Discussion that focuses on the main new messages from our work. We have however added a substantial number of references to relevant prior work.

On line 257, the authors indicate that the cell atlas is “functionally predictive”, yet the clustering has not been verified, and no functional studies of the cell types have been performed. Clarification on what “function” means in this context is needed.

RESPONSE: “Functionally predictive” refers to the data in Figure 2, where we use our atlas to identify selectively-expressed TF genes (*Iz* and *Ibe/Ibl*) and show that these TFs have functions specifically within the OSN populations in which they are expressed. We hope that this intention was clear from the opening lines of the pertinent results section: “*As a first assessment of the predictive ability of our developmental atlas ...*”), and are not sure how to more clearly define function. We hope the reviewer will agree that subsequent analyses using the atlas – for example, to identify TFs required to promote PCD – further demonstrate the

usefulness of the atlas in identifying new, functionally-relevant developmental determinants.

The description of “eavesdropping” (lines 312–315) is unclear. A more detailed explanation is warranted.

RESPONSE: We have rewritten this sentence to remove the rather colloquial term “eavesdropping”.

Methods, data, and analyses:

The pooling of cells from the 80 hr APF timepoint with the “mid” stage in the control dataset (lines 736–737) and the subsequent extrapolation of timepoints based on the PCD-blocked dataset (lines 825–827) is a significant methodological detail. We strongly recommend that this be clearly stated in the main text.

RESPONSE: As described in response to Reviewer 1 above, we now flag this exception in the experimental design more explicitly in the main text, but prefer to refer the reader to the Methods to provide the full details, rather than break up the flow of the text with this technical issue that, ultimately, does not impact the downstream analyses.

If programmed cell death (PCD) is posited to occur between 22–32 hrs APF (lines 141–142), the rationale for not collecting cells in a single time block that captures this window should be addressed.

RESPONSE: The timing of PCD is only approximately known from somewhat low spatio-temporal resolution studies using TUNEL staining in the developing antennal disc (Sen et al., 2004) and of one specific lineage (at1) (Chai et al., 2019), which is unlikely to be precisely the same as in other lineages. Given developmental rate can be influenced slightly (\pm a few hours) by culture conditions (nutrition, temperature etc.) and genotype we preferred to sample broadly across the posited period rather than collecting samples within this narrow time window. Moreover, we stress that our study is not only concerned with development analysis of PCD specification, hence the interest to sample both early and much later developmental stages of OSN development. For reasons of cost and experimental practicalities, we pooled samples, rather than sequencing each time point separately; however, we hope to have convinced the reviewer that even when all the cells are integrated together, the developmental changes in the cellular transcriptional profile along individual lineages remain apparent.

Additionally, Figure 1B depicts PCD as occurring from ~24–38 hrs APF, contrasting with the main text.

RESPONSE: We have corrected this graphical representation to be consistent with the text (notwithstanding the uncertainties about the timing of PCD, as described in the response above).

The integration of control and PCD datasets before clustering raises questions. Would it not be more robust to cluster cell types in the control dataset first and then map the PCD data onto these clusters? If the current approach is necessary due to the developmental time pooling, please elaborate on this decision.

RESPONSE: The reviewer raises a good point, but ultimately it is difficult to know which clustering/annotation approach is more robust without pursuing both to completion. The annotations of the integrated dataset took more than a year of work, so it is not a trivial task to compare and contrast different analysis methods. We opted for the clustering of all datasets together, which encompasses the three developmental timepoints of both the control and PCD datasets, in part because we previously used this approach for annotation of homologous cell types in the brain of three different drosophilid species (Lee et al., 2025), where we first integrated the datasets from each species before subclustering and annotation. Importantly, in that analysis, we subsequently “disintegrated” the clusters of the 107 cell types we could annotate into the three separate species datasets, finding that we recovered the same conserved cell types in each species (see S4 Fig in (Lee et al., 2025)). In other words, the initial global integration of all data did not lead to mistakes or missed cell types between species, and we assume this is true for our integration of control and PCD-blocked datasets in the current work. Importantly, our subsequently numerous experiments visualizing and manipulating the development of surviving and undead neurons supports the soundness of this approach.

The unexpectedly high OSN marker gene score in a portion of the cluster designated as Johnston’s organ neurons (Figure S1B) is confusing and may suggest that the initial clustering did not sufficiently differentiate cell types. Additional explanation is warranted for discerning readers.

RESPONSE: The high marker score of OSNs in some Johnston’s organ neurons reflects an expected degree of overlap of gene expression programs of these chemosensory and mechanosensory neuron cell types (e.g., ion channels, neurotransmission molecular machinery), which were extracted from the Fly Cell Atlas within the context of all the other non-neuronal cell types. We have now added the expression pattern of a few highly-selective marker genes for OSNs and Johnston’s organ neurons (and other antennal cell types) within our datasets, which validate our high-level cell type annotation (Supplementary Figure 1d). We also note that the annotated Johnston’s organ neurons do not express the Ir and Or co-receptors characteristic of OSNs.

In Figure S2A, the presence of long tails in many score distributions suggests that some neurons may have been misassigned to time clusters. A better explanation

of the use and interpretation of the AddModuleScore function in the methods would be valuable.

RESPONSE: We have now described in more detail the computation behind AddModuleScore function in Seurat (see the “Cell type annotation” section in the Methods), which returns an enrichment score for expression levels of gene sets. It is actually unsurprising to observe “long tails” in the developmental time marker scores, because most (if not all) genes that belong to the “early”, “mid” or “late” markers display a continuous decreasing, Gaussian or increasing expression pattern along neuron differentiation trajectories, respectively, as opposed to the discrete/abrupt segmentation of cells into early, mid and late categories. Therefore, the long tails simply reflect non-binary expression patterns of developmental time marker genes. We also note that developmental time marker scores are statistically very significantly different across developmental stages.

Overall, the need for iterative clustering due to poor sequencing depth (a limitation of all single-cell sequencing and not particular to this study) could be explained more clearly. Currently, one has to dig into the iterative clustering in the methods to understand why, in specific figures (such as Figure 2B/D) the highlighted clusters in the UMAPs are not identical to the cells expressing the component genes.

RESPONSE: There appears to be a misunderstanding as to why highlighted clusters are not identical to the set of cells expressing the specific genes (in the case of Figure 2b/d, lineage-specific TFs and specific *Or* genes; our arguments below are relevant for all other UMAPs in our work). First, these UMAPs represent a developmental trajectory of cells, and individual genes can have distinct temporal dynamics. As we illustrate in Supplementary Figure 10, *Or* transcription initiates only fairly late in development, hence the restriction more towards the “tips” of individual branches of specific neuronal lineages. Similarly, the TFs peak either in the early (*Iz*) or mid (*Ibl*, *Ibe*) developmental stages. Such temporal dynamics are explicitly shown in the dot plots in Figure 2c,e. Second, although our sequencing depth is comparatively deep ($5e^4$ read pairs/nucleus, cf 10x platform recommendation of $2e^4$ read pairs/cell), sn/scRNA-seq datasets are inevitably constrained by “drop-out” (Qiu, 2020). This phenomenon refers to a certain fraction of cells of a given type not yielding sequencing reads for a given gene – due to low transcript levels or inefficient mRNA capture – even though the entire population of cells is expected to express this gene. Dropout is a widely-acknowledged phenomenon, and explains why the expression of marker genes (*Or*, TFs or others) is not found in all cell types denoted as belonging to a specific lineage.

Regarding the need for iterative clustering, we presented the logic at the beginning of the second results section (“High-resolution annotation of the developing antennal sensory neurons”). We first emphasise that we performed “subclustering” to group cells based upon similarity in their transcriptomes (a very standard operation for analysis of sn/scRNA-seq datasets), then used an “iterative, retrograde annotation method”. This annotation method was to relate cell clusters

for each neuronal lineage at late developmental stages (which express chemosensory receptors robustly) to clusters of cells within the corresponding lineage at earlier developmental stages (when chemosensory receptors are sparsely or not yet detected). Such a method is conceptually similar to that taken by (McLaughlin et al., 2021), where they related OSN populations from three discrete developmental stages. We hope that the first paragraph of this results section is adequate to provide the rationale and basic approach, without overwhelming the reader with the methodological detail and accompanying figures. There are two large Supplementary Figures (Supplementary Figure 3-4) dedicated to illustrating this procedure for several example neuronal lineages.

In Figure S4A, the legend does not explain the discrepancy between the cluster representing Or47b (left) and the Or47b neurons on the right (which include some, but not all, of cluster 72). These inconsistencies indicate either unclear explanations of the clustering process or errors in data handling/processing.

RESPONSE: In Supplementary Figure 4a, there is in fact no discrepancy between the cluster representing Or47b neurons on the left (green dots) and on the right (dark blue dots). In the UMAP on the left, some of the green dots (Or47b cluster) are above some of the light blue dots (72 cluster) and vice versa. By contrast in the right UMAP all the highlighted dots (blue, green and red) are placed at the “front” of the UMAP (using command line “order = TRUE” in Seurat), resulting in them covering up the grey dots in this crowded UMAP. This results in the illusion of partial incorporation of some dots from cluster 72 in Or47b cluster (blue dots in Supplementary Figure 4a, right). In any case, we note that in subsequent iterations (Supplementary Figure 4c), cluster 72 dots are then integrated into the Or47b lineage, as they represent cells from the mid-developmental stage of this lineage (Figure 1d), which do not yet express Or47b.

It is unclear why, in some cases, neurons with the highest marker gene scores were not assigned to that cluster (e.g., clusters 43 and 0 in the middle panel of S4B and cluster 76 in the bottom panel of S4D). Particularly for cluster 76 in S4D, it is unclear how it was manually assigned to Or88a rather than Or65a/b/c when the score is similar for both. We suggest using a quantitative metric rather than manual assignment to improve cluster assignment fidelity. More worryingly, this indicates that marker gene scores do not unambiguously differentiate cluster assignment (vital to all further experiments).

RESPONSE: We first stress that our annotation method used a quantitative metric by using the expression levels of marker genes for OSN/sensilla types to annotated unassigned clusters. Marker gene scores were occasionally similar between clusters, which is not surprising given these are highly related populations of OSNs. Here, “manual” intervention was necessary, as explained in the “Neuron class annotation” section of the Methods: “*In cases of conflicting scoring of various marker lists, we inspected the expression of a few top marker genes, and privileged shortest and continuous differentiation trajectories of lineages*” as well

as in the figure legend to Supplementary Figure 4b: “Where several clusters had a similar marker gene score, we incorporated information from selected marker genes and the relative position of clusters within the UMAP to assign identity; here, for example, *Or47b* neuron marker genes have a high score in 3 clusters (43, 0, 72), but as *Or67d* neuron marker genes also have a high score in clusters 43 and 0 at the same iteration and these two clusters form a continuum with the *Or67d* cluster, they were ultimately assigned to the *Or67d* lineage and cluster 72 (adjacent to the *Or47b* cluster at iteration 0) was assigned to the *Or47b* lineage”.

In the case the reviewer highlights, *Or47b* neuron marker genes have a high score in three-- clusters (43, 0, 72) (Supplementary Figure 4b). However, what we do not show in this figure is that *Or67d* neuron marker genes also have a high score in clusters 43 and 0 at the same iteration, and inspection of the UMAP on the left of Supplementary Figure 4a, it is evident that the cells in clusters 43 and 0 form a continuum with the *Or67d* cluster. This is why clusters 43 and 0 were ultimately assigned to the *Or67d* lineage and cluster 72 to the *Or47b* lineage. The same logic underlies our ultimate assignment of cluster 76 to the *Or88a* lineage: this cluster is immediately adjacent to *Or88a* cluster in the left UMAP in Supplementary Figure 4c.

We hope the above information helps to reassure the reviewer; accurate annotation of the atlas at such (unusually) high resolution was of course a top priority for us, and entailed a painstaking process that took over a year of work. Where ambiguity in identification was unresolved, we did not overinterpret (e.g., the large “unannotated” cluster of early developmental stage cell in Figure 1e). Our validation analyses and experiments in Figure 1h, and Figures 2-7 give us confidence of the accuracy of our annotations. Lastly, in response to a comment from Reviewer 3, we now show in **Reviewer Figure 5** a comparative profile of differentially expressed TFs from a previous (smaller scale) snRNA-seq analysis of developing OSNs (McLaughlin et al., 2021) and from our current work, which reveals excellent agreement in expression pattern of TFs in OSN classes covering all sensilla types.

Relatedly, how are sensillar boundaries identified (such as in Figure 3C)? Lines 959-960 in the methods indicate that sensilla can be identified by their morphology, location, and odor responses. So, how are they identified in cases where odor responses are not evaluated?

RESPONSE: The approximate boundaries were added manually simply for illustrative purposes to aid the reader. The distinct distributions of ac3 and ac4 sensilla have previously been described (Benton et al., 2009; Silbering et al., 2011). In the example of Figure 3c (now 3b), the sensilla to which different *Or35a*-expressing neurons belong to are clearly recognisable because of the additional *Ir84a* marker, i.e., *Or35a*-positive neurons unpaired with *Ir84a*-positive neurons are in ac3, while those paired with *Ir84a*-neurons are in ac4. We have added a comment about this to the figure legend.

The exclusion of PCD and dying neurons in Figure 1G but not in other analyses

requires clarification. In addition, please elaborate in the methods of the identification and assignment of lineage classes (Naa, Nab, Nba, Nbb).

RESPONSE: This is good point; we initially masked the cells from the PCD-blocked dataset in this UMAP because we could not confidently define precursor identity in some dying cells; we masked arisal lineages from the control dataset for the same reason. However, for consistency of cell representation in the UMAPs of Figure 1f and 1g we now show these cells, marking them as having “unassigned” identity.

To classify normally-surviving neurons by precursor type, we used published data that classified most neuron types (Chai et al., 2019; Endo et al., 2011). For the unclassified neurons, we iteratively “scored” precursor type marker genes to assign them to the various classes, as now described in the “Precursor type annotation” section of the Methods. All surviving neuron classes could be assigned to a precursor type, with the exception of arisal neurons (as described in the legend to Figure 1g).

In Figure 2A/C, the genes expressed in only ~20% of neurons classified as being of that type is not explained by the inclusion of early time points (since the % is low even in the late timepoint row of Figure 2C). If the explanation is that the iterative clustering process classifies neurons not expressing the marker gene as belonging to its type, then making this more straightforward for the reader in the text and figure legend would be helpful.

RESPONSE: As described in the response on pages 18-19, the observation that only a fraction of cells of a particular type express a given receptor gene reflects drop-out expression, and not an issue with annotation of cell clusters as a given type (the clustering of cells into neuron types considers the entire detected transcriptome in each cell, not just a few marker genes). This drop-out can be particularly acute if a gene is expressed at low levels.

To illustrate just one case in Figure 2a as highlighted by the reviewer, only around 20% of cells classified as the Or67a “cell type” express *Or67a*. This likely reflects the low expression of *Or67a* (even in late developmental stages); however, there is no ambiguity about annotation. In Figure 2a, one can see that *Or67a* is only detected in the Or67a cell type, and the Or67a cell type only detectably expresses the *Or67a* tuning receptor. Indeed Supplementary Figure 10 illustrates that most receptors are only expressed in a relatively small fraction of their corresponding neuronal population, even at late pupal stages. We suspect this reflects both comparatively low expression of these receptor genes in general, not least because we have not sampled the most mature OSN stages in the adult antennae (although drop-out is still evident at this stage, as can be seen when exploring the Fly Cell Atlas interface for the adult antenna dataset). From *in situ* experiments (both in this work, and a wealth of prior literature examining receptor expression) there is clearly detectable expression of receptor genes in essentially all cells of a given cell type, emphasising that drop-out is simply an artefact of sn/scRNA-seq measurement sensitivity. Returning to Figure 2a, *Iz* is clearly a very

robust and selective marker of the Or67a (and Or85f/Or49a) cell type, detected in ~60% cells. It seems reasonable to assume that the remaining 40% reflect drop-out – just as for the receptor gene transcripts – rather than a more complex model in which there are Iz^+ and Iz^- subpopulations.

We hope our explanation here and above clarifies the issue for the reviewer (which applies to all the dot plot representations in our work). We reiterate that drop-out is general issue of all sc/snRNA-seq experiments. Given the text and legend length limitation and that we already summarise the methodology to identify marker genes in the main text and Methods sections, we are not sure that adding more details would help readers to understand what we call as “marker gene”. We have added a short note to the Figure 2a legend to highlight this issue of “incomplete” expression of marker genes in a given cell population to attune readers unfamiliar with sn/scRNA-seq data to this common issue, although most sn/sc-RNA-seq studies do not.

The paragraph beginning on line 281 is not supported by the shown data. Figure 3A is referenced to show Or35a expression in Ir84a neurons. But, 3A does not show such expression. The text says that Or35a expression in Ir76a neurons is not demonstrated in 3A but found with in situ stains. However, 3A does show Ir76a expression.

RESPONSE: We acknowledge that the expression of *Or35a* in Ir84a neurons is hard to see in in this plot (now only shown in Supplementary Figure 10, bottom), although just visible if one zooms in. This reflects the challenge of globally representing all receptor expression patterns in all neuron types in one plot (which are necessarily scaled the same way for each neuron class). This is why we actually cited Figure 3b (now Figure 3a in this revision) for this statement in the original sentence; the reference to the global plot comes after the phrase “*in addition to its well-described expression in Or35a/Ir76a ac3 neurons (Supplementary Figure 10).*”, which is clearly visible. For expression of *Or35a* in Ir76a neurons, the evidence for this emerged from our *in situ* analysis, and we do not claim we detected it in the transcriptomic dataset. Indeed we originally explicitly suggested that expression of *Or35a* in Ir76a neurons is “*turned on in these cells at a later time point than we have profiled transcriptionally*”. We acknowledge this is a rather complex set of observations to describe, and we strove to accurately cite the relevant figure panels in support of our claims. We now made a few edits to this paragraph to clarify when genes are or are not detected in the global plot in Supplementary Figure 10 to minimise confusion.

In Figure 4D, the absence of statistical analysis despite claims of significant differences (lines 378–380) should be addressed.

RESPONSE: We could not perform statistical analysis because there is a single dataset for the control and PCD-blocked atlases (i.e., “n=1”). We do not claim statistically-significant differences in the text but merely point out the differences in abundance of different neuron populations in our atlases, which motivated the

subsequent more detailed analysis of ab10 and at4 neurons. In these follow-up experiments (Figure 4e-j), we do provide statistical analyses of the differences in neuron population sizes.

In Figure 5C, the data identifying undead cluster 2 as belonging to the Nbb lineage does not appear statistically significant when compared with Or35a/Ir76a.

RESPONSE: The reviewer is correct, and our reasoning for assigning undead cluster 2 as “tentatively Nbb” is as follows: the assignment of Ir75b/c and Or35/Ir76a neurons to Nab and Nba precursor types, respectively, was previously established based upon *in vivo* genetic analysis/lineage mapping, and we now cite the relevant studies in the text (Chai et al., 2019; Endo et al., 2007; Endo et al., 2011). The “undead 1” neuron has statistically significant higher Naa score than co-housed neurons in ac3, hence our assignment as Naa. This leave only the Nbb precursor unassigned, which by parsimony must be Nbb. We acknowledge that the quantitative/statistical scoring of Nbb identity is limited by the relatively small number of known Nbb cells (Supplementary Figure 7b) from which to extract robust markers.

In Figure S12B, showing gene expression across time would be more informative—one would expect the cell death marker genes to be lowly expressed in the control dataset at late time points.

RESPONSE: We now provide such an analysis in **Reviewer Figure 4**, which shows the expression level of the RGS module (a combination of *rpr*, *grim* and *skl* expression) across developmental time in both control and PCD-blocked datasets. This analysis shows that in all developmental phases, the module’s mean expression level is higher in the PCD-blocked compared to the control dataset. The temporal dynamics (“peaking” in the mid timepoint in the PCD-blocked dataset, while at the lowest level in the same timepoint in the control dataset) are, however, hard to interpret, possibly reflecting the low temporal sampling and/or basal expression levels of RGS module genes. We do not think this analysis is particularly informative for the manuscript, hence our decision to present it only here.

Reviewer Figure 4. Average expression for the RGS module at each developmental stage in control and PCD-blocked datasets.

In Figure S12C, further explanation is needed regarding why certain neuron classes show reduced abundance in the PCD-blocked dataset. Since new clusters are excluded from these analyses, the relative reduction in neuron number is not intuitive.

RESPONSE: We assume reviewer refers to the bottom plot of Supplementary Figure 13c (originally Supplementary Figure 12c), which shows the OSN abundance fold-change in the PCD-blocked dataset relative to the control dataset. While it is clear that some OSN populations have an increased relative representation when PCD is blocked (fold-change ~1.5, further validated by RNA-FISH in our study), most neuron classes show somewhat variable fold-change ranging from ~0.8 to ~1.2. We believe that this moderate apparent reduction (or elevation) in the relative representation of neurons does not necessarily reflect true biological signal but rather falls within the error introduced by the low sampling used as an input to calculate these fold-changes (fold-change = (1 PCD-blocked numerical value)/(1 control numerical value)). In this work, we focused only on the most extreme increases as a means to identify undead neurons embedded within clusters of normal cells. We cannot completely exclude that blocking PCD might lead to reductions in size of specific neuron populations, but this would need to be validated *in situ* first, which is beyond the scope of the current work (and we agree that, even if validated, the underlying biological mechanism is not intuitive).

For Figure S15, quantification of cell numbers would aid in visually conveying the differences observed.

RESPONSE: We now present quantifications of the pairing of *Ir75b/c*-positive and *Ir75d promoter:GFP*-positive neurons in Supplementary Figure 16b, which reveals clear novel expression of the *Ir75d* transgenic reporter in ac3 sensilla, similar to the analysis of *Ir75d* RNA expression (Figure 6c).

For the electrophysiology experiments (Figure 7C/D), we recommend an automated sorting of spike amplitudes rather than manual scoring. The amplitude thresholds should be depicted in the figure, ideally with higher magnification. What are the recordings of evoked activity adding versus just examining spontaneous activity?

RESPONSE: We have tried automated spike sorting but found there was too much variation in spike amplitude of the undead neuron(s) for this to be accurate to define (or illustrate) a set threshold for all sensilla. Given any automated method would require manual correction, we felt it more efficient to count manually (a fairly common procedure in the field, to our knowledge). It is precisely because of the variation in amplitude of additional neurons – as well as the partially penetrant phenotype across sensilla – that we had two observers blinded to the genotype (and who did not perform the experiments) to quantify the number of spike amplitudes, to give us confidence in the conclusions.

Regarding the traces of the evoked activity: while we did not quantify cVA-evoked spiking, we felt it was important to show the recordings are of at1 sensilla (whether or not there is a single or >1 spike amplitude), and also confirm that the neuron with the largest-spike amplitude is the cVA-responsive Or67d neuron, consistent with our previous work (Prieto-Godino et al., 2020).

It is also unclear whether the quantification in Figure 7D pertains to spontaneous activity, evoked activity, or both.

RESPONSE: The quantification of spike amplitude number was assessed from traces of spontaneous activity (the density and amplitude “pinching” of evoked spikes would complicate the distinction of activity of different neurons). We have now added a note regarding this point to the figure legend.

The finding that the proportion of neurons within the at4 sensilla differs between *Drosophila melanogaster* strains is interesting but lacks sufficient controls to verify the claimed significance. For example, differences in neuron numbers in other sensilla not thought to be affected by Slp2 are not evaluated. Additionally, since these inbred lab strains have been reproductively isolated for hundreds of generations, the difference in sensillar composition may have an undiscovered adaptive effect. This interpretation is supported by the relatively constrained proportions of these neurons within strains. It is not evaluated whether the same neurons die between individuals or if a random subset die.

RESPONSE: To bolster our original (admittedly speculative) conclusions, we have now evaluated additional sensillar types, shown in new Supplementary Figure 17. In the original manuscript, we showed that a fraction of Or65a neurons is eliminated by PCD (Figure 4h-j) in a Slp2-dependent manner (Figure 8g-i). Because this trait is not fixed across *D. melanogaster* strains, we suggested that this Or65a neuron loss results from “collateral damage” of Slp2-induced PCD machinery activity, which is more pronounced in some strains (i.e., *peb-Gal4* driver line, *w¹¹¹⁸*) than others (e.g., *Canton S*), as shown in Figure 8j-l. Similarly, in ab10 a fraction of Or85f neurons is eliminated by PCD (Figure 4e-g) in a Slp2-dependent manner (Figure 8d-f). We now show that in *w¹¹¹⁸* but not in *Canton S* flies, the numbers of Or85f neurons are also lower than Or67a neurons (Supplementary Figure 17a-b), leaving several unpaired Or67a neurons. These data provide a second potential example of strain-specific, putative “collateral damage” of the PCD-machinery activity. By contrast, as requested by the reviewer, we find that in two other sensilla (ab4 and ab6, where Slp2 is not highly expressed) the 1:1 ratio of the co-housed neurons is equivalent in all strains analysed (Supplementary Figure 17c-f).

We note that there does exist strain-specific differences in the numbers of sensilla i.e., where the population size for both neurons in a sensillum are higher or lower. For example, Or46aB and Or13a neurons in ab6 are both lower in number in *w¹¹¹⁸* compared to *Oregon-R* or *Canton S* (Supplementary Figure 17e-f). This is an interesting but distinct issue, presumably reflecting specification of SOP type in

the antennal imaginal disc in the larva, study of which is beyond the scope of the current work.

The reviewer points out the relatively constrained variability in neuron number within each strain. This property likely reflects the highly inbred nature of these strains, but it does not necessarily imply that the imbalanced numbers of neuron numbers within a given sensillum type is an adaptive phenotype. Of course, we cannot exclude that the differences in neuron number between strains is an adaptive trait of lab “domestication” – and we do mention in the Discussion the possibility of local adaptation of different strains – but it is difficult to imagine what selective pressure in the lab environment would lead to such phenotypic divergence and prefer not to emphasise this idea too strongly.

Finally, as to whether the same or a random subset of Or65a or Or85f neurons die in different individuals: this is a fascinating question that gets at the heart of whether a given OSN population has heterogeneous or homogeneous identity. In the former scenario, it could be that a reproducible subset expresses, for example, slightly more Slp2 – perhaps because of developmental differences in when/where such neurons are specified – leading to their reproducible death. In the latter, the neuron population could be developmentally homogeneous and random developmental noise in Slp2 levels leads to a certain proportion of cells dying by chance. With the data at hand we are unable to distinguish between these possibilities, but this is something we would hope to explore in the future.

Lines 564-565 indicate that Slp2 has “minimal or no effect” on Or43a. However, Figure 8C shows a significant effect.

RESPONSE: Our original phrasing was “*Loss of slp2 had minimal or no effect on Or19a, Or43a and Or69a expression*”, a compact way of indicating a minimal effect (on Or43a) and no effect on Or19a and Or69a, as is apparent from the figure. To more explicitly describe the effect on the different neuron populations, we have now re-phrased this sentence: “*Loss of slp2 had no effect on Or19a and Or69a expression, and led to a small decrease in Or43a neuron numbers (Figure 8b-c).*”

Clarity:

The terminology “lineage” for cells in a sensilla (e.g., the text in Figure 2), implies that the cells are born from a single stem cell parent. However, this is not shown nor cited in the manuscript. In other cases, lineage refers to the Naa, Nab, Nba, and Nbb cells born from a single SOP. By convention, all cells from a single SOP would constitute a lineage, and cells born from different SOPs would be different lineages. The term lineage should be used carefully and consistently.

RESPONSE: We are not entirely sure of the confusion here. First, a sensilla is by definition a lineage, as shown in Figure 1a and explained in the introduction as “*The canonical view is that an SOP gives rise to a short, fixed lineage of asymmetric cell divisions that produces eight terminal cells with distinct molecular identities (Chai et al., 2019; Endo et al., 2007; Endo et al., 2011). Four of these*

eight cells become support cells (which have functions in sensillum construction and secretion of perireceptor proteins (Schmidt and Benton, 2020)), while the other four – termed Naa, Nab, Nba or Nbb – can potentially give rise to OSNs (Chai et al., 2019; Endo et al., 2011)". In Figure 2b, we indicate "ab10 lineage" because we illustrate the developmental trajectory of cells that have terminally differentiated to express Or67a or Or85f, as these mature neurons are housed in a common sensillum (the ab10 sensillum), which was defined previously (e.g., (Couto et al., 2005)); thus we consider the cells highlighted in the UMAP on the left (in red and blue) to be part of the ab10 SOP lineage. There are, of course, several cells in the ab10 SOP lineage that are not highlighted (or even present) in the UMAP on the left, for example, the neuronal precursors that die or the ab10 support cells, which all arise from the same SOP as the Or67a and Or85f OSNs. However, we thought "ab10 lineage" was a more succinct label than "neurons in the ab10 SOP lineage". (The same logic applies for the "ai2 lineage" label in Figure 2d). Although we cited in the Introduction several studies that have documented the molecular organisation of the *D. melanogaster* olfactory system, we now cite two pertinent papers (Benton et al., 2025; Couto et al., 2005) for the well-established molecular organization of ab10 and ai2, which we hope might clarify this point. We have also reviewed our use of the word "lineage" throughout the manuscript to ensure consistent application.

In Figure 1F (and others), please provide appropriate references if the membership of cells in each sensilla is based on prior knowledge (as indicated on lines 809–810).

RESPONSE: Yes, the membership of cells in sensilla has been defined by many studies in the past two decades, as described in the Introduction: "*Intensive anatomical, molecular and functional analyses of the major olfactory organ, the third antennal segment (hereafter, antenna), have defined a highly stereotyped organization in which ~1200 neurons are categorized into nearly 50 distinct classes of olfactory sensory neurons (OSNs), as well as several types of hygrosensory and thermosensory neurons (Benton, 2022; Benton et al., 2025; Couto et al., 2005; Li et al., 2022; Li et al., 2020; McLaughlin et al., 2021; Schlegel et al., 2021; Vosshall and Stocker, 2007) OSNs are grouped in stereotyped combinations of 1-4 neurons underlying sensory hairs (sensilla)*". While we cite again (Couto et al., 2005) and (Benton et al., 2025) for the organization of ab10 and ai2, as described in the response to the comment above, we feel it is unnecessary to do this for every sensillum we describe in the manuscript.

In Figures 2H/K (and others), the methods do not indicate how the control populations (here, Or67b and Or43a) were selected.

RESPONSE: The control receptors were chosen because they are expressed in other small basiconic and intermediate sensilla, respectively. We have now added this information to the main text.

Based on prior research, the authors acknowledge that observed co-expression of specific transcripts does not reflect true gene expression but is an error (lines 272-275). We appreciate the openness with which the authors share this limitation. However, the text could be edited for clarity as it was currently confusing.

RESPONSE: We included this information because of previous misassignment of receptor expression in our own early work (e.g., (Benton et al., 2009)), which has unfortunately perdured in more recent studies (e.g., (Li et al., 2020)) despite our reporting of the distinct protein expression of Ir75a, Ir75b and Ir75c in the interim (Prieto-Godino et al., 2017) . We are not quite sure which part of our explanation was confusing; while we acknowledge our flagging of this issue is compact, we felt it would render this paragraph too long and complex to provide further mechanistic explanation for apparent co-expression, which is fully documented in the cited papers (Mika et al., 2021; Prieto-Godino et al., 2017).

Minor inconsistencies hamper readability for readers not previously knowledgeable about the fly antennal lobe. Some specific cases:

“Co-receptors” defined in line 81 do not include Ir93a, while this receptor is considered a co-receptor later on.

RESPONSE: We have now added Ir93a to the initial listing of Ir co-receptors.

It is unclear whether the “antennal sensory neurons” in line 83 are a synonym for OSNs (introduced in lines 74-75) or refer to OSNs and other neurons in the antennal region.

RESPONSE: We originally use the term “antennal sensory neurons” to cover both OSNs and thermo-/hygrosensory neurons mentioned earlier in this paragraph. This has now been re-phrased to a simpler “sensory neurons”, but again covering all types of antennal neurons, which makes more sense with other revisions to the text (notably adding mention of Ir93a co-receptor for hygro/thermosensory Irs).

The morphological classes of ASNs listed in line 85 omit the sacculus and arista. Additionally, the terms introduced do not match the sensilla types depicted in the right panel of Figure 1A.

RESPONSE: We have revised the text, including introducing the terms sacculus and arista, to better match the schematic.

While line 233 states that Figures 2B/D show temporal information, those figures do not convey this information.

RESPONSE: We cite Figures 2b-e for this point; these figure panels do contain temporal information: the UMAPs in Figure 2b and Figure 2d represent the

developmental atlas composed of three phases (early, mid, late, as illustrated in Figure 1d). However, for full clarity, we extracted expression levels of the TFs for the three developmental phases of these atlases as plotted in Figure 2c and 2e. We thus thought it made sense to cite all four panels, but felt it was not necessary to clutter the UMAPs by indicating developmental time information, given this was more clearly indicated in the dot-plots.

In figure 3A, a cell type of interest is labeled as Or35a/Ir76a, whereas it is instead referred to as Or35a in the main text (line 282). This confuses the reader as to whether the same cell type is being described.

RESPONSE: The ac3 Or35a/Ir76a neurons represent an unusual case of co-expression only very recently described in (Benton et al., 2025), where both Or35a and Ir76a (together with their respective co-receptors) contribute to the functional responses of this neuron. This is distinct from the ac4 “Ir76a/Or35a” neuron, where Or35a is likely non-functional due to the absence of expression of Orco. To avoid confusion between the “Or35a/Ir76a neuron” and “Ir76a/Or35a neuron” terminology, we felt it is clearer to refer to Or35a function in ac4, but we have now rephrased this slightly to refer specifically to the Or35a receptor, rather than the ac3 neuron in which it is expressed: “... *indeed, these neurons do not respond to ligands that activate Or35a in ac3.*”

In Figure 3C, please clarify what the arrows indicate and verify whether “Ir75d” is a typographical error for “Ir76a.”

RESPONSE: We now indicate in the legend of Figure 3b (formerly Figure 3c) that the arrowheads point to ac4 sensilla with two neurons expressing Or35a, one of which co-expresses Ir84a and the other Ir76a.

There was not a typo in “Ir75d” in the original figure; the set of panels was indeed a double RNA FISH for Or35a and Ir75d showing the pairing of Or35a neurons with Ir75d neurons, which provided evidence that Or35a is indeed expressed in non-ac3 sensilla neurons (Ir75d neurons are not present in ac3). On reflection, we realize this set of panels is redundant with the explicit co-expression analysis of Or35a with Ir84a and Ir76a and have now removed this staining. This removal makes the figure clearer and more concordant with the text.

On lines 286-288, Figure 3A is referenced to show that Orco protein is not expressed in the cells of interest. However, the figure shows transcript expression, not protein expression.

RESPONSE: We have corrected this, and also added a citation to (Benton et al., 2009), which provided direct *in situ* evidence for the lack of expression of Orco in these coeloconic sensilla.

On line 927, please specify whether the “PB solution” used in the immunohistochemistry experiments is different from the PBS mentioned earlier.

RESPONSE: We have now described the chemical composition of the PB solution.

In the paragraph beginning on line 421 (associated with Figure 5), the cell cluster Or35a/Ir76a is considered part of ac3 sensilla. However, in a previous section (lines 284-285), this same cell cluster was said to be a part of the ac4 sensilla.

RESPONSE: We appreciate the risk of confusion as it is rather complex! As described above, we now know there are two types of neurons that co-express *Or35a* and *Ir76a*. Only recently characterized in detail in (Benton et al., 2025), ac3 Or35a/Orco neurons also express low levels of Ir76a and its relevant co-receptors (Ir25a/Ir76b); both sets of receptors contribute to the functional odor response properties of these “Or35a/Ir76a” neurons. Second (characterized in this work), ac4 neurons robustly express Ir76a (plus its co-receptors) and, weakly, Or35a but not Orco; these neurons’ response properties are defined only by Ir76a (for simplicity, we refer to these neurons in this manuscript as “Ir76a neurons” as we suspect the *Or35a* expression reflects developmental noise).

Additionally, several figures and captions would benefit from additional detail or minor modifications:

Consider including the main finding or takeaway for each set of panels in the captions so that readers need not reference the text to understand the significance.

RESPONSE: We appreciate this point but feel that re-stating results in the figure legend is redundant with the text and not something typically done beyond the figure title. More importantly, we know that there are quite strict limits on word count in figure legends for *Nature Communications*, so rather strive to keep these as compact as possible and only include the necessary technical detail to understand the figures.

The duplication of Figure S3A (right panel) and S4A (left panel) with a different coloring scheme was slightly confusing. We recommend using the same colors and mentioning when a figure is a duplicate of a previously shown figure.

RESPONSE: Supplementary Figure 3a (right panel) and Supplementary Figure 4a (left panel) are actually not duplicated (we have explicitly mentioned when data are reproduced in two different panels). In addition, Seurat uses a default ordered list of colours so that clusters are coloured according to their ordered size (number of cells). Because the order of cluster sizes is re-shuffled at each annotating iteration, but not the colour vector (the largest cluster is always colour “X”), it will be extremely tedious to manually force the consistency of cluster colours over iteration, because this would require matching two vectors of hundreds of components at each iteration. In addition, we believe that the indicated annotation of clusters (number/OSN/sensilla) over iterations is sufficient for the reader to

navigate UMAPs in Supplementary Figure 4, whose point is to illustrate the iterative annotation of the at4 neurons.

In Figure 1H, consider revising the Y-axis label to “Expected OSN ratio based on in situ experiments.”

RESPONSE: We have now labelled this “Expected OSN ratio from *in situ* data”, for compactness and consistency of phrasing with the x-axis label.

The scales for dot sizes in dot plots in Figure 2 and others should be made consistent across panels to allow for comparison.

RESPONSE: We appreciate this point, but dot plot scales across the manuscript have been chosen to maximise the expression level contrasts within individual comparisons. Uniformising dot plot scales across the entire manuscript would likely have an opposite, more-confusing effect on the conveyed messages for the reader. We also believe there are few, if any, cases where a reader would actually need to compare dot plots *between* figures.

For Figure 2A, clarity would be improved by reordering the rows such that the relevant cell types are on the bottom. Alternatively, highlight the relevant cell types.

RESPONSE: We would rather maintain the same row order that is used for all other dot plots illustrating all antennal neuronal cell types elsewhere in the manuscript. We also considered options for highlighting the relevant cell types (with boxes, arrows or color), but felt these graphical additions were not absolutely necessary (and just added “clutter” to the figure). Given the sparseness of *Iz* and *Ibel/Ibl* expression, these dark blue dots of the data – positioned very close to the cell type names – seem sufficient to “highlight” the relevant cell types.

Only Figure 2H/K uses asterisks for significance rather than letters. Maintaining consistency in symbols between figures will improve readability.

RESPONSE: We have now corrected this inconsistency in Figure 2k, noting that we use asterisks for t-tests (comparisons between two conditions) and letters for Wilcoxon tests (comparisons between >2 conditions) for our quantifications of neuron numbers throughout the manuscript.

In Figure 3A, please highlight the rows and columns mentioned in the text.

RESPONSE: Note that the original Figure 3a was a duplicate of current Supplementary Figure 10, and this panel has been removed to allow space in this main figure for new data, as described above. We tried highlighting the entire row/column of pertinent data (either with boxes or asterisks), but this simply led to cluttering of the plot, not least because we refer to many different cell types both when we first present the receptor expression patterns and in subsequent analyses

in the work, meaning we ending up “highlighting” nearly half of the cell types! On reflection, therefore we decided to keep Supplementary Figure 10 panel as is, without additional highlighting, and believe that our logical ordering of receptor genes will facilitate a reader being able to navigate this dataset relatively easily (potentially using a search function if necessary, which should recognise the text labels).

In Figure 3F (and others), please match labels of cell types between panels (i.e., always exclude or include sensillar class).

RESPONSE: We were not sure about the reviewer’s request to “exclude or include sensillar class” in the previous Figure 3f. The set of UMAPs in current Figure 3e-f are of developing ab5B neurons (which are reported to express *Or47a* and *Or33b* (although we did not detect *Or33b* in our dataset), Fig. 3e) and of developing at1 Or67d neurons (there is no letter suffix on at1 as this sensillum houses only one neuron, Fig. 3f).

In Figure 4B, highlighting the unique “undead” populations would be helpful.

RESPONSE: Figure 4b was meant for the reader to qualitatively visualize the higher expression of the pro-apoptotic RGS module in the PCD-blocked compared to control conditions (further quantified in Figure 4c). Given the small size of Figure 4b and that “undead” neurons found in many various places in these UMAPs, we believe that specifically pointing to “undead” neurons population would unnecessarily crowd this plot. We note that all the “undead” neurons we could discover are illustrated in detail later on in the manuscript. Following the reviewer’s comment, and one also from Reviewer 1, we now provide in Supplementary Figure 13d an annotated version of the UMAP from Figure 4b in which we highlight all the various undead neuron types we were able to annotate as a synthesis of the previous analyses.

In Figure 5 C/J (and others), different Y-axes make comparisons difficult. Consider using the same range across panels.

RESPONSE: The precursor type scores are strongly influenced by the number of marker genes, which are not the same for the four precursor types. Thus, these (and similar) plots are designed to permit comparisons of precursor type score *across* the different classes of neurons, while comparison of precursor scores *within* an individual neuron class is not meaningful. We therefore did not strive to unify the y-axes, but rather adjust them to best accommodate the data ranges for individual precursor types.

In Figure 9, are the top two rows from the mid and late phases missing from the early phase by error or because they are not born yet?

RESPONSE: The top two rows are absent simply because we did not identify any cells of these lineages (Ir21a-arista and Gr28b) from the early developmental phase.

We hope that these detailed comments help clarify the points of concern. We believe that addressing these issues will substantially improve the clarity, reproducibility, and overall impact of your manuscript.

Reviewer #3

This study investigates the paradox of functional precision and evolutionary flexibility in the olfactory system of *Drosophila melanogaster*. By creating a single-cell transcriptomic atlas of developing antennal sensory lineages, including neurons that undergo programmed cell death (PCD), the authors reveal that transcriptional control over sensory receptor expression is robust but not entirely perfect. The research highlights three layers of precision: receptor subunit expression, intersection of functionally interacting receptors, and PCD patterning. Additionally, lineage-specific transcriptional control of PCD contributes to variability in neuronal numbers. The findings suggest that while developmental mechanisms drive the stereotypy of the olfactory system, variability in receptor expression and PCD may provide the flexibility necessary for evolutionary adaptations, such as the emergence of new sensory pathways.

Overall, this is an important study and this work offers novel insights into the molecular dynamics of the *Drosophila* olfactory system, particularly how latent, dying neurons may contribute to the evolution of new cell types. The use of high-resolution single-cell transcriptomics to generate an atlas of developing antennal sensory lineages, including latent populations destined for programmed cell death, is a valuable approach that opens up new avenues for investigating the evolution of sensory systems.

However, the paper could benefit from a more explicit discussion of the broader implications of these findings in other species and sensory systems, particularly how the observed developmental variability may influence the evolution of olfactory receptor diversity. Additionally, a deeper exploration of the molecular pathways involved in PCD and identified TFs (Mamo and Slp2) expression would strengthen the impact of the study.

Below are my specific comments:

1. Clustering and cell type annotation are not well described and presented:
 - a. Fig 1C: A better approach would be to first show the neuronal (OSN, JON) and support cell lineages with well-defined markers, and then specifically focus on the OSN lineage.

RESPONSE: Please see our response to the related comment 1c below.

b. The Luo lab has previously profiled 24h and 42h sorted OSNs using Smart-seq2 for higher quality data. How do the current datasets align with these previous datasets?

RESPONSE: We show in **Reviewer Figure 5** a comparative profile of differentially expressed TFs identified by the Luo lab (McLaughlin et al., 2021) and in our current work. This reveals excellent agreement of expression pattern of TFs in multiple OSN classes covering all sensilla types. Therefore, our current datasets align well with the previous smaller-scale analysis, despite the difference in sequencing approaches (10x Genomics in our work versus Smart-seq2 in the previous study) and depths of coverage.

Reviewer Figure 5. Comparative expression patterns of differentially expressed TFs in (McLaughlin et al., 2021) [reworked from Figure 8C of that paper] and in our current work (control dataset) in the indicated OSN classes representative of the various sensilla types.

c. The cell type-specific clusters are noisy (Fig S1D). For example, the OSN marker genes label many cells outside of the support/epithelial cells, and the Johnston's Organ Neuron (JON) marker genes show high expression in OSNs. These observations may indicate poor cell quality or data processing. This needs to be improved and clarified.

RESPONSE: The high marker score of OSNs in some Johnston's organ neurons reflects an expected degree of overlap of gene expression programs of these chemosensory and mechanosensory neuron cell types (e.g., ion channels, neurotransmission molecular machinery), which were extracted from the Fly Cell Atlas within the context of all the other non-neuronal cell types. We have now added the expression pattern of a few highly-selective marker genes for OSNs and Johnston's organ neurons (and other antennal cell types) within our datasets, which validate our high-level cell type annotation (Supplementary Figure 1d). We also note that the annotated Johnston's organ neurons do not express the Ir and Or co-receptors characteristic of OSNs.

2. Fig 2K: The authors observed an interesting olfactory receptor loss phenotype. Is this phenotype due to just OR loss or OSN cell death (a developmental defect)? Are these genes only required for OSN development, or required to maintain OR expression in adult as well? It would be useful to perform an adult-specific RNAi experiment to solve this.

RESPONSE: We have now tested these questions, as shown in new data in Supplementary Figure 9 and as described in the text: "... we repeated the TF RNAi experiment in combination with a thermosensitive *tub-Gal80^{ts}* transgene, which blocks Gal4-induced RNAi at 18°C but not at 29°C. In *ab10* sensilla of flies grown from egg-laying to adults at 18°C or 29°C, the numbers of Or67a and Or85f neurons matched those of control or *Iz^{RNAi}* flies, respectively, observed in the previous experiment (Supplementary Fig. 9a-b). When *Iz^{RNAi}* was induced only in adults or only from egg-laying until 48 h APF, we observed an identical phenotype: a partial decrease in Or67a neuron numbers and a complete loss of Or85f neuron numbers (Supplementary Fig. 9a-b). These results suggest that *Iz* functions in both neuronal specification and maintenance of receptor expression in Or85f neurons and similarly (albeit partially) in Or67a neurons. The incompletely penetrant phenotype in the latter neurons might reflect a partly redundant function of *Iz* with other TFs and/or lower efficiency of *Iz* RNAi in these cells. ... In *ai2* sensilla of flies grown from egg-laying to adults at 18°C, the numbers of Or23a and Or83c neurons were similar to those of control flies observed previously (Fig. 2j-k). When *lbe^{RNAi}/lbi^{RNAi}* was induced only in adults, no changes in *ai2* neuron numbers were observed (Supplementary Fig. 9c-d), suggesting the TFs are required for *ai2* OSNs specification but not to maintain receptor expression. However, this conclusion is tempered by our failure to recover *lbe^{RNAi}/lbi^{RNAi}* flies grown at 29°C to examine a developmental function of *Lbe/Lbi*."

3. Fig 3A: The expression of the Or35 gene in Ir84a OSNs is not obvious or even visible (it can be seen in Amt neurons). It is unclear how this co-expression is identified.

RESPONSE: We acknowledge that *Or35a* in Ir84a neurons is hard to see in in this plot (now only shown in Supplementary Figure 10, bottom), although just visible if one zooms in. This reflects the challenge of globally representing all receptor expression patterns in all neuron types in one plot (which are necessarily scaled the same way for each neuron class). This co-expression is much easier to appreciate in the specific UMAPs for the ac4A lineage shown in new Figure 3a and validated *in situ* in Figure 3b.

4. Fig 3: The authors should more carefully check whether co-expression is observed at the cluster level or cell level. For example, in Fig 3F, Ir31a seems to be expressed in the same cluster as Or47a or Or67d, but not in the same cells. If this is the case, it will lead to a different interpretation of the data. This issue has been discussed in the Fly Cell Atlas (Fig S7B).

RESPONSE: Due to the sparse nature of single-cell sequencing data, it is a widely-appreciated phenomenon that low/mid-level-expressed genes “drop-out” from the final gene counts/cell matrices (this is apparent in UMAPs throughout the manuscript where not all cells within a given annotated cluster display detectable transcripts for a particular receptor, even though we know all such neurons should express the receptor. Thus, analysis of co-expression patterns of genes detected in a single cluster can yield many false negatives (i.e., cells that only express one or other gene). For example, in the case of the Or67d neuron cluster mentioned by the reviewer, we did detect some Or67d neurons that co-express *Or67d* and *Ir31a* at the single cell level (**Reviewer Figure 6**), while many clearly do not (and several cells do not express *either* of these receptors); this does not diminish confidence that this cluster is a homogenous cell population. In addition, as described above (page 10), we have now performed RNA-FISH at different antennal developmental stages using probes for *Ir31a/Or67d* or *Ir31a/Or47a*, which provides *in situ* validation of the ectopic developmental expression of *Ir31a* in these Or neuron populations inferred from the snRNA-seq data (Figure 3g-i).

Lastly, the reviewer mentions this issue in the context of Figure S7B in the Fly Cell Atlas paper (Li et al., 2022). That panel illustrates maxillary palp neurons as a single cluster even though we know there are six distinct types of neurons. These types were only visualized by examining the expression of the *Or* genes in distinct subsets of cells. We suspect the palp neurons could not be subclustered because of limited numbers of cells in this dataset, which is much less of a limitation in our atlases, where we could subcluster and identify every known antennal neuron type.

Reviewer Figure 6. Expression of *Or67d*, *Ir31a*, or *Or67d + Ir31a* in *Or67d* neurons (control dataset).

5. Fig 4F: With UAS-p35, it appears that there are a few additional *Or85f* green dots on the left edge of the antenna, away from the main sensillar area. What are these cells? Are they true *Or85f*+ cells? Could they contribute to the increase in *Or85f*+ cells, rather than blocking cell death in the ab10 *Or67a/Or85f* sensilla?

RESPONSE: These fluorescence spots are not cells, but accumulations of fluorescence dye that we observe occasionally in histological stainings of the antenna, often attached to the cuticular surface (as is the case here) or contained within the lumen of cuticular hairs.

6. The authors show that *Slp2* is specifically expressed in the "undead" cluster in Fig 7B. The expression pattern is specific and convincing. What about *Mamo* expression in Fig 6? Is its expression pattern also specific?

RESPONSE: We have now added *mamo* expression pattern across all annotated OSNs in Figure 6g, showing that it is expressed in many neuron classes.

7. To further understand the mechanism, it will be necessary to investigate the epistatic regulation between the identified transcription factors (*Mamo* and *Slp2*) and apoptosis. For example, could UAS-*Mamo*RNAi block UAS-*Rpr* (or UAS-*hid*) induced PCD in Fig 6? Could UAS-P35 rescue UAS-*Slp2*-induced PCD for *Or67d* in Fig 7F-G?

RESPONSE: We are as keen as the reviewer to uncover mechanistic information on if and how *Slp2* and *Mamo* directly activate the pro-apoptotic genes to clarify the molecular cascade leading to PCD of OSNs. However, doing this thoroughly and convincingly would go beyond the scope of the current work (which already represents the culmination of >4-5 years' work).

Nevertheless, we have tested the epistatic interaction of *Slp2* and the PCD machinery as illustrated in **Reviewer Figure 7**. Unexpectedly, over-expression of P35 within the at1 lineage is not sufficient to block *Slp2*-induced PCD of *Or67d*

neurons. However, this experiment is difficult to control and the lack of PCD blockage might arise from technical artefacts. For example, P35 over-expression might not reach a sufficient level or be too late to counteract Slp2-induced caspase activity. In principle, it is possible that some Or67d neurons are eliminated by a PCD-independent but Slp2-induced mechanism, but we feel it would be much too speculative to put forward this provocative idea based on this single experiment. Given the difficulty in interpreting this negative result, we prefer not to include these data in the main manuscript.

Regarding the epistasis experiment with Mamo the reviewer proposes, we were less certain about how we would be able to interpret such an experiment. The pro-apoptotic genes trigger cell death by antagonising the anti-apoptotic protein Diap1 (which inhibits caspase activity). It is hard to imagine how a transcription factor could intervene in such a biochemical pathway. Ectopic expression (sometimes in combination) of pro-apoptotic genes has previously been used as a means of killing cells (e.g., (Zhou et al., 1997)), but even if this worked in a *mamo*^{RNAi} background, it would not directly inform how Mamo impinges on the PCD pathway. We speculate that Mamo might promote PCD by transcriptionally activating pro-apoptotic gene expression, following the dogma of how developmental PCD is precisely patterned. However, we already know it is more complicated than this model, as Mamo is expressed in many OSN populations that do not die (as illustrated in new Figure 6g).

Reviewer Figure 7. Left: RNA-FISH on whole-mount antennae of *at1-Gal4,UAS-p35,UAS-slp2* flies with probes targeting *Or67d* transcripts; scale bar, 25 μ m. Right: quantification of *Or67d* neuron numbers (data for the first three genotypes are replotted from Figure 7g; *n* is indicated underneath. A and B letters indicate significant differences: $P < 0.05$ in pairwise comparisons (Wilcoxon rank sum test followed by Bonferroni correction for multiple comparisons).

8. I do not entirely agree with the authors' interpretation of Fig 8J, specifically the comparison of three "wild-type" strains. If the variation in *Or65a/b/c* neurons among them is a result of different *Slp2* levels (as the authors suggest), induced by different genetic backgrounds (W1118, OreR, and Canton-S), then the result doesn't necessarily support the idea of "promiscuous" receptor expression. More

strictly, if the authors observe such variation within the same genetic background, it would better support the idea of "promiscuity."

RESPONSE: We are not sure if we fully understand this comment, but assume the reviewer is referring to this passage in the Discussion: "*Alternatively, heterogeneous PCD might reflect promiscuity in transcriptional specification of PCD resulting from overlap in gene regulatory networks of surviving and dying lineages, akin to the promiscuity observed in receptor expression.*". The term "akin" was to highlight a potential parallel in the observed phenomena (i.e., heterogeneous PCD and promiscuous receptor expression), without claiming a clear mechanistic link, notably because our documented examples of heterogeneous PCD involves classes of neurons (e.g., Or65a/b/c, Or85f) that are distinct from examples of promiscuous receptor expression (e.g., Or35a). Moreover, in this study, we have not examined interspecific variation in receptor expression, in the same way we examined interspecific variation in PCD, although cases of this will certainly exist (one known example is in reported examples of strain-specific expression of *Or33a/b/c* (see Dataset EV1 in (Benton et al., 2025))). Conversely, one can argue that there is promiscuity in PCD *within* individual strains, based upon the variation in cell numbers observed in any given experiment.

The point we endeavored to highlight in our work is that the *D. melanogaster* olfactory system is not as precisely specified as described "in the textbooks" in terms of neuron numbers and receptor expression patterns. Future determination of the genetic basis of such variation, and whether it can be subject to selection, will be necessary to understand how it might contribute to the evolution of olfactory pathways.

Finally, we take the opportunity to indicate that in this revision we have added similar analyses of neuron numbers in ab10, ab4 and ab6 sensilla in different strains (Supplementary Figure 17), as discussed in more detail on pages 26-27.

Reviewer references

Auer, T.O., Khallaf, M.A., Silbering, A.F., Zappia, G., Ellis, K., Alvarez-Ocana, R., Arguello, J.R., Hansson, B.S., Jefferis, G.S.X.E., Caron, S.J.C., *et al.* (2020). Olfactory receptor and circuit evolution promote host specialization. *Nature* 579, 402-408.

Benton, R. (2022). *Drosophila* olfaction: past, present and future. *Proc Biol Sci* 289, 20222054.

Benton, R., Mermet, J., Jang, A., Endo, K., Cruchet, S., and Menuz, K. (2025). An integrated anatomical, functional and evolutionary view of the *Drosophila* olfactory system. *EMBO Rep* 26, 3204-3225.

Benton, R., Sachse, S., Michnick, S.W., and Vosshall, L.B. (2006). Atypical membrane topology and heteromeric function of *Drosophila* odorant receptors *in vivo*. *PLOS Biol* 4, e20.

Benton, R., Vannice, K.S., Gomez-Diaz, C., and Vosshall, L.B. (2009). Variant ionotropic glutamate receptors as chemosensory receptors in *Drosophila*. *Cell* 136, 149-162.

Chai, P.C., Cruchet, S., Wigger, L., and Benton, R. (2019). Sensory neuron lineage mapping and manipulation in the *Drosophila* olfactory system. *Nat Commun* 10, 643.

Couto, A., Alenius, M., and Dickson, B.J. (2005). Molecular, anatomical, and functional organization of the *Drosophila* olfactory system. *Curr Biol* 15, 1535-1547.

de Bruyne, M., Foster, K., and Carlson, J.R. (2001). Odor coding in the *Drosophila* antenna. *Neuron* 30, 537-552.

Endo, K., Aoki, T., Yoda, Y., Kimura, K., and Hama, C. (2007). Notch signal organizes the *Drosophila* olfactory circuitry by diversifying the sensory neuronal lineages. *Nat Neurosci* 10, 153-160.

Endo, K., Karim, M.R., Taniguchi, H., Krejci, A., Kinameri, E., Siebert, M., Ito, K., Bray, S.J., and Moore, A.W. (2011). Chromatin modification of Notch targets in olfactory receptor neuron diversification. *Nature Neuroscience* 15, 224-233.

Jin, X., Ha, T.S., and Smith, D.P. (2008). SNMP is a signaling component required for pheromone sensitivity in *Drosophila*. *PNAS* 105, 10996-11001.

Lee, D., Shahandeh, M.P., Abuin, L., and Benton, R. (2025). Comparative single-cell transcriptomic atlases of drosophilid brains suggest glial evolution during ecological adaptation. *PLOS Biol* 23, e3003120.

Lee, G., Sehgal, R., Wang, Z., Nair, S., Kikuno, K., Chen, C.H., Hay, B., and Park, J.H. (2013). Essential role of grim-led programmed cell death for the establishment of corazonin-producing peptidergic nervous system during embryogenesis and metamorphosis in *Drosophila melanogaster*. *Biology open* 2, 283-294.

Li, H., Janssens, J., De Waegeneer, M., Kolluru, S.S., Davie, K., Gardeux, V., Saelens, W., David, F.P.A., Brbic, M., Spanier, K., *et al.* (2022). Fly Cell Atlas: A single-nucleus transcriptomic atlas of the adult fruit fly. *Science* 375, eabk2432.

Li, H., Li, T., Horns, F., Li, J., Xie, Q., Xu, C., Wu, B., Kechschull, J.M., McLaughlin, C.N., Kolluru, S.S., *et al.* (2020). Single-Cell Transcriptomes Reveal Diverse Regulatory Strategies for Olfactory Receptor Expression and Axon Targeting. *Curr Biol* 30, 1189-1198 e1185.

McLaughlin, C.N., Brbic, M., Xie, Q., Li, T., Horns, F., Kolluru, S.S., Kechschull, J.M., Vacek, D., Xie, A., Li, J., *et al.* (2021). Single-cell transcriptomes of developing and adult olfactory receptor neurons in *Drosophila*. *Elife* 10, e63856.

Mika, K., Cruchet, S., Chai, P.C., Prieto-Godino, L.L., Auer, T.O., Pradervand, S., and Benton, R. (2021). Olfactory receptor-dependent receptor repression in *Drosophila*. *Science advances* 7, eabe3745.

Munch, D., and Galizia, C.G. (2016). DoOR 2.0--Comprehensive Mapping of *Drosophila melanogaster* Odorant Responses. *Sci Rep* 6, 21841.

Prieto-Godino, L.L., Rytz, R., Cruchet, S., Bargeton, B., Abuin, L., Silbering, A.F., Ruta, V., Dal Peraro, M., and Benton, R. (2017). Evolution of Acid-Sensing Olfactory Circuits in Drosophilids. *Neuron* 93, 661-676.

Prieto-Godino, L.L., Silbering, A.F., Khallaf, M.A., Cruchet, S., Bojkowska, K., Pradervand, S., Hansson, B.S., Knaden, M., and Benton, R. (2020). Functional

integration of "undead" neurons in the olfactory system. *Science advances* 6, eaaz7238.

Qiu, P. (2020). Embracing the dropouts in single-cell RNA-seq analysis. *Nat Commun* 11, 1169.

Ramdya, P., and Benton, R. (2010). Evolving olfactory systems on the fly. *Trends Genet* 26, 307-316.

Rodrigues, V., and Hummel, T. (2008). Development of the *Drosophila* olfactory system. *Adv Exp Med Biol* 628, 82-101.

Schlegel, P., Bates, A.S., Sturner, T., Jagannathan, S.R., Drummond, N., Hsu, J., Serratos Capdevila, L., Javier, A., Marin, E.C., Barth-Maron, A., *et al.* (2021). Information flow, cell types and stereotypy in a full olfactory connectome. *Elife* 10, e66018.

Sen, A., Kuruvilla, D., Pinto, L., Sarin, A., and Rodrigues, V. (2004). Programmed cell death and context dependent activation of the EGF pathway regulate gliogenesis in the *Drosophila* olfactory system. *Mech Dev* 121, 65-78.

Sen, A., Shetty, C., Jhaveri, D., and Rodrigues, V. (2005). Distinct types of glial cells populate the *Drosophila* antenna. *BMC Dev Biol* 5, 25.

Silbering, A.F., Rytz, R., Grosjean, Y., Abuin, L., Ramdya, P., Jefferis, G.S., and Benton, R. (2011). Complementary Function and Integrated Wiring of the Evolutionarily Distinct *Drosophila* Olfactory Subsystems. *The Journal of Neuroscience* 31, 13357-13375.

van der Goes van Naters, W., and Carlson, J.R. (2007). Receptors and neurons for fly odors in *Drosophila*. *Curr Biol* 17, 606-612.

Vosshall, L.B., and Stocker, R.F. (2007). Molecular Architecture of Smell and Taste in *Drosophila*. *Annu Rev Neurosci* 30, 505-533.

Zhou, L., Schnitzler, A., Agapite, J., Schwartz, L.M., Steller, H., and Nambu, J.R. (1997). Cooperative functions of the reaper and head involution defective genes in the programmed cell death of *Drosophila* central nervous system midline cells. *Proceedings of the National Academy of Sciences of the United States of America* 94, 5131-5136.

NCOMMS-25-10413A: RESPONSE TO REVIEWERS

We thank the reviewers for their careful re-reading of our manuscript, and are happy we have been able to assuage their initial concerns.

Reviewer #1 (Remarks to the Author):

This paper received three incredibly detailed and in depth sets of reviews, many of which converge on similar points. Considering the large number of points raised, the authors have done an admirable job at responding honestly and often firmly to the reviewers' many points and it would be difficult to add any more requests or to dispute the answers.

I therefore strongly believe that this paper should be published with no further changes as it represents an already very high standards paper with high significance that has been made even more rigorous by the review process. It would be useless to further delay publication.

Reviewer #2 (Remarks to the Author):

Thank you all of my concerns were addressed in this revision.

Reviewer #3 (Remarks to the Author):

The Authors have carefully addressed all my concerns by performing additional experiments and analyses, and added clarification when needed. I support the publication of this work.